# Global anthropogenic CO₂ emissions and uncertainties as prior for Earth system modelling and data assimilation

Margarita Choulga[1], Greet Janssens-Maenhout[2], Ingrid Super[3], Efisio Solazzo[2], Anna Agusti-Panareda[1], Gianpaolo Balsamo[1], Nicolas Bousserez[1], Monica Crippa[2], Hugo Denier van der Gon[3], Richard Engelen[1], Diego Guizzardi[2], Jeroen Kuenen[3], Joe McNorton[1], Gabriel Oreggioni[2], and Antoon Visschedijk[3]

[1]Research Department, ECMWF, Reading, RG2 9AX, United Kingdom
[2]European Commission, Joint Research Centre, JRC, Ispra, 21027, Italy
[3]TNO, Department of Climate, Air and Sustainability, Utrecht, 3584 CB, The Netherlands

*Correspondence to*: Margarita Choulga (margarita.choulga@ecmwf.int)

**Abstract.** The growth in anthropogenic carbon dioxide ($CO_2$) emissions acts as a major climate-change driver, which has widespread implications across society, influencing the scientific, political and public sectors. For an increased understanding of the $CO_2$ emission sources, patterns and trends, a link between the emission inventories and observed $CO_2$ concentrations is best established via Earth system modelling and data assimilation. In this study anthropogenic $CO_2$ emission inventories are processed into gridded maps to provide an estimate of $CO_2$ emissions for 7 main emissions groups: 1) energy production super-emitters, 2) energy production standard-emitters, 3) manufacturing, 4) settlements, 5) aviation, 6) other transport at ground level and 7) others, with estimation of their uncertainty and covariance to be included in the European Centre for Medium-Range Weather Forecasts (ECMWF) Integrated Forecasting System (IFS). The emission inventories are sourced from the Intergovernmental Panel on Climate Change (IPCC) 2006 Guidelines for National Greenhouse Gas Inventories and revised information from its 2019 Refinements, and the global grid-maps of Emissions Database for Global Atmospheric Research (EDGAR) inventory. The anthropogenic $CO_2$ emissions for 2012 and 2015, (EDGAR versions 4.3.2 and 4.3.2_FT2015 respectively) are considered, updated with improved apportionment of the energy sector (decreased by 8 %) and the energy usage for manufacturing (increased by 18 %), and with newly generated diffusive $CO_2$ emissions from coal mines. These emissions aggregated into 7 ECMWF groups with their emission uncertainties are calculated per country (considering its statistical infrastructure development level) and sector (considering the most typical fuel type) and use the IPCC recommended error propagation method assuming fully uncorrelated emissions. While the uncertainty of most groups remains relatively small (5-20 %), the largest contribution (usually over 40 %) to the total uncertainty is determined by the OTHER group with usually the smallest budget, consisting of oil refineries and transformation industry, fuel exploitation, coal production, agricultural soils and solvents and products use emissions, with uncertainties more than 100 %. Several sensitivity studies are performed: 1) for country type – by analysing the impact of assuming either a well or less-well developed statistical infrastructure, 2) for fuel type specification – by adding explicit information for each fuel type used per each IPCC activity, and 3) for national emission source distribution – by aggregating

all emission sources and evenly redistributing them over the country – highlights the importance of spatial mapping. Uncertainties are compared with United Nations Framework Convention on Climate Change (UNFCCC) and the Netherlands Organisation for Applied Scientific Research (TNO) data. Upgraded anthropogenic $CO_2$ emission maps with their yearly and monthly uncertainties are combined into the CHE_EDGAR-ECMWF_2015 dataset (Choulga et al., 2020) available from https://doi.org/10.5281/zenodo.3967439. CHE_EDGAR-ECMWF_2015 consists of 11 global NetCDF files with gridded yearly and monthly upper and lower bounds of uncertainties in % and $kg \cdot m^{-2} \cdot s^{-1}$ for each ECMWF group and their sum, and 1 Excel file with 16 spreadsheets with the same information listed per country (metadata, emissions, uncertainties, statistical parameters).

## 1 Introduction

Carbon dioxide ($CO_2$) is the most abundant greenhouse gas (GHG) (NOAA, 2019) contributing to the climate change. This study focuses on anthropogenic (man-made) long-cycle carbon $CO_2$ emissions (i.e. emissions from fossil fuel use and industrial processes: cement production, carbonate use of limestone and dolomite, non-energy use of fuels and other combustion, chemical and metal processes, solvents, agricultural liming and urea, waste and fossil fuel fires (Janssens-Maenhout et al., 2019)), that occur on top of an active natural carbon cycle, and generation of a reliable uncertainty band globally for different emission types that can be used in Earth system modelling and data assimilation.

The $CO_2$ growth rate varies from year to year with a tendency toward higher growth rates since the early 2000s. The added $CO_2$ has a long life-time and only a portion of it transfers each year from the atmosphere to the oceans and to vegetation on land. The atmosphere exchanges carbon mainly between: (i) the terrestrial biosphere – impact on growth rate through deforestation and other forms of land management; (ii) the oceans – impact on growth rate through marine ecosystems implications due to $CO_2$ in the form of carbonic acid absorption in surface waters and their mix with deep ocean waters; (iii) the fossil fuels and cement and other $CO_2$ process emissions – when around 1920 fossil fuel burning became the dominant source of anthropogenic emissions to the atmosphere, with a clear increase of 91 ppm in the past six decades (from 316 ppm in 1959 till $407.4 \pm 0.1$ ppm in 2018), according to NOAA (2019).

Accurate assessment of anthropogenic $CO_2$ emissions is important to better understand the global carbon cycle. Efforts towards a global anthropogenic $CO_2$ monitoring and verification support capacity as described by Janssens-Maenhout et al. (2020), rely on atmospheric modelling and atmospheric observations (in-situ from e.g. the Integrated Carbon Observation System, ICOS, air-borne from e.g. aircraft campaigns, or space-borne from e.g. the Orbiting Carbon Observatory, OCO-2, and the Greenhouse gases Observing Satellite, GOSAT). All measurements are assimilated by global tracer transport models to infer atmospheric $CO_2$ changes or by flux inversion systems to estimate the large-scale surface $CO_2$ fluxes. ECMWF for example applies both inverse modelling and direct modelling of global concentrations of $CO_2$ in the atmosphere assimilating several types of observations.

The global transport models require an initial best estimate of the $CO_2$ emission fields with uncertainties, the so-called prior
information. The intensity of the emission fields is corrected through minimization of the difference between the modelled
and measured concentration values for $CO_2$. The uncertainty of these corrected $CO_2$ fluxes based on inverse modelling will
be lower with the increase of $CO_2$ observations and its accuracy. The disentanglement of the fossil $CO_2$ emissions from the
total atmospheric $CO_2$ emissions remains challenging, e.g. in 2018 total anthropogenic $CO_2$ concentrations ($42.5 \pm 3.3$ Gt
$CO_2$) represented only 1.3 % of the global atmospheric $CO_2$ concentration ($407.4 \pm 0.1$ ppm) (Friedlingstein et al., 2019),
which states the need for high accuracy of measurements ($\geq 1.0$ %). Emission fields are often supplied through emission
inventories. Bottom-up emission inventories start from human activity statistics and emission factors are defined for each
activity and provided at international or country level (e.g. National greenhouse gas Inventory Report, NIR). Such bottom-up
inventories need to be gridded and characterised with uncertainties in order to represent a prior data set useful for numerical
modelling. Table 1 shows examples of most used global gridded $CO_2$ emission datasets, for more details see Andrew (2020),
Janssens-Maenhout et al. (2019, Table 3) and Cong et al. (2018, Table 1).

There is strong evidence to suggest fixed annual emissions do not represent sufficient temporal variability, for example,
natural gas consumption has two seasonal peaks, with consumption patterns predominantly driven by weather – the largest
peak occurs during winter due to low temperatures when natural gas is used more to heat residential and commercial spaces,
the smaller peak occurs during summer due to high temperatures when natural gas, coal or petroleum-fired generators are
used to generate more electric power for air conditioning (Bradley, 2015; Comstock, 2020). New version of EDGAR v5.0 is
addressing high temporal disaggregation of emissions (Crippa et al., 2020).

**Table 1: Examples of global gridded anthropogenic $CO_2$ emission bottom-up datasets**

| Name | Resolution | Period | Main assumptions, uncertainties | Source |
|---|---|---|---|---|
| Carbon Dioxide Information Analysis Center (CDIAC) | Spatial: 1.0°×1.0° Temporal: annual, monthly Sectoral: 1 | 1751-2013 | Use population density to disaggregate emissions, the mass-emissions data based on fossil-fuel consumption estimates. Provide gridded annual and monthly uncertainty estimates for 1950-2013 | Andres et al., 1996; Andres et al., 2016 |
| Open-Data Inventory for Anthropogenic Carbon dioxide (ODIAC) | Spatial: 1×1 km$^2$, 0.1°×0.1° Temporal: monthly Sectoral: 6 | 1979-2018 | First introduced the combined use of nightlight data and individual power plant emission/location profiles | Oda and Maksyutov, 2011; Oda et al. 2018; ODIAC, 2020 |
| Emissions Database for Global Atmospheric Research (EDGAR) | Spatial: 0.1°×0.1° Temporal: annual, monthly Sectoral: 26 | 1970-(year-1) | Based on international statistics, covers all IPCC (2006) reporting categories, consistent methodology applied to all the world countries | Janssens-Maenhout et al., 2019 |
| Fossil Fuel Data Assimilation System (FFDAS) | Spatial: 0.1°×0.1° Temporal: annual Sectoral: 2 | 1997-2012 | Provide gridded posterior uncertainty (version 2.2); in addition, provide monthly, weekly, and hourly fractions from annual $CO_2$ emissions | Asefi-Najafabady et al., 2014 |
| Community Emissions Data System (CEDS) | Spatial: 0.1°×0.1° Temporal: annual, monthly Sectoral: 55 | 1750-2014 | Provide emissions of $CO_2$ and other GHGs and pollutants | Hoesly et al., 2018 |
| Peking University Fuel combustion inventory (PKU-FUEL) | Spatial: 0.1°×0.1° Temporal: monthly Sectoral: 6 | 1960-2014 | By request provide daily emissions and the results of Monte Carlo simulation-based uncertainty analyses | Chen et al., 2016; Liu et al., 2015 |

Though there are global anthropogenic emission gridded datasets, most of them have scarce evaluation of uncertainties, which needs enhancement with the relative errors for sector-specific country totals and the uncertainties in trends with the appropriate probability density functions. Only 3 datasets from Table 1 provide uncertainty estimates, namely CDIAC, FFDAS and PKU-FUEL. CDIAC uncertainties have no sectors and include contributions from the tabular fossil fuel $CO_2$ emissions (assigned per 7 country types, values are constant over time), geography map (power plant location), and

population map (has details both in time and space, is used to distribute fossil fuel $CO_2$ emissions). Population map uncertainty strongly dominates in the generated gridded fossil fuel $CO_2$ uncertainties (Andres et al., 2016). CDIAC uncertainties have no sectoral distribution and are presented on 1.0°×1.0° grid. FFDAS provides only posterior uncertainties, which are based on a model inversion. These posterior uncertainties could be used as prior uncertainties for separate inversion systems, however this would make the characterisation of uncertainty more complex if there were similarities in

the model and observations used. PKU-FUEL uncertainty estimates of $CO_2$ emission maps associated with uncertain fuel data and uncertain activity data in the spatial disaggregation process are based on Monte Carlo ensemble simulations. Input data was randomly sampled 1000 times from an a priori normal uncertainty distribution with a certain coefficient of variation: for fuel consumptions from ships/aviation sector coefficient of variation is set to be 20 %, for wildfires sector – 18 %, for all other fuel data – 10 %, for combustion rates – 20 %. Additional coefficient of variation was assigned for each

country or subnational unit based on its size to consider uncertainty of spatial fuel data disaggregation (e.g. coefficient of variation for the largest subnational unit of the world Asian part of the Russian Federation is 1000 %). Emission factor coefficient of variation was constant value of 5 % (Wang et al., 2013). PKU-FUEL uncertainties were heavily based on subjective assumptions and rather detailed information of fuel type, which makes is difficult to use for IPCC (2006) reporting categories.

In this study, we focus on fossil emissions (from fossil fuel combustion, use and production, and process emissions from cement production and others such as glass, chemicals, urea) and we distinguish between point sources and sources with wider spatial distribution. The scope of this research is to generate a reliable uncertainty band on 0.1°×0.1° grid with global coverage based on emission type for the yearly and monthly emission budgets, that are the composite of anthropogenic fossil fluxes, and that are aligned with updated IPCC requirements. Uncertainty characterisation is key for optimally combining the

bottom-up inventories with the top-down data assimilation.

In this study 2015 is chosen as a base year to analyse anthropogenic $CO_2$ budgets (i.e. global, regional, national) from different sources (i.e. global statistics, national reports), benefitting the availability of observations (both in-situ and space-borne) as well as reported and verified emission inventories. Global $CO_2$ emissions from fossil fuel and industrial processes such as cement production reached a total of 36.2 Pg $CO_2$ in 2015 according to EDGAR inventory version 4.3.2_FT2015

(Olivier et al., 2016a). The use of energy represents by far the largest source of emissions (89 % share globally), and in particular the energy industry sector (38 % share) (including both combustion and fugitive gas releases from use but also production, processes, transmission and storage of fuels for energy and heat generation). Another reason for choosing 2015 is that it's the year of the Paris Agreement and the reference year for several Nationally Determined Contributions (NDCs).

Countries have submitted their pledges to the United Nations (UN), setting out how far they plan to reduce their GHG

emissions – NDCs (CarbonBrief, 2020). Yet concentrations are still growing. In 2015, the average concentration of $CO_2$ (399 ppm) was about 40 % higher than in the mid-1800s, with an average growth of 2 ppm/yr in the last ten years. Furthermore, according to JRC 2019 Report (Crippa et al., 2019) between 2015 till 2018, just in three years global $CO_2$ emissions have raised by 4.3 % (1575.2 Mt $CO_2$/yr), of which the international component of $CO_2$ emissions (shipping and aviation bunker fuel) has even raised by 6.3 % (75.1 Mt $CO_2$/yr).

Following the Intergovernmental Panel on Climate Change (IPCC) 2006 Guidelines for National Greenhouse Gas Inventories and revised information from its 2019 Refinements (IPCC-TFI, 2019) we start from the global fossil $CO_2$ grid-maps of EDGAR inventory versions 4.3.2 (Janssens-Maenhout et al., 2019) and 4.3.2_FT2015 (Olivier et al., 2016a), for 2012 and 2015 respectively, and derive an updated emission dataset as prior input to the ECMWF model: CHE_EDGAR-ECMWF_2015 (CHE stands for the $CO_2$ Human Emissions project (CHE, 2020)). We improve the EDGARv4.3.2 dataset by

correcting the allocation of the autoproducers (autoproducers are defined by International Energy Agency (IEA) and include the energy (electricity and heat) generated by an industry for its own use, mostly for the manufacturing) to the manufacturing sector instead of the energy sector and by adding the diffusive $CO_2$ emissions from coal mines. We then aggregate the sectors in 7 emission groups while tracking 232 countries separately. Uncertainties are calculated per country and sector considering the most typical fuel type using the error propagation method of the IPCC (2006) guidelines. According to the

IPCC (2006) guidance all emissions are considered to be fully uncorrelated; this assumption is further used to calculate uncertainty and covariance matrices. The country-based uncertainties and the share to the total uncertainty are presented for the 7 ECMWF emission groups, with calculations based on 20 EDGAR sectors for two distinct country types with well- and less well-developed statistical infrastructures. While the uncertainty of most groups (i.e. power industry, combustion for manufacturing, and road transport) remains small (5-20 %), the largest contribution (over 40 %) to the total uncertainty is

determined by rather small but relative uncertain (more than 100 %) sectors (i.e. non energy use of fuels, chemical processes, fuel exploitation, and coal production) emissions.

This paper is organised as follows. Section 2 describes the data sources and includes the description of the anthropogenic $CO_2$ emission datasets used to calculate emission uncertainties, data pre-processing, emission sectors and groups, and geographical treatment of emissions. Section 3 discusses the uncertainty calculation methodology applied to the datasets, to

calculate both yearly and monthly uncertainties. Section 4 provides details on the newly generated dataset. National sectorial emission budgets are compared in Section 5. The main results, a discussion and further research guidance are covered in the conclusion in Section 6. This paper also has Supplementary Information with details on methods and assumptions used.

## 2 Data

### 2.1 Update of fossil CO₂ emissions as input for the ECMWF model

Main requirements for datasets in order to be used in global numerical models are being global and gridded, and preferably with continuous update. In this study it was decided to use EDGARv4.3.2 (and EDGARv4.3.2_FT2015) because it is based on international statistics, mainly IEA data, has a unique global geo-coverage with 228 countries/regions and continuous updates of the time-series. The most relevant activity data for both EDGARv4.3.2 and CHE_EDGAR-ECMWF_2015 are the energy statistics from IEA (2014), which has been corrected for few outliers and for the revised Chinese coal statistics of

2015. For the update from 2012 to 2015 we used the fast track approach of Olivier et al. (2016b), with IEA (2016) energy statistics and BP (2017) statistics. EDGAR distributes anthropogenic emissions for each source category over a uniform, global 0.1°×0.1° grid defined with lower left coordinates and provides annual and monthly global emissions grid-maps. The bottom-up emissions calculation methodology and emission factors, either defaults recommended by IPCC (2006) guidelines or region-specific ones justified by scientific evidence, are consistently applied to all countries in order to achieve

comparability and full transparency.

We focus on long-cycle carbon $CO_2$ and therefore consider the $CO_2$ from fossil fuel use (combustion and other use of 42 fossil fuels) and from industrial processes (cement production, carbonate use of limestone and dolomite, non-energy use of fuels and other combustion, chemical and metal processes, solvents, agricultural liming and urea, waste and fossil fuel fires). Excluded are consumption of biofuels and short-cycle biomass burning (such as agricultural waste burning), large-scale

biomass burning (such as forest fires, savannah burning, woodland and peatland fires) and carbon emissions/removals of land-use, land-use change and forestry (LULUCF)[1]. Based on the Global Carbon Budget 2018 findings this sector showed no significant trend since 1960s, only high year-to-year variability and high uncertainty (Bastos et al., 2020; Le Quéré et al., 2018; Arneth et al., 2017). We excluded also the fossil fuel fires, because we focus only on 2015, for which the Kuwait oil fires of 1991 are of no importance and the coal mine fires data are considered to be very uncertain.

Starting from EDGARv4.3.2_FT2015, the following updates were considered necessary for the derivation of the CHE_EDGAR-ECMWF_2015 dataset. Firstly, there was a need to reallocate the part of autoproducers in the energy sector to the manufacturing industry in line with UNFCCC reporting. The autoproducers' energy generated and used for industrial manufacturing was added to the manufacturing sector (causing an increase of 18 %) and taken away from the energy sector (leading to a decrease of 8 %). The reallocation of the autoproducers part was done using the energy statistics reported by

every country separately (IEA, 2016) but the correction remained limited to 30 % of the national total energy sector. More details are given in the Supplementary Information, section S.1.

Secondly, super power plants were considered to be treated separately, because they are expected to operate at full capacity with maximum availability. Super power plants are defined in this study as a large power plant or a group of closely located

---

[1] Following the UNFCCC national inventory reporting guidelines, emissions of biofuel combustion are only a memo item and have to be reported under the LULUCF sector. Together with all short-cycle carbon emissions they are excluded from this study.

power plants causing $CO_2$ plumes from a single grid cell with a $CO_2$ flux $\geq 7.9 \cdot 10^{-6}$ kg·m$^{-2}$·s$^{-1}$. According to expert knowledge the upper bound of uncertainty for super power plants is not larger than +3.0 %, whereas for small plants which operate based on day-to-day needs, this can reach up to +15.0 %. Currently 30 grid-cells from 12 countries represent energy generated by the super power plants (7.1 % or 896.7 Mton of the remaining energy sector after autoproducers part separation 12705.5 Mton). Top 3 countries that produce energy using super power plants are China, Russia and India. For the detailed ranking of the power plant sites in function of their emission intensity, we refer to the Supplementary Information, section S.1.

Finally, an extra emission source of fugitive $CO_2$ from coal mines was added, following the recommendations from IPCC-TFI (2019). Even though this emission source is not that large globally, usually the coalseam gas is composed dominantly from methane, but in some coalmines (in Australia, and also in Brazil) seam gas consists predominantly (> 95 %) from $CO_2$ (Beamish and Vance, 1992), leading to significant atmospheric $CO_2$ concentration increases. An additional map for CHE_EDGAR-ECMWF_2015 with coal mining emissions from underground mines has been generated, following the IPCC-TFI (2019) default values and the coal mining activity of the methane ($CH_4$) emission grid-maps from hard and brown coal production of EDGARv4.3.2. More details are given in the Supplementary Information, section S.2, in which Table S3 lists all differences between EDGARv4.3.2_FT2015 and CHE_EDGAR-ECMWF_2015.

The detailed EDGARv4.3.2 spatial distribution is used for mapping the updated 2015 emission values (Janssens-Maenhout et al. (2019) provide all special details on how emissions are spatially distributed and what proxies are used for that in EDGARv4.3.2). The relative changes per sector, fuel type and country from 2012 to 2015 are then applied on the EDGARv4.3.2 reference maps to obtain EDGARv4.3.2_FT2015. For non-energy use of fuels, chemical processes, and solvents and products use we used directly the EDGARv4.3.2 maps. Also, the $CO_2$ emission maps from coal production are based on the 2012 maps of $CH_4$ from EDGARv4.3.2. Gridded monthly multiplication factors are obtained from 2010 monthly gridded emissions and applied to the final set of yearly emission maps of CHE_EDGAR-ECMWF_2015.

## 2.2 Aggregation of $CO_2$ emission groups for the ECMWF model

EDGARv4.3.2_FT2015 (as well as EDGARv4.3.2) has 20 global maps with anthropogenic long-cycle carbon $CO_2$ flux values for energy, fugitives, industrial processes, solvents and products use, agriculture and waste involved sectors. In this study these sectors had to be grouped for the use of global flux inversion and ensemble perturbation systems. Grouping was done keeping in mind possible future evolution of present systems and sector common features: activity type (point sources, 3D field, etc.), amount of knowledge for the activity (uncertainty value), geographical distribution (e.g. over urban areas only), size of sector covariance matrix (computationally affordable size for the inversion system of the ECMWF model is covariance matrix of 7×7). Table 2 shows additional grouping of 20 EDGAR sectors into 7 ECMWF groups, and emission budget difference between EDGARv4.3.2_FT2015 and CHE_EDGAR-ECMWF_2015 datasets due to reallocation of the

autoproducers from the energy sector (-8 %) to the manufacturing sector (+18 %), and due to the extra emission source of diffusive coal mine $CO_2$.

**Table 2: Grouping of anthropogenic long-cycle carbon $CO_2$ emission EDGAR sectors into ECMWF groups, note provides main information and typical fuel type, global emission budgets for 2015 in Mton provides values for EDGARv4.3.2_FT2015 and CHE_EDGAR-ECMWF_2015; _italics_ – values with biggest differences, [*] – values that were replaced from EDGARv4.3.2**

| № | ECMWF group | IPCC (2006) activities per EDGAR sector | Note | Emission budget 2015, Mton | |
|---|---|---|---|---|---|
| | | | | EDGARv4.3.2_FT2015 | CHE_EDGAR-ECMWF_2015 |
| 1 | ENERGY_S | 1.A.1.a (subset) | Power industry (without autoproducers): super emitting power plants | _13704.0_ | _896.7_ |
| 2 | ENERGY_A | 1.A.1.a (rest) | Power industry (without autoproducers): standard emitting power plants | | _11671.6_ |
| | | 4.C | Solid waste incineration | 137.2 | 137.2 |
| 3 | MANUFACTURING | 1.A.2 | Combustion for manufacturing (including autoproducers) | _6182.8_ | _7320.4_ |
| | | 2.C.1, 2.C.2 | Iron and steel production | 233.6 | 233.6 |
| | | 2.C.3, 2.C.4, 2.C.5, 2.C.6, 2.C.7 | Non-ferrous metals production | 91.4 | 91.4 |
| | | 2.D.1, 2.D.2, 2.D.4 | Non energy use of fuels | 24.7[*] | 24.6 |
| | | 2.A.1, 2.A.2, 2.A.3, 2.A.4 | Non-metallic minerals production | 1748.8 | 1749.0 |
| | | 2.B.1, 2.B.2, 2.B.3, 2.B.4, 2.B.5, 2.B.6, 2.B.8 | Chemical processes | 678.8[*] | 677.0 |
| 4 | SETTLEMENTS | 1.A.4, 1.A.5.a, 1.A.5.b.i, 1.A.5.b.ii | Energy for buildings | 3321.9 | 3322.7 |
| 5 | AVIATION | 1.A.3.a_CRS | Aviation cruise; typical fuel: jet kerosene | 412.2 | 412.2 |
| | | 1.A.3.a_CDS | Aviation climbing & descent; typical fuel: jet kerosene | 305.5 | 305.5 |
| | | 1.A.3.a_LTO | Aviation landing & take off; typical fuel: jet kerosene | 97.7 | 97.7 |
| 6 | TRANSPORT | 1.A.3.b | Road transportation; typical fuel: most typical emission factor uncertainty | 5530.2 | 5530.6 |
| | | 1.A.3.d | Shipping; typical fuel: composition of 80 % diesel and 20 % residual fuel oil | 819.0 | 819.1 |
| | | 1.A.3.c, 1.A.3.e | Railways, pipelines, off-road transport; typical fuel: railways – diesel, off-road transport – most typical emission factor uncertainty | 255.2 | 255.2 |
| 7 | OTHER | 1.A.1.b, 1.A.1.c, 1.A.5.b.iii, 1.B.1.c, 1.B.2.a.iii.4, 1.B.2.a.iii.6, 1.B.2.b.iii.3 | Oil refineries and Transformation industry | 1917.4 | 1917.8 |
| | | 1.B.2.a.ii, 1.B.2.a.iii.2, 1.B.2.a.iii.3, 1.B.2.b.ii, 1.B.2.b.iii.2, 1.B.2.b.iii.4, 1.B.2.b.iii.5, 1.C | Fuel exploitation | 258.4 | 258.4 |
| | | 1.B.1.a | Coal production | _0.0_ | _7.0_ |
| | | 3.C.2, 3.C.3, 3.C.4, 3.C.7 | Agricultural soils | 99.0 | 99.1 |
| | | 2.D.3, 2.B.9, 2.E, 2.F, 2.G | Solvents and products use | 168.7[*] | 168.3 |

## 3 Uncertainty calculation methodology

### 3.1 Overview

The IPCC (2006) Guidelines for NIR for fossil $CO_2$ uncertainty calculations and updated IPCC-TFI (2019) provide vast information about numerous human activities emitting $CO_2$ and how certain these values are. Use of the IPCC-TFI (2019) permitted to consider the 2019 emission factor and activity data uncertainties for petroleum refining, solid fuel manufacturing, transformation, processing and transport and oil and gas production, which differed significantly from the 2006 defaults. In order to use the same methodology globally and because $CO_2$ emissions are not technologically dependant,

it was decided to omit regional (e.g. Europe) detailed information and use only information required for the most basic and simplest (Tier 1) approach for emission reporting. The Tier 1 methodology to estimate $CO_2$ emissions from fossil fuel combustion follows the concept of carbon conservation (from the fuel combusted into $CO_2$). Uncertainties for all emission activities, sectors and groups can be derived following two different approaches of IPCC (2006): (Approach 1) propagation of error – gives informative results even if the criterion "standard deviation divided by the mean value is less than 0.3" is not

strictly met and data still have some correlation. The advantages are that it only needs uncertainty ranges for activity data and emission factors, that are provided by IPCC and that it is relatively easy to improve in case of large and asymmetric uncertainties; (Approach 2) Monte Carlo simulation or similar techniques – suitable only if detailed category-by-category uncertainty information is available and complex calculations can be done. In order to use the same methodology for all world countries/geographical entities (i.e. not needing detailed information for each emission activity) it was decided to use

the error propagation method (Approach 1).

To summarize, the final uncertainties per geographical entity per ECMWF fossil $CO_2$ emission group are based on: emission budgets calculated from CHE_EDGAR-ECMWF_2015 maps (upgraded combination of EDGARv4.3.2 and EDGARv4.3.2_FT2015), uncertainty default values from IPCC (2006) and IPCC-TFI (2019), Tier 1 approach (error propagation method) and the definition of a log-normal distribution (needed for non-negative anthropogenic $CO_2$ emissions).

It should be noted that all uncertainty calculations were done per country (geographical entity) and only then for comparison purposes aggregated to Europe (28 members till end 2019) or global values assuming no correlation following IPCC (2006). Figure 1 shows a simplified scheme of the uncertainty calculation roadmap, followed by a detailed description below on how exactly yearly and monthly uncertainties are calculated.

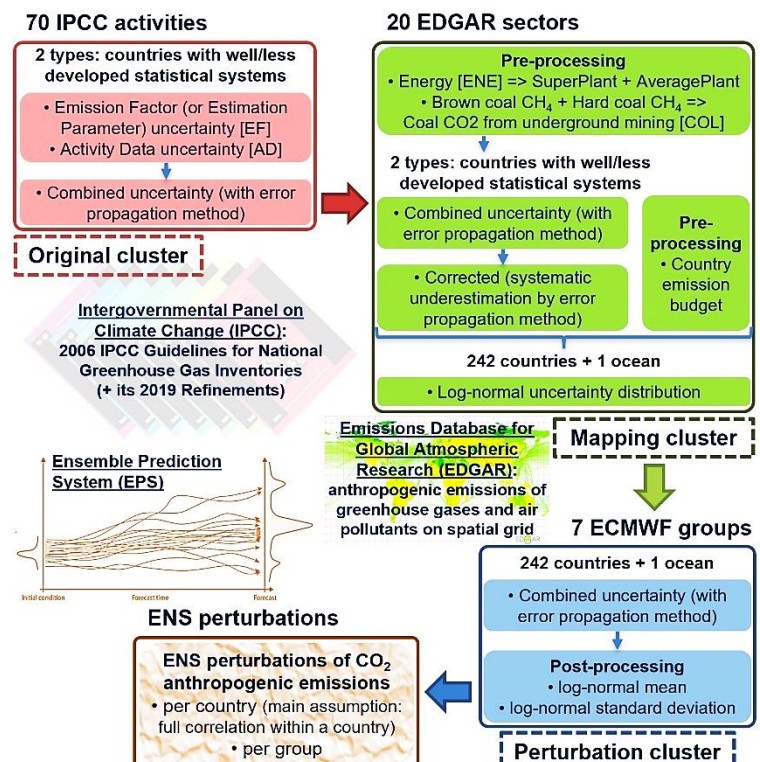

Figure 1: Simplified roadmap for yearly uncertainty calculation

## 3.2 Yearly uncertainties

### 3.2.1 Calculating uncertainty per each IPCC activity

Uncertainties in the emissions per IPCC activity from Table 2 – Combined Uncertainties $UC_{IPCCi}$ – were calculated using uncertainties for emission factors $EF_{IPCCi}$ and activity data $AD_{IPCCi}$ in % provided in IPCC (2006) and IPCC-TFI (2019) following Eq. (1):

$$UC_{IPCCi} = \sqrt{EF_{IPCCi}^2 + AD_{IPCCi}^2}.$$ (1)

It should be noted that IPCC (2006) and IPCC-TFI (2019) provide upper and lower limits of emission factor and activity

data, which are not always symmetrical. In order to preserve as much initial information as possible (and not to inflate artificially lower or upper limits of log-normal emission distributions) all calculations were performed for upper and lower uncertainty limits separately although it is not required by the Approach 1 methodology. Moreover, IPCC (2006) provide default emission factor values for different fuels in transport-related activities (e.g. railways, aviation, etc.). Detailed fuel consumption information per each IPCC activity that result in long-cycle carbon was not available and it was decided to use

the most typical and consumed (common) fuel type (its emission factor value). Table 2 shows the most typical fuels for each transport related sector.

### 3.2.2 Calculating uncertainty for each EDGAR sector

Uncertainties for each of the 70 IPCC activities from Table 2 are calculated with the error propagation method and combined into the 20 EDGAR sectors, following Eq. (2):


$$UC_{EDGARj} = \sqrt{UC_{IPCC1}^2 + UC_{IPCC2}^2 + \dots + UC_{IPCCn}^2}, \tag{2}$$

where $EDGARj$ – combined uncertainty per sector $j$, and $1,2,...,n$ – IPCC activities that are taken into account in a particular EDGAR sector; $UC_{IPCC1}$, $UC_{IPCC2}$,..., $UC_{IPCCn}$ used in %.

### 3.2.3 Correction of EDGAR sector uncertainty due to underestimation by the chosen method

The EDGAR sector uncertainty had to be corrected, as the error propagation method of Approach 1 systematically
underestimates the uncertainty unless the model is purely additive, which was not the case as EDGAR emissions are estimated based on the sum of several product terms. To fix this underestimation IPCC (2006) advises using a correction factor. One example of a correction factor is proposed in Frey (2003), where the performance of an analytical approach for combining uncertainty in comparison to a Monte Carlo simulation with large sample sizes for many cases involving different ranges of uncertainty for additive, multiplicative, and quotient models are evaluated. Frey found that error propagation and
Monte Carlo simulated estimates of the uncertainty half-range of the model output agreed well for values of less than 100 %, but with the increase of the uncertainty a systematic underestimation of uncertainty in the total inventory by the error propagation approach appeared. The relationship between the simulated and propagated error estimates was found to be well-behaved, which led to a correction factor development for the large (i.e. greater than 100 %) total inventory uncertainties. This correction factor will not necessarily be reliable for very large uncertainties (i.e. greater than 230 %)
because it was calibrated over the range of 10 to 230 %. As such, the correction factor $FC$, calculated following Eq. (3), was applied if half-range uncertainty estimated from the error propagation method was > 100 and < 230 % following Eq. (4):

$$FC_{EDGARj} = \left[ \frac{-0.7200 + 1.0921 \cdot UC_{EDGARj} - 1.63 \cdot 10^{-3} \cdot UC_{EDGARj}^2 + 1.11 \cdot 10^{-5} \cdot UC_{EDGARj}^3}{UC_{EDGARj}} \right]^2, \tag{3}$$

$$\left( UC_{EDGARj} \right)_{corr} = UC_{EDGARj} \cdot FC_{EDGARj}, \tag{4}$$

where $corr$ corresponds to the corrected uncertainty; $UC_{EDGARj}$ is given in %. In cases where $UC_{EDGARj}$ was $\leq 100$ and $\geq 230$
%, $FC_{EDGARj}$ was assumed to be equal to one. Only four sectors with non energy use of fuels, chemical processes, fuel exploitation and coal production emissions were corrected, Table 3 shows how these uncertainties were corrected. It should be noted that some uncertainty ranges for emission factors and/or activity data in IPCC (2006) and IPCC-TFI (2019) are not symmetrical and have higher uncertainty values for the lower bound than for the upper bound, due to input from expert knowledge or available in-situ data, which lead to the same pattern in final prior uncertainty bounds.

**Table 3: Sectors with corrected uncertainties (lower and upper bounds) for countries with well- (WDS) and less well-developed (LDS) statistical infrastructures**

| № | ECMWF group | IPCC (2006) activities per EDGAR sector | Note | Country type | Prior uncertainty bounds, % | | | |
|---|---|---|---|---|---|---|---|---|
| | | | | | Before correction | | After correction | |
| | | | | | Low | Up | Low | Up |
| 3 | MANUFACTURING (part) | 2.D.1, 2.D.2, 2.D.4 | Non energy use of fuels | WDS | 112.0 | 112.0 | 121.7 | 121.7 |
| | | | | LDS | 113.8 | 113.8 | 124.0 | 124.0 |
| | | 2.B.1, 2.B.2, 2.B.3, 2.B.4, 2.B.5, 2.B.6, 2.B.8 | Chemical processes | WDS | 100.9 | 89.9 | 107.8 | 89.9 |
| | | | | LDS | 100.9 | 89.9 | 107.8 | 89.9 |
| 7 | OTHER (part) | 1.B.2.a.ii, 1.B.2.a.iii.2, 1.B.2.a.iii.3, 1.B.2.b.ii, 1.B.2.b.iii.2, 1.B.2.b.iii.4, 1.B.2.b.iii.5, 1.C | Fuel exploitation | WDS | 156.6 | 215.7 | 191.1 | 339.1 |
| | | | | LDS | 166.8 | 223.2 | 210.9 | 364.5 |
| | | 1.B.1.a | Coal production | WDS | 107.4 | 300.5 | 115.8 | 300.5 |
| | | | | LDS | 107.4 | 300.5 | 115.8 | 300.5 |

### 3.2.4 Forcing lognormal distribution on corrected EDGAR sector uncertainty

For models that are purely additive, and for which the half range of uncertainty is less than approximately 50 %, a normal distribution is often an accurate assumption for the model output form. In this case, a symmetric probability distribution with respect to the mean can be assumed. But this is not the case for multiplicative (or mixed) models, or when the uncertainty is large for a non-negative variable such as anthropogenic $CO_2$ emissions. A log-normal distribution is typically an accurate assumption for the model output form, where the uncertainty range is not symmetric with respect to the mean, even though

the variance for the total inventory may be correctly estimated from Approach 1. IPCC (2006) guidelines provide a practical methodology based on Frey (2003) for approximate asymmetric uncertainty range calculations based on the error propagation method. According to this methodology key characteristics of the 95 % confidence intervals are: (i) approximately symmetric for small ranges of uncertainty, and (ii) positively skewed for large ranges of uncertainty. This methodology was applied if the corrected lower half-range uncertainty estimated from error propagation method was ≥ 50 %.

IPCC (2006) suggests to define parameters of the lognormal distribution in terms of the geometric mean $\mu g$ (which can be estimated based upon the arithmetic mean and the arithmetic standard deviation) following Eq. (5) and geometric standard deviation $\sigma g$ following Eq. (6):

$$\mu g_{EDGARj} = exp\left\{ ln\left(E_{EDGARj}\right) - \frac{1}{2} \cdot ln\left(1 + \left[\frac{(UC_{EDGARj})_{corr}}{200}\right]^2\right)\right\}, \tag{5}$$

$$\sigma g_{EDGARj} = exp\left\{ \sqrt{ln\left(1 + \left[\frac{(UC_{EDGARj})_{corr}}{200}\right]^2\right)}\right\}, \tag{6}$$

where $E_{EDGARj}$ is the anthropogenic $CO_2$ emissions per sector $j$; $corr$ corresponds to the corrected uncertainty (i.e. corrected for the systematic underestimation of uncertainty calculated by the error propagation approach used in this study comparing to uncertainties calculated by using the Monte Carlo approach); $UC_{EDGARj}$ is in %.

Because calculations were performed for upper and lower uncertainty limits separately, there are two values of $\left(UC_{EDGARj}\right)_{corr}$ : $\left[\left(UC_{EDGARj}\right)_{corr}\right]_{low}$ – the absolute value of the lower uncertainty limit of sector $j$, and

$\left[\left(UC_{EDGARj}\right)_{corr}\right]_{high}$ – the absolute value of the upper uncertainty limit of sector $j$. As it is preferred to preserve as much accuracy (extra knowledge) as possible in our calculations and not to inflate uncertainty upper or lower bounds artificially, lower $\left\{\left[\left(UC_{EDGARj}\right)_{corr}\right]_{low}\right\}_{ln}$ and upper $\left\{\left[\left(UC_{EDGARj}\right)_{corr}\right]_{high}\right\}_{ln}$ uncertainty half-range from the error propagation method were calculated with a logarithmic transformation using $\left[\mu g_{EDGARj}\right]_{low}$ , $\left[\mu g_{EDGARj}\right]_{high}$ and $\left[\sigma g_{EDGARj}\right]_{low}$ , $\left[\sigma g_{EDGARj}\right]_{high}$ respectively according to the following Eq. (7) and Eq. (8) (see Figure 2 for visual representation of these

equations):

$$\left\{\left[\left(UC_{EDGARj}\right)_{corr}\right]_{low}\right\}_{ln} = \left(\frac{exp\{ln\left(\left[\mu g_{EDGARj}\right]_{low}\right)-1.96 \cdot ln\left(\left[\sigma g_{EDGARj}\right]_{low}\right)\}-E_{EDGARj}}{E_{EDGARj}}\right) \times 100, \tag{7}$$

$$\left\{\left[\left(UC_{EDGARj}\right)_{corr}\right]_{high}\right\}_{ln} = \left(\frac{exp\{ln\left(\left[\mu g_{EDGARj}\right]_{high}\right)+1.96 \cdot ln\left(\left[\sigma g_{EDGARj}\right]_{high}\right)\}-E_{EDGARj}}{E_{EDGARj}}\right) \times 100, \tag{8}$$

where $ln$ corresponds to logarithmic transformation of the distribution; resulting values are not absolute.

It should be noted that according to this methodology (with constants for 2.5th and 97.5th percentiles, +1.96 and -1.96

respectively, from the Z-table[2]) the lower uncertainty half-range $\left\{\left[\left(UC_{EDGARj}\right)_{corr}\right]_{low}\right\}_{ln}$ will always be less than 100.0 %.

Upper uncertainty half-range $\left\{\left[\left(UC_{EDGARj}\right)_{corr}\right]_{high}\right\}_{ln}$ is approximately symmetric relative to the 0 (Gaussian distribution) up to ~20.0 %, then has rather rapid growth till ~500.0 % (which with logarithmic transformation results in ~486.0 %), maxima at ~1350.0 % (which with logarithmic transformation results in ~582.6 %) and further gradual decrease.

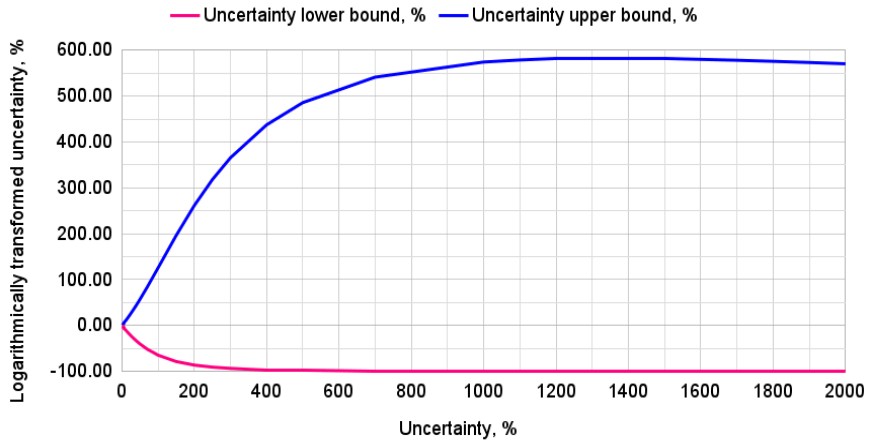


---

[2] The Z-table is a mathematical table for the values of the cumulative distribution function of the normal distribution.

**Figure 2: Visual representation of an empirical logarithmic transformation formula for upper and lower uncertainty bounds according IPCC (2006)**

Table 4 shows the prior uncertainty values for each EDGAR sector and for two geographical entity types (i.e. with well- and less well-developed statistical infrastructure). These values are a combined IPCC activity uncertainty aggregated to EDGAR sectors with the error propagation method and corrected for this method's underestimation. Also, as an example, Table 4 shows aggregated to ECMWF groups uncertainties with ensured log-normal distribution for China (CHN), Europe (28 members till end 2019) and all world countries.

**Table 4: Prior uncertainties (lower and upper bounds) per each EDGAR emission sector and two geographical entity types (with well- (WDS) and less well-developed (LDS) statistical infrastructure) based on IPCC (2006) and IPCC-TFI (2019), and aggregated to the ECMWF group uncertainties for China (CHN), Europe (E28) and globe (GLB)**

| № | ECMWF group | IPCC (2006) activities per EDGAR sector | Prior uncertainty bounds, % | | | | Uncertainty bounds, % | | | | | |
| | | | WDS countries | | LDS countries | | CHN, WDS | | E28, WDS | | GLB, mix | |
| | | | Low | Up | Low | Up | Low | Up | Low | Up | Low | Up |
|---|---|---|---|---|---|---|---|---|---|---|---|---|
| 1 | ENERGY_S | 1.A.1.a (subset) | 8.6 | 3.0 | 12.2 | 3.0 | 8.6 | 3.0 | 5.4 | 1.9 | 3.6 | 1.0 |
| 2 | ENERGY_A | 1.A.1.a (rest) | 8.6 | 8.6 | 12.2 | 12.2 | 8.6 | 8.6 | 2.8 | 2.8 | 3.5 | 3.5 |
| | | 4.C | 40.3 | 40.3 | 41.2 | 41.2 | | | | | | |
| 3 | MANUFACTURING | 1.A.2 | 8.6 | 8.6 | 12.2 | 12.2 | 12.8 | 19.4 | 3.9 | 5.8 | 5.7 | 8.6 |
| | | 2.C.1, 2.C.2 | 37.1 | 37.1 | 37.1 | 37.1 | | | | | | |
| | | 2.C.3, 2.C.4, 2.C.5, 2.C.6, 2.C.7 | 73.2 | 73.2 | 73.2 | 73.2 | | | | | | |
| | | 2.D.1, 2.D.2, 2.D.4 | 121.7 | 121.7 | 124.0 | 124.0 | | | | | | |
| | | 2.A.1, 2.A.2, 2.A.3, 2.A.4 | 70.9 | 70.9 | 93.0 | 93.0 | | | | | | |
| | | 2.B.1, 2.B.2, 2.B.3, 2.B.4, 2.B.5, 2.B.6, 2.B.8 | 107.8 | 89.9 | 107.8 | 89.9 | | | | | | |
| 4 | SETTLEMENTS | 1.A.4, 1.A.5.a, 1.A.5.b.i, 1.A.5.b.ii | 12.2 | 12.2 | 26.0 | 26.0 | 12.2 | 12.2 | 4.2 | 4.2 | 3.9 | 3.9 |
| 5 | AVIATION | 1.A.3.a_CRS | 5.5 | 6.4 | 50.1 | 106.8 | 3.5 | 4.1 | 1.4 | 1.6 | 17.3 | 58.1 |
| | | 1.A.3.a_CDS | 5.5 | 6.4 | 50.1 | 106.8 | | | | | | |
| | | 1.A.3.a_LTO | 5.5 | 6.4 | 50.1 | 106.8 | | | | | | |
| 6 | TRANSPORT | 1.A.3.b | 5.4 | 5.4 | 7.1 | 7.1 | 5.1 | 8.2 | 1.6 | 1.8 | 4.3 | 6.4 |
| | | 1.A.3.d | 5.4 | 5.1 | 50.0 | 50.0 | | | | | | |
| | | 1.A.3.c, 1.A.3.e | 50.3 | 106.9 | 50.5 | 107.0 | | | | | | |
| 7 | OTHER | 1.A.1.b, 1.A.1.c, 1.A.5.b.iii, 1.B.1.c, 1.B.2.a.iii.4, 1.B.2.a.iii.6, 1.B.2.b.iii.3 | 54.4 | 149.3 | 57.7 | 151.4 | 39.7 | 180.9 | 10.1 | 45.3 | 11.5 | 52.4 |
| | | 1.B.2.a.ii, 1.B.2.a.iii.2, 1.B.2.a.iii.3, 1.B.2.b.ii, 1.B.2.b.iii.2, 1.B.2.b.iii.4, 1.B.2.b.iii.5, 1.C | 191.1 | 339.1 | 210.9 | 364.5 | | | | | | |
| | | 1.B.1.a | 115.8 | 300.5 | 115.8 | 300.5 | | | | | | |
| | | 3.C.2, 3.C.3, 3.C.4, 3.C.7 | 70.7 | 0.0 | 70.7 | 0.0 | | | | | | |
| | | 2.D.3, 2.B.9, 2.E, 2.F, 2.G | 25.0 | 25.0 | 50.0 | 50.0 | | | | | | |

### 3.2.5 Calculating uncertainty for each ECMWF group

The next step is to combine these prior uncertainties for each EDGAR sector into ECMWF group uncertainties (see Table 4). Sector uncertainties are combined into group uncertainties by addition following Eq. (9) and Eq. (10):

$$UC_{ECMWFk} = \frac{\sqrt{(\{(UC_{EDGAR1})_{corr}\}_{ln}\cdot E_{EDGAR1})^2 + (\{(UC_{EDGAR2})_{corr}\}_{ln}\cdot E_{EDGAR2})^2 + \cdots + (\{(UC_{EDGARn})_{corr}\}_{ln}\cdot E_{EDGARn})^2}}{|E_{EDGAR1}+E_{EDGAR2}+\cdots+E_{EDGARn}|}, \tag{9}$$

$$E_{ECMWFk} = E_{EDGAR1} + E_{EDGAR2} + \cdots + E_{EDGARn}, \tag{10}$$

where $UC_{ECMWFk}$ and $E_{ECMWFk}$ – combined uncertainty and total emissions per group $k$; $1,2,...,n$ – EDGAR emission sectors that are combined in a particular ECMWF group $k$; $\{(UC_{EDGAR1})_{corr}\}_{ln}, \{(UC_{EDGAR2})_{corr}\}_{ln}, ..., \{(UC_{EDGARn})_{corr}\}_{ln}$ are in %. Combined group uncertainties are country-specific, because they take into account sector budget and adjust uncertainty values accordingly.

### 3.2.6 Calculating mean and standard deviation of lognormally distributed ECMWF group uncertainty

Finally, we needed to ensure a log-normal distribution of $CO_2$ emissions. Upper and lower uncertainty half-range values per ECMWF group $k$ $ECMWFk$ are descriptive, but not straight forward to use for emission perturbations in ensemble runs or flux inversions, where mean and standard deviation of the distribution are usually used. The lower and upper bounds of the 95 % probability range, which are the 2.5[th] and 97.5[th] percentiles respectively, calculated assuming a log-normal distribution based on a corrected estimated uncertainty half-range from an error propagation approach, are lower and upper uncertainty values. Taking this into account and using the Z-table for 2.5[th] and 97.5[th] percentiles $p$, mean $\mu^{ln}$ and standard deviation $\sigma^{ln}$ of log-normal distribution can be calculated following Eq. (11):

$$Z_p = \frac{ln([E_{ECMWFk}]_p) - \mu^{ln}_{ECMWFk}}{\sigma^{ln}_{ECMWFk}}, \tag{11}$$

where the following variables are known:

$$p = 2.5 \Rightarrow Z_{2.5} = -1.96, [E_{ECMWFk}]_{2.5} = E_{ECMWFk} \cdot \left(1 + \frac{[UC_{ECMWFk}]_{low}}{100}\right), \tag{12}$$

$$p = 97.5 \Rightarrow Z_{97.5} = 1.96, [E_{ECMWFk}]_{97.5} = E_{ECMWFk} \cdot \left(1 + \frac{[UC_{ECMWFk}]_{high}}{100}\right), \tag{13}$$

then simple system could be composed and solved accordingly following Eq. (14) and Eq. (15):

$$\mu^{ln}_{ECMWFk} = ln(E_{ECMWFk}) + \frac{1}{2}ln\left(1 + \frac{[UC_{ECMWFk}]_{low}}{100}\right) + \frac{1}{2}ln\left(1 + \frac{[UC_{ECMWFk}]_{high}}{100}\right), \tag{14}$$

$$\sigma^{ln}_{ECMWFk} = \frac{ln\left(1 + \frac{[UC_{ECMWFk}]_{low}}{100}\right) - ln\left(1 + \frac{[UC_{ECMWFk}]_{high}}{100}\right)}{-3.92}, \tag{15}$$

where $[UC_{ECMWFk}]_{low}$ and $[UC_{ECMWFk}]_{high}$ are in %.

### 3.2.7 Example of uncertainty calculation

Table 5 shows a step-by-step example of how yearly uncertainties are calculated. Example shows calculations for TRANSPORT group, that consists of several EDGAR emission sectors (one EDGAR sector consists even of several IPCC activities). Example shows two countries with different statistical infrastructure development levels (country with well-developed statistical infrastructures is Germany, country with less well-developed statistical infrastructures is the Russian Federation) and significant differences in emission budgets.


**Table 5a: Preparatory step for yearly uncertainty calculation – data collection, same values are applied for all countries of the same type, namely for countries with well- (WDS) and less well-developed (LDS) statistical infrastructures**

| Country (Type) | ECMWF group | IPCC (2006) activities per EDGAR sector | IPCC (2006) activity | Note | Typical fuel | Emission factor uncertainty | | Activity data uncertainty | |
|---|---|---|---|---|---|---|---|---|---|
| | | | | | | Low | Up | Low | Up |
| Germany (WDS) | TRANSPORT | 1.A.3.b | 1.A.3.b | Road transportation | most typical emission factor | 2.0 | 2.0 | 5.0 | 5.0 |
| | | 1.A.3.d | 1.A.3.d | Water-borne navigation | composition of 80 % diesel and 20 % residual fuel oil | 2.1 | 1.1 | 5.0 | 5.0 |
| | | 1.A.3.c, 1.A.3.e | 1.A.3.c | Railways | diesel | 2.0 | 0.9 | 5.0 | 5.0 |
| | | | 1.A.3.e | Other transportation – Pipeline | none (suggested to neglect) | 0.0 | 0.0 | 0.0 | 0.0 |
| | | | | Other transportation – Off-road | most typical emission factor | 2.0 | 2.0 | 50.0 | 100.0 |
| Russian Federation (LDS) | TRANSPORT | 1.A.3.b | 1.A.3.b | Road transportation | most typical emission factor | 5.0 | 5.0 | 5.0 | 5.0 |
| | | 1.A.3.d | 1.A.3.d | Water-borne navigation | composition of 80 % diesel and 20 % residual fuel oil | 2.1 | 1.1 | 50.0 | 50.0 |
| | | 1.A.3.c, 1.A.3.e | 1.A.3.c | Railways | diesel | 2.0 | 0.9 | 5.0 | 5.0 |
| | | | 1.A.3.e | Other transportation – Pipeline | none (suggested to neglect) | 0.0 | 0.0 | 0.0 | 0.0 |
| | | | | Other transportation – Off-road | most typical emission factor | 5.0 | 5.0 | 50.0 | 100.0 |

**Table 5b: First part of yearly uncertainty calculation – same values are applied for all countries of the same type, namely for countries with well- (WDS) and less well-developed (LDS) statistical infrastructures**

| Country (Type) | IPCC (2006) activities per EDGAR sector | IPCC (2006) activity | Combined uncertainty per IPCC (2006) activity, see Eq. (1) | | Combined uncertainty per EDGAR sector, see Eq. (2) | | Corrected combined uncertainty per EDGAR sector, see Eq. (3)-(4) | |
|---|---|---|---|---|---|---|---|---|
| | | | Low | Up | Low | Up | Low | Up |
| Germany (WDS) | 1.A.3.b | 1.A.3.b | 5.4 | 5.4 | 5.4 | 5.4 | 5.4 | 5.4 |
| | 1.A.3.d | 1.A.3.d | 5.4 | 5.1 | 5.4 | 5.1 | 5.4 | 5.1 |
| | 1.A.3.c, 1.A.3.e | 1.A.3.c | 5.4 | 5.1 | 50.3 | 100.1 | 50.3 | 106.9 |
| | | 1.A.3.e | 0.0 | 0.0 | | | | |
| | | | 50.0 | 100.0 | | | | |
| Russian Federation (LDS) | 1.A.3.b | 1.A.3.b | 7.1 | 7.1 | 7.1 | 7.1 | 7.1 | 7.1 |
| | 1.A.3.d | 1.A.3.d | 50.0 | 50.0 | 50.0 | 50.0 | 50.0 | 50.0 |
| | 1.A.3.c, 1.A.3.e | 1.A.3.c | 5.4 | 5.1 | 50.5 | 100.3 | 50.5 | 107.0 |
| | | 1.A.3.e | 0.0 | 0.0 | | | | |
| | | | 50.2 | 100.1 | | | | |

**Table 5c: Second part of yearly uncertainty calculation – values are specific per each geographical entity, take into account country type, namely if country has well- (WDS) or less well-developed (LDS) statistical infrastructure, and countries emission budget (values are from CHE_EDGAR-ECMWF_2015)**

| Country (Type) | IPCC (2006) activities per EDGAR sector | Emission budget 2015 per EDGAR sector, Mton | Uncertainty with assumed lognormal distribution per EDGAR sector, see Eq. (5)-(8) | | Emission budget 2015 per ECMWF group, Mton | Grouped uncertainty with assumed lognormal distribution per ECMWF group, see Eq. (9)-(10) | | Lognormal parameters of grouped uncertainty with assumed lognormal distribution per ECMWF group, see Eq. (14)-(15) | |
|---|---|---|---|---|---|---|---|---|---|
| | | | Low | Up | | Low | Up | mean | standard deviation |
| Germany (WDS) | 1.A.3.b | 139.6 | 5.4 | 5.4 | 143.0 | 5.3 | 5.7 | 11.9 | 0.0 |
| | 1.A.3.d | 1.0 | 5.4 | 5.1 | | | | | |
| | 1.A.3.c, 1.A.3.e | 2.3 | 40.3 | 135.5 | | | | | |
| Russian Federation (LDS) | 1.A.3.b | 131.7 | 7.1 | 7.1 | 206.9 | 14.1 | 44.8 | 12.3 | 0.1 |
| | 1.A.3.d | 7.4 | 40.1 | 57.2 | | | | | |
| | 1.A.3.c, 1.A.3.e | 67.9 | 40.5 | 135.7 | | | | | |


## 3.3 Monthly uncertainties

For Earth system modelling and data assimilation purposes a sub-yearly time scale is more appropriate. Monthly profiles of anthropogenic emissions are available and used in air quality models and are more certain than the sub-monthly profiles. The monthly profiles used in EDGARv4.3.2 are standardised to 12 monthly shares per EDGAR sector and per region (i.e.

Northern temperate zone, Equator, Southern temperate zone). They do not take into account the specificity of a single year and are not varying within a geographical entity (country). We used these global yearly and monthly emission maps for 2010 to calculate for each month a multiplication factor per 0.1°×0.1° grid-cell of the sector-specific maps. Then multiplication factors were combined with CHE_EDGAR-ECMWF_2015 maps and monthly country- and sector-specific $CO_2$ emission budgets are calculated.

Uncertainties for monthly budgets are obviously larger than yearly ones and instead of one standard deviation $\sigma$ (Quilcaille et al, 2018) two or three standard deviations, $2\sigma$ or $3\sigma$ respectively are commonly used (Oda et al., 2018; Andres et al., 2014; Andres et al., 2011). We decided to be more analytical:

1)      to use the same procedure as for annual uncertainty calculation but base it on monthly emission budgets (i.e. uncertainties for IPCC activities are combined to EDGAR sectors with error propagation method, corrected for systematic

underestimation by error propagation method, and adapted to have log-normal distribution; see Eq. (1)-(8)). Obtained monthly uncertainties are the same or even smaller than the yearly ones, because empirical equations applied use emission budgets, which are smaller for individual months compared to the yearly values;

2)      to calculate the correlation $\alpha$ (an uncertainty boosting parameter) between yearly and monthly uncertainties based on an analysis of the variations over the different months following Eq. (16):

$$(E_{YEAR} \cdot UC_{YEAR})^2 = \alpha^2 \cdot ((E_{MONTH1} \cdot UC_{MONTH1})^2 + (E_{MONTH2} \cdot UC_{MONTH2})^2 + \cdots + (E_{MONTH12} \cdot UC_{MONTH12})^2), \quad (16)$$

where *E* and *UC* correspond to sectoral emission budget and uncertainty in kton and % respectively, *YEAR,MONTH1,MONTH2,...,MONTH12* – yearly and monthly (January, February, …, December) values. Eq. (16) is based on the rule for combining uncorrelated uncertainties under addition of the error propagation equation (see Eq. (9)) and assumption that each month's uncertainty should be enhanced (boosted) by the same value;

3)    to multiply the prior yearly uncertainties from Table 4 by the boosting parameter (specific per country and emission sector) and use the result as a first guess of monthly prior uncertainties;

    4)    to iterate calculation steps 1) to 3) in order to find the best boosting parameter (to have the best fit between yearly and combined 12-month uncertainties) from Eq. (16) for each country and emission sector. Once the best boosting parameter was found (i.e. maximum difference between $\alpha$ from previous iteration and the current one over all countries and emission

sectors became less than acceptable threshold) calculated monthly uncertainties per each EDGAR sector were grouped into 7 ECMWF groups and log-normal distribution of $CO_2$ emissions was ensured, see Eq. (9)-(15).

Figure 3 has simplified roadmaps for yearly and monthly uncertainty calculations.

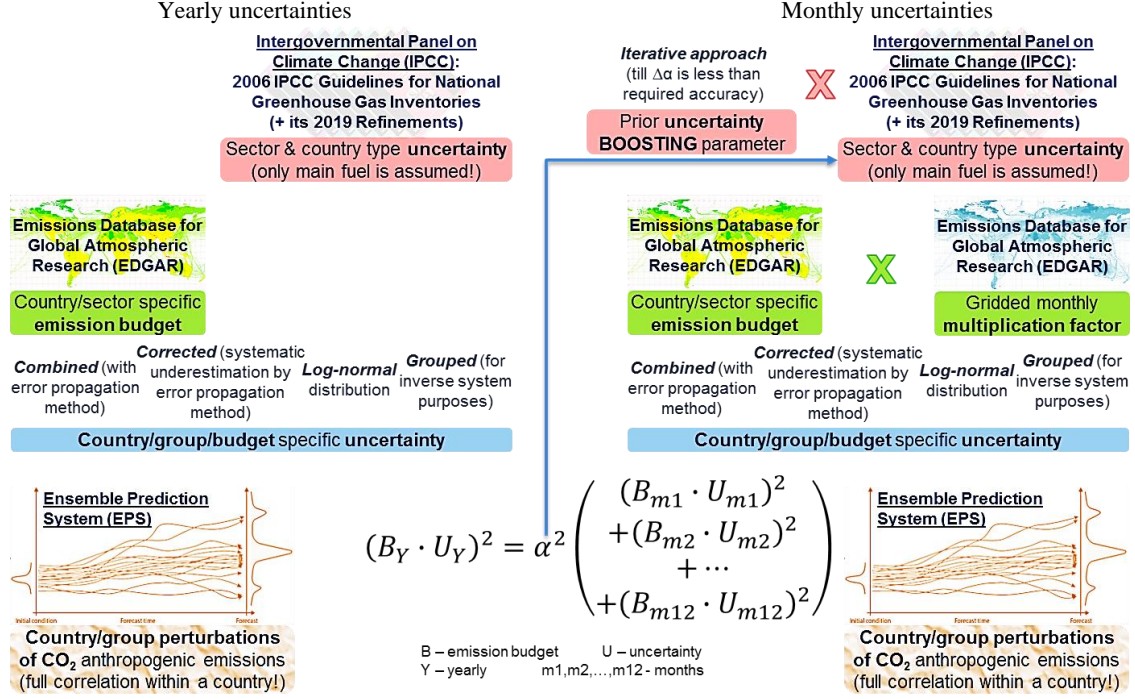

**Figure 3: Simplified roadmaps for yearly (left) and monthly (right) uncertainty calculation and their relation (bottom)**


The prior error covariance matrix of the emission inventory is required as an input to the inversion system. According to the IPCC (2006) all anthropogenic $CO_2$ emissions are assumed to be fully uncorrelated, hence the prior error correlations between grid-cell emissions from the same sector should be assumed negligible if country- and/or sector-specific

information is lacking. For the first implementation, ECMWF group covariance matrices per each geographical entity have the same representation – emission group is fully correlated with itself and fully uncorrelated with any other group. For an example of ECMWF group covariance matrices see Table S4 from the Supplementary Information, section S.3. Due to the lack of information available to properly characterize the error correlations and error variances in the inventory, a refinement of those prior statistics will be carried out in a follow-on paper (Bousserez, 2019) using atmospheric $CO_2$ observations. For this, the maximum likelihood of the prior error standard deviations and error correlation lengths will be estimated following approaches described in Wu et al. (2013).

### 3.4 Gridding uncertainties

Calculated yearly and monthly uncertainties per country and sector were assigned to each grid-box on the global map. National uncertainties were applied uniformly across each country. Figure 4 shows an example of the upper and lower uncertainty limits of anthropogenic $CO_2$ emission flux for TRANSPORT group. It should be noted that uncertainties related to the spatial distribution (representativeness of the proxy data and their uncertainty) should be much higher than the ones presented in this study. This research does not address uncertainties related to the spatial distribution. In the future we plan to address these uncertainties too. For example, following Oda et al. (2019) to characterize spatial patterns of the disaggregation errors in our emission maps.

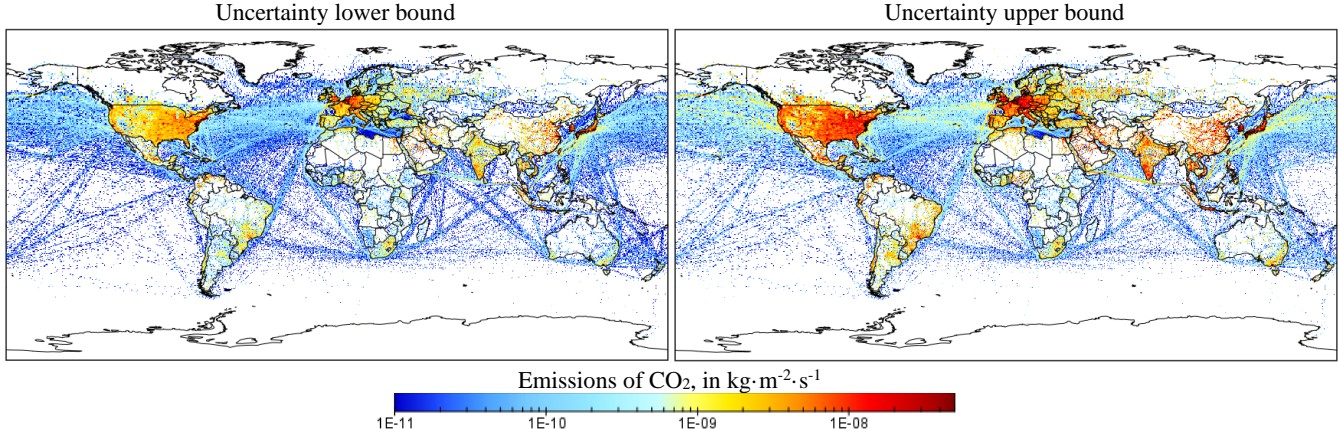

**Figure 4: $CO_2$ emission flux uncertainties (lower (left) and upper (right) bounds) for TRANSPORT group in $kg \cdot m^{-2} \cdot s^{-1}$**

## 4 Generated dataset

CHE_EDGAR-ECMWF_2015 data (Choulga et al., 2020) are freely available https://doi.org/10.5281/zenodo.3967439, and consist of 11 grid-maps in NetCDF format and one Excel file with information on anthropogenic $CO_2$ emissions and their uncertainties. For detailed information on each file see Table 6.

**Table 6: Detailed information on CHE_EDGAR-ECMWF_2015 data**

| File | General note | Field / Spreadsheet |
|---|---|---|
| Annual_Upper_Lower_Uncertainties_Percentage_0.1_0.1.nc | file has 2×8 fields with annual upper and lower uncertainty bounds in % per each emission group and for all groups summed together on a regular grid with 1800 pixels along the latitude and 3600 pixels along the longitude, where values represent centre of the grid-cell | "Lower" – lower uncertainty bound (2.5th percentile of log-normal distribution) for yearly emissions, in % |
| | | "Upper" – upper uncertainty bound (97.5th percentile of log-normal distribution) for yearly emissions, in % |
| | | "Sector" – emission sector numerical name. "0" represents emission group ENERGY_S (with IPCC (2006) activity 1.A.1.a (subset)) standing for power industry emissions from super emitting power plants; "1" group ENERGY_A (1.A.1.a (rest), 4.C) – power industry emissions from standard emitting power plants, & solid waste incineration; "2" group MANUFACTURING (1.A.2, 2.C.1, 2.C.2, 2.C.3, 2.C.4, 2.C.5, 2.C.6, 2.C.7, 2.D.1, 2.D.2, 2.D.4, 2.A.1, 2.A.2, 2.A.3, 2.A.4, 2.B.1, 2.B.2, 2.B.3, 2.B.4, 2.B.5, 2.B.6, 2.B.8) – combustion for manufacturing (including autoproducers), & iron and steel production, & non-ferrous metals production, & non energy use of fuels, & non-metallic minerals production, & chemical processes; "3" group SETTLEMENTS (1.A.4, 1.A.5.a, 1.A.5.b.i, 1.A.5.b.ii) – energy for buildings, residential heating; "4" group AVIATION (1.A.3.a_CRS, 1.A.3.a_CDS, 1.A.3.a_LTO) – aviation cruise, & climbing and descent, & landing and take off; "5" group TRANSPORT (1.A.3.b, 1.A.3.d, 1.A.3.c, 1.A.3.e) – road transportation, & shipping, & railways, pipelines, off-road transport; "6" group OTHER (1.A.1.b, 1.A.1.c, 1.A.5.b.iii, 1.B.1.c, 1.B.2.a.iii.4, 1.B.2.a.iii.6, 1.B.2.b.iii.3, 1.B.2.a.ii, 1.B.2.a.iii.2, 1.B.2.a.iii.3, 1.B.2.b.ii, 1.B.2.b.iii.2, 1.B.2.b.iii.4, 1.B.2.b.iii.5, 1.C, 1.B.1.a, 3.C.2, 3.C.3, 3.C.4, 3.C.7, 2.D.3, 2.B.9, 2.E, 2.F, 2.G) – oil refineries and transformation industry, & fuel exploitation, & coal production, & agricultural soils, & solvents and products use; "7" represents all groups summed together |
| Monthly_Upper_Lower_Uncertainties_Percentage_0.1_0.1.nc | file has 2×8×12 fields with monthly upper and lower uncertainty bounds in % per each emission group and for all groups summed together on a regular grid with 1800 pixels along the latitude and 3600 pixels along the longitude, where values represent centre of the grid-cell | file structure is identical to the file Annual_Upper_Lower_Uncertainties_Percentage_0.1_0.1.nc, but per month (1, 2, …, 12 correspond to January, February, …, December) |
| Annual_Upper_Lower_Uncertainties_0.1_0.1.nc | file has 3×8 fields with annual emissions, and upper and lower uncertainty bounds in kg·m$^{-2}$·s$^{-1}$ per each emission group and for all groups summed together on a regular grid with 1800 pixels along the latitude and 3600 pixels along the longitude, where values represent centre of the grid-cell | "Sup_lower" – lower uncertainty bound (2.5th percentile of log-normal distribution) for yearly emissions of ENERGY_S group, in kg·m$^{-2}$·s$^{-1}$, "Sup_upper" – upper uncertainty bound (97.5th percentile of log-normal distribution) for yearly emissions of ENERGY_S group, in kg·m$^{-2}$·s$^{-1}$, "Sup_flux" – yearly emissions of ENERGY_S group, in kg·m$^{-2}$·s$^{-1}$ |
| | | "Ene_lower", "ene_upper", "ene_flux" – same, but for ENERGY_A group, in kg·m$^{-2}$·s$^{-1}$ |
| | | "Man_lower", "man_upper", "man_flux" – same, but for MANUFACTURING group, in kg·m$^{-2}$·s$^{-1}$ |
| | | "Set_lower", "set_upper", "set_flux" – same, but for SETTLEMENTS group, in kg·m$^{-2}$·s$^{-1}$ |
| | | "Avi_lower", "avi_upper", "avi_flux" – same, but for AVIATION group, in kg·m$^{-2}$·s$^{-1}$ |
| | | "Tra_lower", "tra_upper", "tra_flux" – same, but for TRANSPORT group, in kg·m$^{-2}$·s$^{-1}$ |
| | | "Oth_lower", "oth_upper", "oth_flux" – same, but for OTHER group, in kg·m$^{-2}$·s$^{-1}$ |

| | | |
|---|---|---|
| | | "All_lower", "all_upper", "all_flux" – same, but for all groups summed together, in kg·m$^{-2}$·s$^{-1}$ |
| Monthly_Sup_Upper_Lower_Uncertainties_0.1_0.1.nc | file has 3×12 fields with monthly emissions, and upper and lower uncertainty bounds in kg·m$^{-2}$·s$^{-1}$ per ENERGY_S emission group on a regular grid with 1800 pixels along the latitude and 3600 pixels along the longitude, where values represent centre of the grid-cell | "Sup_lower" – lower uncertainty bound (2.5$^{th}$ percentile of log-normal distribution) for monthly emissions of ENERGY_S group, in kg·m$^{-2}$·s$^{-1}$ |
| | | "Sup_upper" – upper uncertainty bound (97.5$^{th}$ percentile of log-normal distribution) for monthly emissions of ENERGY_S group, in kg·m$^{-2}$·s$^{-1}$ |
| | | "Sup_flux" – monthly emissions of ENERGY_S group, in kg·m$^{-2}$·s$^{-1}$ |
| | | "Month" – month numerical name, where 1, 2, …, 12 correspond to January, February, …, December |
| Monthly_Ene_Upper_Lower_Uncertainties_0.1_0.1.nc | file has 3×12 fields with monthly emissions, and upper and lower uncertainty bounds in kg·m$^{-2}$·s$^{-1}$ per ENERGY_A emission group on a regular grid with 1800 pixels along the latitude and 3600 pixels along the longitude, where values represent centre of the grid-cell | file structure is identical to the file Monthly_Sup_Upper_Lower_Uncertainties_0.1_0.1.nc, but with "ene_lower", "ene_upper", "ene_flux" fields |
| Monthly_Man_Upper_Lower_Uncertainties_0.1_0.1.nc | file has 3×12 fields with monthly emissions, and upper and lower uncertainty bounds in kg·m$^{-2}$·s$^{-1}$ per MANUFACTURING emission group on a regular grid with 1800 pixels along the latitude and 3600 pixels along the longitude, where values represent centre of the grid-cell | file structure is identical to the file Monthly_Sup_Upper_Lower_Uncertainties_0.1_0.1.nc, but with "man_lower", "man_upper", "man_flux" fields |
| Monthly_Set_Upper_Lower_Uncertainties_0.1_0.1.nc | file has 3×12 fields with monthly emissions, and upper and lower uncertainty bounds in kg·m$^{-2}$·s$^{-1}$ per SETTLEMENTS emission group on a regular grid with 1800 pixels along the latitude and 3600 pixels along the longitude, where values represent centre of the grid-cell | file structure is identical to the file Monthly_Sup_Upper_Lower_Uncertainties_0.1_0.1.nc, but with "set_lower", "set_upper", "set_flux" fields |
| Monthly_Avi_Upper_Lower_Uncertainties_0.1_0.1.nc | file has 3×12 fields with monthly emissions, and upper and lower uncertainty bounds in kg·m$^{-2}$·s$^{-1}$ per AVIATION emission group on a regular grid with 1800 pixels along the latitude and 3600 pixels along the longitude, where values represent centre of the grid-cell | file structure is identical to the file Monthly_Sup_Upper_Lower_Uncertainties_0.1_0.1.nc, but with "avi_lower", "avi_upper", "avi_flux" fields |
| Monthly_Tra_Upper_Lower_Uncertainties_0.1_0.1.nc | file has 3×12 fields with monthly emissions, and upper and lower uncertainty bounds in kg·m$^{-2}$·s$^{-1}$ per TRANSPORT emission group on a regular grid with 1800 pixels along the latitude and 3600 pixels along the longitude, where values represent centre of the grid-cell | file structure is identical to the file Monthly_Sup_Upper_Lower_Uncertainties_0.1_0.1.nc, but with "tra_lower", "tra_upper", "tra_flux" fields |

| | | |
|---|---|---|
| Monthly_Oth_Upper_Lower_Uncertainties_0.1_0.1.nc | file has 3×12 fields with monthly emissions, and upper and lower uncertainty bounds in kg·m$^{-2}$·s$^{-1}$ per OTHER emission group on a regular grid with 1800 pixels along the latitude and 3600 pixels along the longitude, where values represent centre of the grid-cell | file structure is identical to the file Monthly_Sup_Upper_Lower_Uncertainties_0.1_0.1.nc, but with "oth_lower", "oth_upper", "oth_flux" fields |
| Monthly_All_Upper_Lower_Uncertainties_0.1_0.1.nc | file has 3×12 fields with monthly emissions, and upper and lower uncertainty bounds in kg·m$^{-2}$·s$^{-1}$ for all groups summed together on a regular grid with 1800 pixels along the latitude and 3600 pixels along the longitude, where values represent centre of the grid-cell | file structure is identical to the file Monthly_Sup_Upper_Lower_Uncertainties_0.1_0.1.nc, but with "all_lower", "all_upper", "all_flux" fields |
| CHE_EDGAR_2015.xlsx | file has 16 spreadsheets with listed information per country (metadata, emissions, uncertainties, statistical parameters) | "COUNTRY" – ISO Code (3-letter abbreviation of a geographical entity), Geographical name (name of a geographical entity), Type (development level of countries statistical infrastructure, meaning with well-/less well-developed statistical infrastructure), Main country (dependency, which country geographical entity in question belongs to), Full information (full name of a geographical entity, and what territory it occupies on the map of this study) |
| | | "GROUP" – № (number of anthropogenic $CO_2$ emission group), ECMWF group (group name), IPCC (2006) activity (IPCC activities that are included in each group), Note (short explanation of the group), Global emission budget 2015, Mton (total global emissions per group), Prior uncertainty bounds, % (initial, calculated purely based on assumptions from IPCC, lower and upper uncertainty bounds for countries with well-/less well-developed statistical infrastructures) |
| | | "YEARLY" – ISO Code (3-letter abbreviation of a geographical entity), ECMWF group (group name), Budget, kton (yearly anthropogenic $CO_2$ emission budget per group and total per geographical entity), Uncertainty bounds, % (calculated based on Prior uncertainty bounds and Budgets yearly uncertainties per group and total per geographical entity, uncertainties lower/upper/symmetrical bounds), Contribution to total countries uncertainty, % (share of each group in geographical entities total yearly uncertainty, total contribution is always 100 %), Parameters of log-normal distribution (anthropogenic $CO_2$ emission distribution is assumed to be log-normal, so additionally for modelling purposes log-normal mean, log-normal standard deviation and log-normal variance were calculated) |
| | | "MONTHLY_01", "MONTHLY_02", …, "MONTHLY_12" – same explanation as for spreadsheet "YEARLY", but for a month (01, 02, …, 12 correspond to January, February, …, December) |

## 5 Comparison and sensitivity

Calculated emissions and uncertainties of fossil $CO_2$ have been compared to other data sets based on the country-specific data reported to UNFCCC and on fuel-specific data reported in the energy statistics of IEA. The global values and their uncertainty at a $2\sigma$ range for the CHE_EDGAR-ECMWF_2015 dataset show the lowest value of -4.7/+9.6 % or ±7.1 % range, see Table 7. This result might be attributed to the methodology, in particular considering that (i) all calculations were done at the country level and then aggregated to global level assuming no correlation following IPCC (2006), (ii) all

calculations were done separately for upper and lower uncertainty bounds to preserve original information with asymmetric confidence intervals for large uncertainties (not required for the Approach 1 described in IPCC (2006), according to Approach 1 from IPCC (2006) only higher uncertainty value of asymmetric interval should be used – leads to artificial inflation of uncertainty upper or lower limit), and (iii) might be also because in this study we were not taking into account proxy grid-map uncertainties.


**Table 7: Comparison of global anthropogenic CO₂ emission uncertainty at 2σ associated with certain emission datasets**

| Name | Global uncertainty at 2σ, % | References |
|---|---|---|
| BP | no quantitative assessment of uncertainty associated with its emissions dataset | Andrew (2020) |
| CDIAC | ±8.4 % | Andres et al. (2016) |
| CEDS | no quantitative assessment of uncertainty associated with its emissions dataset, limited information in | Hoesly et al. (2018) |
| CHE_EDGAR-ECMWF_2015 | ±7.1 % (-4.7/+9.6 %) | Andrew (2020) |
| EDGAR | ±9.0 % | Janssens-Maenhout et al. (2019) |
| EIA | no quantitative assessment of uncertainty associated with its emissions dataset | Andrew (2020) |
| Global Carbon Project (GCP) | ±10.0 % | Friedlingstein et al. (2019) |
| IEA | no quantitative assessment of uncertainty associated with its emissions dataset | Andrew (2020) |
| ODIAC | ±8.4 %[3] | Oda et al. (2018) |

In this paper we decided to focus on some specific geographical areas – chosen to be among most emitting in total or per emission group, most typical or most influential for a certain region. A list of these geographical entities and development

levels of their statistical infrastructures are presented in Table 8.

**Table 8: List of selected geographical entities with their statistical infrastructure's development levels**

| ISO Code | Geographical name | Type |
|---|---|---|
| GLB | All World Countries | mixed-developed statistical infrastructure |
| E28 | Europe (28 members till end 2019) | well-developed statistical infrastructure |
| DEU | Germany | well-developed statistical infrastructure |
| ESP | Spain | well-developed statistical infrastructure |
| FRA | France | well-developed statistical infrastructure |
| GBR | United Kingdom | well-developed statistical infrastructure |
| POL | Poland | well-developed statistical infrastructure |
| BRA | Brazil | less well-developed statistical infrastructure |
| CHN | China | well-developed statistical infrastructure |
| IDN | Indonesia | less well-developed statistical infrastructure |
| IND | India | well-developed statistical infrastructure |
| JPN | Japan | well-developed statistical infrastructure |
| RUS | Russian Federation | less well-developed statistical infrastructure |
| USA | United States of America | well-developed statistical infrastructure |

---

[3] The difference between ODIAC and CDIAC gridded data is 3.3-5.7 % (Oda et al., 2018).

## 5.1 Global versus country-specific results

In order to see how development level of country's or geographical entity's statistical infrastructure is influencing emission uncertainty of that country or geographical entity itself and (possibly) the globe, uncertainty calculations for selected entities were performed twice – with their original and inverse types (i.e. country with well-developed statistical infrastructure becomes country with less well-developed statistical infrastructure and vice versa). More details on geographical entity's statistical infrastructure development level (e.g. how it was determined) are given in the Supplementary Information, section

S.4. Figure 5 shows sectoral emission budgets, uncertainties and contributions in percentage to the total uncertainty of country or geographical entity with its original and inverse statistical infrastructure development levels. The biggest impact of development level change can be noticed for countries with larger emission budgets. On average total uncertainties of selected countries (see Table 8) changed by 1-2 %; group uncertainties changed in line with prior uncertainties from Table 4 and countries emission budgets, as reported in Table 9.


**Table 9: Country's statistical infrastructure (countries with well- (WDS) and less well-developed (LDS) statistical infrastructures) influence on emission uncertainty**

| Impact on the uncertainty | ECMWF group | Cause description |
|---|---|---|
| most substantial | SETTLEMENTS | • consists only from residential heating emissions; <br> • high differences in prior uncertainties for WDS and LDS, ±12.2 % and ±26.0 % respectively |
| strong | MANUFACTURING | • budget usually makes a significant part of country's total emission budget; <br> • globally mainly composed from combustion for manufacturing with rather low prior uncertainty (±8.6 % and ±12.2 % for WDS and LDS respectively) and non-metallic minerals production with much higher uncertainties (±70.9 % and ±93.0 % for WDS and LDS respectively); <br> • also contains emissions from very uncertain non-energy use of fuels (±121.7 % and ±124.0 % for WDS and LDS respectively) and chemical processes (-107.8/+89.9 % both for WDS and LDS) emissions, though their global share in this group is only ~7.0 % |
| strong | ENERGY_A | • budget usually makes a significant part of country's total emission budget; <br> • composed of emissions from standard power plants with rather low uncertainties (±8.6 % and ±12.2 % for WDS and LDS respectively) and solid waste incineration with much higher uncertainties (±40.3 % and ±41.2 % for WDS and LDS respectively); <br> • for the Globe the ratio of solid waste incineration to energy emissions is ~1/100, which keeps the total group prior uncertainty quite low ±3.5 %; <br> • NB! geographical entities with higher ratios will have higher uncertainties |
| strong | ENERGY_S | • composed of emissions from super power plants only with rather low prior uncertainties (-8.6/+3.0 % and -12.2/+3.0 % for WDS and LDS respectively) for all geographical entities |
| mild | TRANSPORT | • globally mainly composed of road transportation with rather low uncertainty (±5.4 % and ±7.1 % for WDS and LDS respectively) and shipping emissions with low uncertainties - 5.4/+5.1 % for WDS and high uncertainties ±50.0 % for LDS; <br> • also contains rather uncertain railways, pipelines and off-road transport emissions (~ -50.4/+107.0 % for both WDS and LDS), though their global share in this group is ~16.0 % only; <br> • NB! all international shipping is included in All World Countries geographical entity |
| small | AVIATION | • extremely high differences in prior uncertainties for WDS and LDS (-5.5/+6.4 % and |

| | | |
|---|---|---|
| | | -50.1/+106.8 % respectively), though this groups share in global emissions is only 2.3 %; <br>•     NB! all international aviation is included in All World Countries geographical entity |
| negligible | OTHER | •     composed of very uncertain components with usually almost the same prior uncertainties for WDS and LDS; <br>•     main composite globally (~78.0 %) are emissions from oil refineries and transformation industry with prior uncertainties -54.4/+149.3 % and -57.7/+151.4 % for WDS and LDS respectively; <br>•     also usually has the highest contribution to the country's total uncertainty |

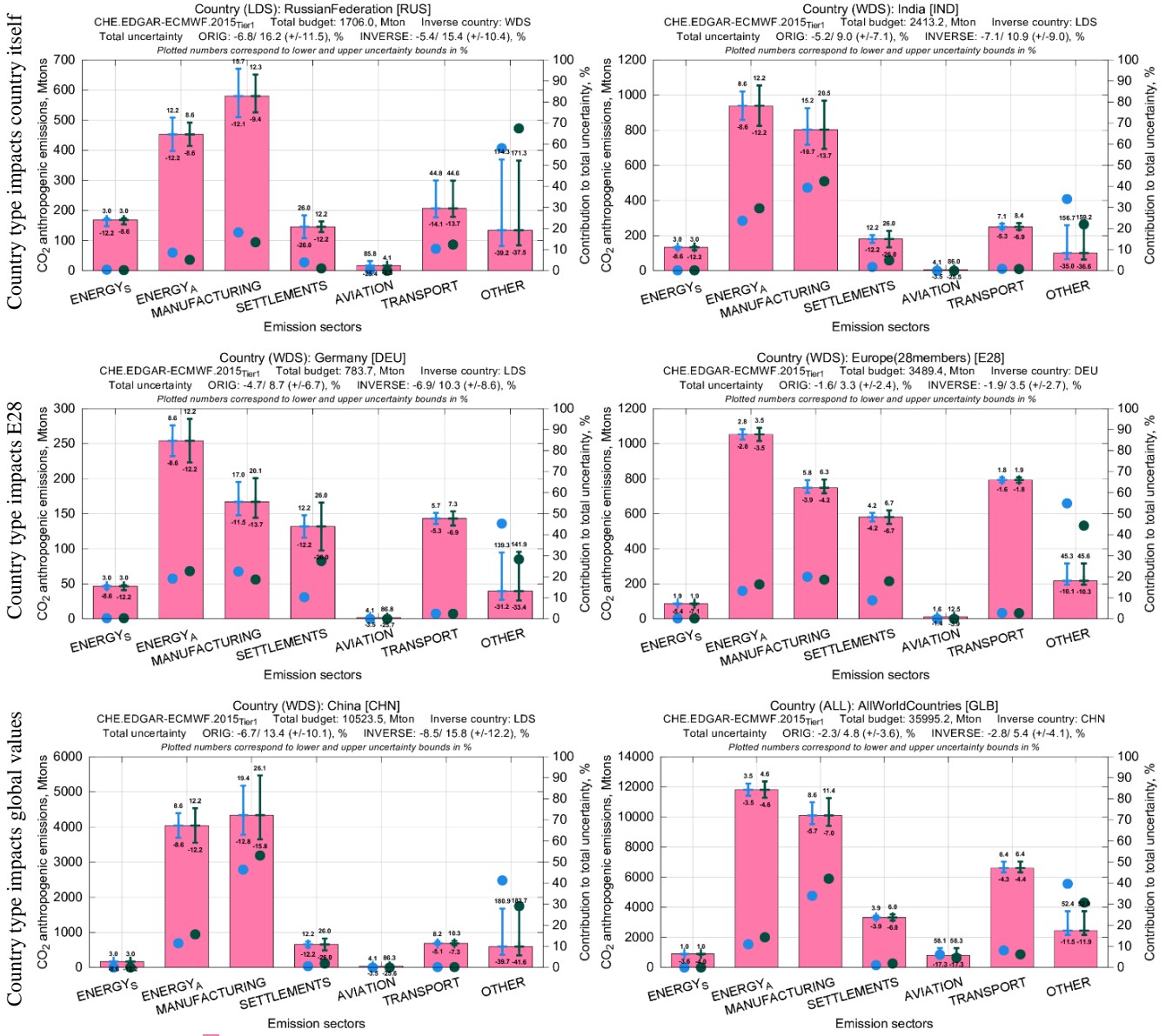

Group emission budget ▮, in Mtons    Upper and lower group uncertainty bound for    Group contribution to countries total uncertainty
Group uncertainty ⁴⁵·⁶ , in %    countries original ▌ and inverse ▌ type, in Mtons    for countries original ● and inverse ● type, in %

Alterations in some countries' (e.g. Germany, France) statistical infrastructure's development levels lead to changes in Europe (28 members till end 2019) uncertainties, with most substantial change for SETTLEMENTS group (e.g. 2.5 and 1.0 % respectively). Huge changes (> 10.0 %) in Europe's (28 members till end 2019) AVIATION group uncertainty % value can be due to the variation of statistical infrastructure development level for Germany, United Kingdom, France or Spain, though this groups contribution to the Europe's (28 members till end 2019) total uncertainty remains negligible. Alterations in statistical infrastructure development levels for China or the United States of America modify even global uncertainties because these countries substantially contribute to the global emission budget, e.g. China emits ~1/3 of the global anthropogenic $CO_2$ budget and can change global total uncertainty up to 0.5 %.

## 5.2 Yearly and monthly uncertainties

In order to increase the emission temporal resolution, monthly emissions and their uncertainties were calculated combining yearly emissions, monthly multiplication factors, and adapted uncertainty calculation methodology (see Section 3.3). Prior yearly uncertainties were multiplied by dimensionless uncertainty boosting parameter $\alpha$ (same value for each month) to compute prior monthly uncertainties, which were further used together with monthly emission budgets for countries monthly uncertainty calculation. Monthly uncertainties (just like yearly uncertainties) are determined by empirical formulas from IPCC (2006), hence their values depend on monthly emission budgets, which relate to number of days in a month (e.g. even with a flat yearly cycle months with more days have higher emission budgets, i.e. month emissions are sum of daily values). To eliminate this dependency, we looked straight away at dimensionless uncertainty boosting parameter $\alpha$, see Table 10 for most common values for countries with well- and less well-developed statistical infrastructures per EDGAR sectors. Boosting parameters become active ($\alpha \neq 1$) when absolute uncertainty values are $\geq 25.0$ %, $\alpha$ increases with the increase of absolute uncertainty following third order polynomial. For lower bound uncertainties $\alpha$ has bigger values and steeper growth than for upper bound uncertainties (e.g. -25.0 % $\triangleq \alpha = 1.5$ and -124.0 % $\triangleq \alpha = 2.6$; +25.0 % $\triangleq \alpha = 0.8$ and +124.0 % $\triangleq \alpha = 1.2$), $\alpha$ behaves in the same way for countries with well- and less well-developed statistical infrastructures. Discrepancies in different geographical entity's (country's) boosting parameters might be for several reasons, main ones are: (i) sector emissions were zero (e.g. super power plant emissions of the energy sector had no emissions); (ii) sector uncertainties were $\geq$ 50.0 % and needed to be adapted accordingly by log-normal distribution technique (e.g. agriculture soils sector with prior uncertainties -70.7/+0.0 % both for countries with well- and less well-developed statistical infrastructures). Most significant discrepancies in $\alpha$ are for agriculture soils sector (e.g. instead of lower/upper values from Table 10 for countries with well-developed statistical infrastructures France has $\alpha = 1.8/3.1$, United Kingdom – 1.8/7.2, China – 1.8/8.4, Japan – 1.8/10.8;

instead of lower/upper values from Table 10 for countries with less well-developed statistical infrastructures Brazil has $\alpha =$ 1.8/0.0, the Russian Federation – 1.8/5.6).

**Table 10: Dimensionless (DN) boosting parameter uncertainties (lower and upper bounds) for countries with well- (WDS) and less well-developed (LDS) statistical infrastructures**

| № | ECMWF group | IPCC (2006) activities per EDGAR sector | Uncertainty boosting parameter, DN | | | |
|---|---|---|---|---|---|---|
| | | | WDS countries | | LDS countries | |
| | | | Low | Up | Low | Up |
| 1 | ENERGY_S | 1.A.1.a (subset) | 1.0 | 1.0 | 1.0 | 1.0 |
| 2 | ENERGY_A | 1.A.1.a (rest) | 1.0 | 1.0 | 1.0 | 1.0 |
| | | 4.C | 1.8 | 0.8 | 1.9 | 0.8 |
| 3 | MANUFACTURING | 1.A.2 | 1.0 | 1.0 | 1.0 | 1.0 |
| | | 2.C.1, 2.C.2 | 1.7 | 0.8 | 1.7 | 0.8 |
| | | 2.C.3, 2.C.4, 2.C.5, 2.C.6, 2.C.7 | 2.0 | 0.9 | 2.0 | 0.9 |
| | | 2.D.1, 2.D.2, 2.D.4 | 2.6 | 1.2 | 2.6 | 1.2 |
| | | 2.A.1, 2.A.2, 2.A.3, 2.A.4 | 2.0 | 0.9 | 2.3 | 1.0 |
| | | 2.B.1, 2.B.2, 2.B.3, 2.B.4, 2.B.5, 2.B.6, 2.B.8 | 2.4 | 1.0 | 2.4 | 1.0 |
| 4 | SETTLEMENTS | 1.A.4, 1.A.5.a, 1.A.5.b.i, 1.A.5.b.ii | 1.0 | 1.0 | 1.5 | 0.9 |
| 5 | AVIATION | 1.A.3.a_CRS | 1.0 | 1.0 | 1.7 | 1.1 |
| | | 1.A.3.a_CDS | 1.0 | 1.0 | 1.7 | 1.1 |
| | | 1.A.3.a_LTO | 1.0 | 1.0 | 1.7 | 1.1 |
| 6 | TRANSPORT | 1.A.3.b | 1.0 | 1.0 | 1.0 | 1.0 |
| | | 1.A.3.d | 1.0 | 1.0 | 1.7 | 0.9 |
| | | 1.A.3.c, 1.A.3.e | 1.7 | 1.1 | 1.7 | 1.1 |
| 7 | OTHER | 1.A.1.b, 1.A.1.c, 1.A.5.b.iii, 1.B.1.c, 1.B.2.a.iii.4, 1.B.2.a.iii.6, 1.B.2.b.iii.3 | 1.7 | 1.4 | 1.8 | 1.4 |
| | | 1.B.2.a.ii, 1.B.2.a.iii.2, 1.B.2.a.iii.3, 1.B.2.b.ii, 1.B.2.b.iii.2, 1.B.2.b.iii.4, 1.B.2.b.iii.5, 1.C | 3.0 | 2.4 | 3.1 | 2.5 |
| | | 1.B.1.a | 2.5 | 2.2 | 2.5 | 2.2 |
| | | 3.C.2, 3.C.3, 3.C.4, 3.C.7 | 1.8 | 0.0 | 2.0 | 0.0 |
| | | 2.D.3, 2.B.9, 2.E, 2.F, 2.G | 1.5 | 0.8 | 1.7 | 0.9 |

In general, Brazil, Indonesia and India have a very weak yearly cycle with quite high monthly uncertainties throughout the year. Globe, Europe (28 members till end 2019), Germany, Spain, France, United Kingdom, Poland, China, Japan, the
Russian Federation, and the United States of America have more pronounced yearly cycles, most significant for SETTLEMENTS and ENERGY_A (and ENERGY_S where present) groups, and less significant for AVIATION, TRANSPORT and MANUFACTURING groups. This is in line with the monthly profiles applied in EDGARv4.3.2 for Northern and Southern temperate zones, and Equator, see Janssens-Maenhout et al. (2019). In summer months for Northern temperate zone, a strong decrease in SETTLEMENT and ENERGY_A (and ENERGY_S where present) groups emissions
was observed, a light decrease in MANUFACTURING group emissions, and a light increase in AVIATION and TRANSPORT groups emissions. This corresponds rather well with the assumption that most of the population in the Northern hemisphere must heat their houses during winter, and that they take holidays and travel more during summer.

## 5.3 Comparison with UNFCCC, TNO and other data

The CHE_EDGAR-ECMWF_2015 dataset containing 7 global gridded fossil $CO_2$ emission flux maps, and country- and
ECMWF-group-specific emission budgets and uncertainties have been assessed with independent data. Global emission
budget values from different datasets are almost never the same, therefore it is important to first identify why estimates differ
between datasets – datasets might use same country-level information as primary input, nevertheless differences in inclusion,
interpretation, and treatment of that data lead to diverse results in emissions; second – try to harmonise e.g. data inclusion or
omission across datasets to have more clarity in the discrepancies.

For Europe (28 members till end 2019), Germany, Spain, France, United Kingdom, Poland, Japan, the Russian Federation
and the United States of America emission and uncertainty data was collected from UNFCCC NIR. The aggregation of the
IPCC (2006) activity-specific emissions and uncertainties into 7 ECMWF groups was done assuming no correlation,
following IPCC (2006). Although IPCC (2006) has a standard table to report GHG emissions, uncertainties can be reported
in less detail by a more general category (e.g. 2.D only instead of 2.D.1, 2.D.2, 2.D.3, 2.D.4), meaning information
harmonization required lots of careful time-consuming country-specific technical work.

The Netherlands Organisation for Applied Scientific Research (TNO) has recently prepared the first version of their GHG
and co-emitted species emission database (TNO_GHGco_v1.1) that covers the entire European domain (at 0.1°×0.05°
resolution) also for $CO_2$ (distinguishing between fossil fuel and biofuel). Initial emission data is from the UNFCCC
(Common reporting format (CRF) tables) and the European Monitoring and Evaluation Programme/Centre on Emission
Inventories and Projections for air pollutants (EMEP/CEIP). These data were harmonized, checked for gaps, errors and
inconsistencies, and (where needed) replaced or completed using emission data from the Greenhouse gas-Air pollution
Interactions and Synergies (GAINS) model (Amann et al., 2011). Moreover, inland shipping emissions were replaced with
TNO's own estimates and sea shipping is based on automatic identification system (AIS) based tracks. Expert judgement is
used to assess the quality of each data source and to make choices on which source to use. The resulting emissions were
checked in detail with regard to their absolute value and trends (Kuenen et al., 2014). In this study we used emission budgets
from 30 TNO sectors provided by TNO (Super et al., February 2020, personal communication), and prior uncertainties
calculated from IPCC (2006) and IPCC-TFI (2019) see Table 11 (NB! all uncertainty calculations were done per country and
only then for comparison purposes aggregated to Europe (28 members till end 2019) values assuming no correlation
following IPCC (2006)). In addition, TNO has provided Tier 2 (Monte Carlo approach) uncertainties based on the same
budgets and uncertainties from submitted NIR reports based on Tier 1 approach. The Monte Carlo simulations were done at
the highest detail level (nomenclature for reporting (NFR) sector/fuel type) assuming correlations between certain sectors
(for more information see Super et al. (2020)), and then emissions were aggregated to ECMWF groups assuming no
correlation.

**Table 11: Prior uncertainties (lower and upper bounds) per each TNO emission sector based on IPCC (2006) and IPCC-TFI
(2019), and aggregated to the ECMWF group uncertainties for Germany (DEU) and Europe (E28)**

| № | ECMWF group | IPCC (2006) activities per TNO sector | Prior uncertainty bounds, % WDS countries | | Uncertainty bounds, % DEU | | E28 | |
|---|---|---|---|---|---|---|---|---|
| | | | Low | Up | Low | Up | Low | Up |
| 1 | ENERGY_S | 1.A.1.a (subset) | 8.6 | 3.0 | 0.0 | 0.0 | 0.0 | 0.0 |
| 2 | ENERGY_A | 1.A.1.a (rest) | 8.6 | 8.6 | 8.6 | 8.6 | 3.1 | 3.1 |
| | | 4.C | 40.3 | 40.3 | | | | |
| 3 | MANUFACTURING | 1.A.2 | 8.6 | 8.6 | 8.3 | 9.0 | 3.0 | 3.6 |
| | | 2.C.1, 2.C.2 | 37.1 | 37.1 | | | | |
| | | 2.C.3 | 10.2 | 10.2 | | | | |
| | | 2.C.4, 2.C.5, 2.C.6, 2.C.7 | 72.5 | 72.5 | | | | |
| | | 2.D.2 | 106.8 | 106.8 | | | | |
| | | 2.D.1, 2.D.4 | 50.3 | 50.3 | | | | |
| | | 2.A.1 | 36.7 | 36.7 | | | | |
| | | 2.A.2, 2.A.3, 2.A.4 | 60.7 | 60.7 | | | | |
| | | 2.B.1, 2.B.2, 2.B.3, 2.B.4, 2.B.5, 2.B.6, 2.B.8 | 107.8 | 89.9 | | | | |
| 4 | SETTLEMENTS | 1.A.4 | 12.2 | 12.2 | 12.1 | 12.1 | 4.2 | 4.2 |
| | | 1.A.5.a, 1.A.5.b.i, 1.A.5.b.ii | 0.0 | 0.0 | | | | |
| 5 | AVIATION | 1.A.3.a_CRS | 5.5 | 6.4 | 5.5 | 6.4 | 1.9 | 2.2 |
| | | 1.A.3.a_CDS | 5.5 | 6.4 | | | | |
| | | 1.A.3.a_LTO | 5.5 | 6.4 | | | | |
| 6 | TRANSPORT | 1.A.3.b | 5.4 | 5.4 | 5.4 | 7.4 | 1.8 | 3.1 |
| | | 1.A.3.d | 5.4 | 5.1 | | | | |
| | | 1.A.3.c | 5.4 | 5.1 | | | | |
| | | 1.A.3.e | 50.0 | 106.7 | | | | |
| 7 | OTHER | 1.A.1.b | 8.6 | 8.6 | 8.1 | 19.6 | 3.7 | 12.4 |
| | | 1.A.1.c | 12.2 | 12.2 | | | | |
| | | 1.A.5.b.iii, 1.B.1.c, 1.B.2.a.iii.4, 1.B.2.a.iii.6, 1.B.2.b.iii.3 | 0.0 | 0.0 | | | | |
| | | 1.B.2.a.ii, 1.B.2.a.iii.2, 1.B.2.a.iii.3, 1.B.2.b.ii, 1.B.2.b.iii.2, 1.B.2.b.iii.4, 1.B.2.b.iii.5 | 176.3 | 267.2 | | | | |
| | | 1.C | 50.0 | 100.0 | | | | |
| | | 1.B.1.a | 115.8 | 300.5 | | | | |
| | | 3.C.2 | 50.0 | 0.0 | | | | |
| | | 3.C.3, 3.C.4, 3.C.7 | 50.0 | 0.0 | | | | |
| | | 2.D.3, 2.B.9, 2.E, 2.F, 2.G | 25.0 | 25.0 | | | | |

Figure 6 shows emission budgets and uncertainties in Mtons, and contributions in % to the total geographical entity's uncertainty for Europe (28 members till end 2019), Germany, France and United Kingdom with their original statistical

infrastructure development types based on data from CHE_EDGAR-ECMWF_2015 (in pink), UNFCCC (in yellow), and TNO_GHGco_v1.1 Tier 1 (in blue) and Tier 2 (in green); plots for Spain and Poland are not shown here. Out of the four different sources, usually UNFCCC and TNO_GHGco_v1.1 Tier 2 uncertainties are the lowest ones and CHE_EDGAR-ECMWF_2015 – the highest one. It should be noted that: (i) UNFCCC uncertainties were aggregated to ECMWF groups individually per each country as uncertainties are reported in a rather free form thus could be aggregated from different

levels of precision, (ii) uncertainties for Europe (28 members till end 2019) from CHE_EDGAR-ECMWF_2015 are rather

low as they were calculated by aggregating information of 28 countries, rather than assuming it to be a one geographical entity from the beginning as it is done in UNFCCC, and (iii) differences in uncertainties of CHE_EDGAR-ECMWF_2015 with other sources, especially in fuel dependent emission groups, might be due to biofuels, as CHE_EDGAR-ECMWF_2015 is not taking them into account (NB! other datasets do take biofuels into account), and other sources (e.g. according to

UNFCCC SETTLEMENT group uncertainties for United Kingdom are ±24.5 % (contributes 95 % of United Kingdom's total uncertainty), which is twice higher according to other sources – it might be explained by use of other fuels, e.g. wood and/or coal for residential heating). Differences in uncertainties between CHE_EDGAR-ECMWF_2015 and TNO_GHGco_v1.1 Tier 1 show additional value in more detailed emission budget knowledge, i.e. if we know for certain that country has no glass production then this rather uncertain activity can be excluded from non-metallic minerals

production sector overall uncertainty calculation. Differences in uncertainties between TNO_GHGco_v1.1 Tier 1 and TNO_GHGco_v1.1 Tier 2 show additional value in advanced calculation technique, using a more sophisticated, data demanding Monte Carlo approach instead of simple error propagation. Overall there is quite good agreement in emission budgets and uncertainties from different sources of emission data.

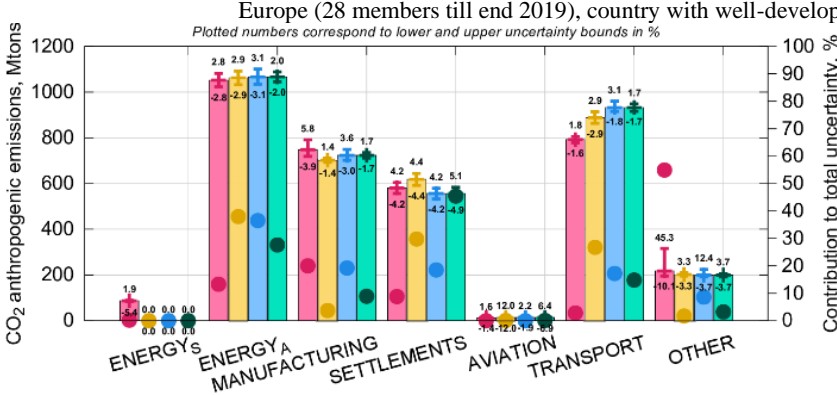

Europe (28 members till end 2019), country with well-developed statistical infrastructure

| Data | Emission budget 2015, Mton | Uncertainty bound, % | |
|---|---|---|---|
| | | Low/Up | Range |
| CHE_EDGAR-ECMWF_2015 | 3489.4 | -1.6/+3.3 | ±2.4 |
| UNFCCC$_{Tier1}$ | 3486.7 | -1.4/+1.4 | ±1.4 |
| TNO_GHGco_v1.1$_{Tier1}$ | 3492.2 | -1.4/+1.8 | ±1.6 |
| TNO_GHGco_v1.1$_{Tier2}$ | 3492.3 | -1.2/+1.2 | ±1.2 |

Germany, country with well-developed statistical infrastructure

| Data | Emission budget 2015, Mton | Uncertainty bound, % | |
|---|---|---|---|
| | | Low/Up | Range |
| CHE_EDGAR-ECMWF_2015 | 783.7 | -4.7/+8.7 | ±6.7 |
| UNFCCC$_{Tier1}$ | 794.0 | -2.7/+2.7 | ±2.7 |
| TNO_GHGco_v1.1$_{Tier1}$ | 791.2 | -4.3/+4.6 | ±4.5 |
| TNO_GHGco_v1.1$_{Tier2}$ | 791.2 | -3.8/+3.6 | ±3.7 |

France, country with well-developed statistical infrastructure

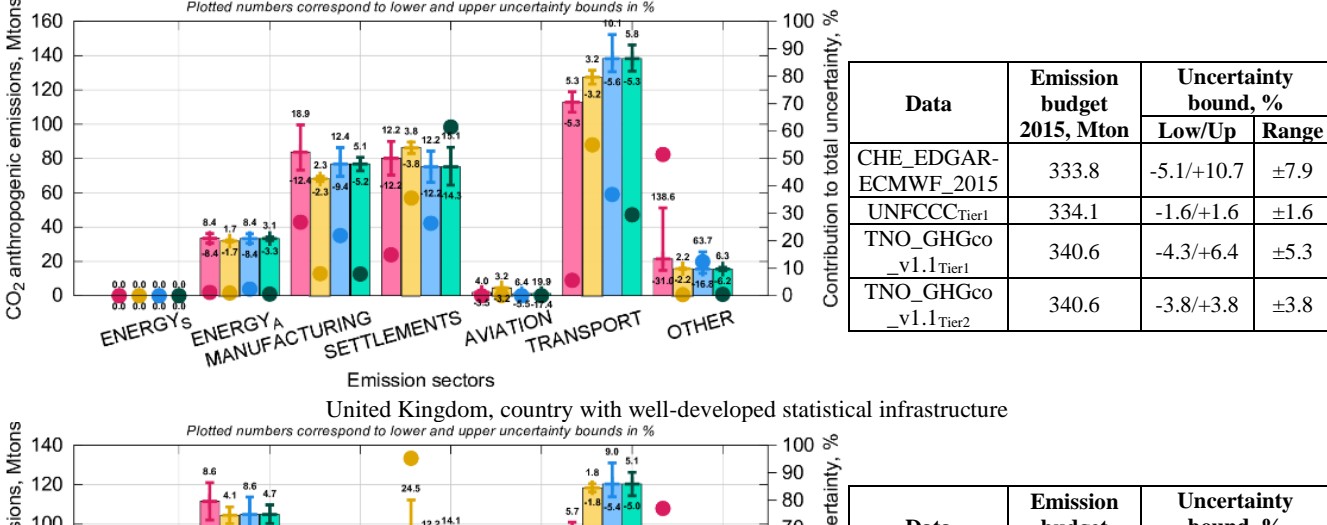

United Kingdom, country with well-developed statistical infrastructure

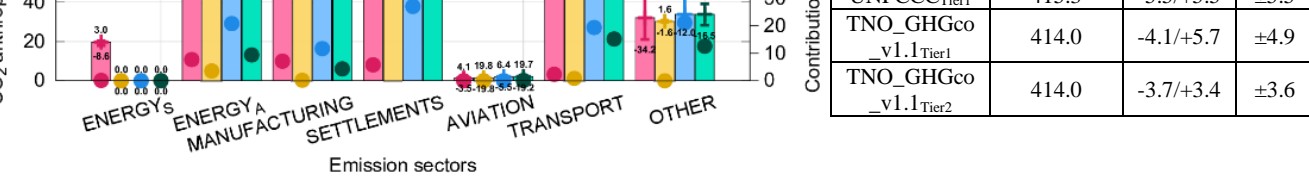

Figure 6: Emission budgets, uncertainties and contributions in percentage to the total uncertainty for Europe (E28), Germany (DEU), France (FRA) and United Kingdom (GBR) with their original statistical infrastructure development types

Emission budgets, Tier 1 uncertainties, and contributions in percentage to the total geographical entity's uncertainty for Japan, the Russian Federation and the United States of America from CHE_EDGAR-ECMWF_2015 could be compared only with UNFCCC data (plots not shown here). UNFCCC uncertainties are usually lower than the ones calculated in this study. Main reason for that is use of country-specific emission data and activity data uncertainties, which are lower than default values suggested by IPCC (2006) and IPCC-TFI (2019). Only for fuel dependent groups (e.g. AVIATION) UNFCCC uncertainties might be higher than in this study as rather uncertain biofuels might be taken into account (NB! CHE_EDGAR-ECMWF_2015 does not take biofuels into account). Also, emission budgets reported to UNFCCC show some differences from the ones from CHE_EDGAR-ECMWF_2015. For Japan group budgets agree rather well, and total budget difference is ~1.0 %. For the Russian Federation major differences are in ENERGY_A (and ENERGY_S) and MANUFACTURING groups, which results in ~6.0 % higher total budget of CHE_EDGAR-ECMWF_2015. For the United States of America major differences are ~200 Mton and ~100 Mton for SETTLEMENTS and OTHER groups respectively, which results in

~4.0 % higher total budget than based on UNFCCC data. Recent comparison of different gridded global datasets by Andrew

(2020) pointed out that only few of these datasets provide quantitative uncertainty assessment, see summary in Table 7. Comparing to other global emission uncertainty values CHE_EDGAR-ECMWF_2015 shows lowest values mainly due to the aggregation technique.

## 5.4 Sensitivity to the fuel specificity

As mentioned above, for transport related emission uncertainty calculations only the most typical fuel type (for aviation,

railways, shipping) and emission factor uncertainty (for road and off-road transport) were used, because detailed fuel consumption information per IPCC activity was not available for this study. EDGAR dataset development team do have specific fuel information globally, which could be used for uncertainty calculation. EDGAR dataset with incorporated fuel-specific activity data and emission factor uncertainties and Tier 1 approach for uncertainty calculation (see Supplementary Information, section S.5) hereinafter referred to as EDGAR-JRC. Country budget uncertainties were calculated by

considering "full fuel" splitting and by taking into consideration the assumption that emission factor from sectors sharing the same fuel are fully correlated. This latter assumption transformed the sum in quadrature of Eq. (2) into a linear summation (Bond et al., 2004; Bergamaschi et al., 2015). The uncertainty of activity data were set in accordance with IPCC (2006) guidelines, in the range 5.0 to 10.0 % for combustion activities, 10.0 to 20.0 % for combustion in the residential sector, 25.0 % for bunker fuels in the marine transport, 35.0 % for industrial processes of cement, lime, glass, ammonia (the range of

uncertainty values refers to the 95 % confidence interval of the mean, assigned separately to countries with well- and less well-developed statistical infrastructures). Uncertainties from EDGAR-JRC dataset aggregated to the ECMWF group level were compared with the ones from CHE_EDGAR-ECMWF_2015, see Table 12 for Europe (28 members till end 2019) and all world countries, and Table S8 from the Supplementary Information, section S.5, for all the rest geographical entities from Table 8. NB! Group contribution to the geographical entity's (country's) total uncertainty is zero when group has no

emissions. Emission uncertainties from EDGAR-JRC reflect the share of fuel composing the emission of each country and are in line with the estimates by CHE_EDGAR-ECMWF_2015 for those countries where the fuel-composite uncertainty is closer to the average value assigned (see Table 4). Uncertainties calculated with fuel-specific data are usually smaller; when prevailing fuel coincides with typical fuel type from CHE_EDGAR-ECMWF_2015 emission group uncertainties from both sources are quite similar. It should be noted here that: (i) countries total uncertainty is higher in EDGAR-JRC due to

aggregation technique (full correlation is assumed), (ii) AVIATION group uncertainties are higher in EDGAR-JRC due to prior aggregation of all three aviation connected sectors (cruise, climbing & descent, and landing & take off).

**Table 12: Aggregated to the ECMWF group level uncertainties (lower and upper bounds) in % and contributions in % to the total uncertainty (CV) for Europe (E28) and globe (GLB) from EDGAR-JRC (with extra fuel type knowledge) and CHE_EDGAR-**
**ECMWF_2015 (with typical fuel only)**

| Country | ECMWF group | EDGAR-JRC | | | CHE_EDGAR-ECMWF_2015 | | |
|---------|-------------|-----------|------|------|------|------|------|
| | | Low, % | Up, % | CV, % | Low, % | Up, % | CV, % |

| | | | | | | | |
|---|---|---|---|---|---|---|---|
| GLB | ENERGY_S | 0.0 | 0.0 | 0.0 | -3.6 | 1.0 | 0.0 |
| | ENERGY_A | -2.9 | 2.7 | 42.4 | -3.5 | 3.5 | 11.0 |
| | MANUFACTURING | -4.3 | 4.3 | 41.3 | -5.7 | 8.6 | 34.0 |
| | SETTLEMENTS | -2.5 | 2.5 | 1.9 | -3.9 | 3.9 | 1.1 |
| | AVIATION | -4.2 | 5.8 | 0.5 | -17.3 | 58.1 | 6.1 |
| | TRANSPORT | -2.5 | 2.6 | 7.7 | -4.3 | 6.4 | 8.1 |
| | OTHER | -5.9 | 6.2 | 6.2 | -11.5 | 52.4 | 39.7 |
| | *TOTAL* | *-4.8* | *4.8* | *100.0* | *-2.3* | *4.8* | *100.0* |
| E28 | ENERGY_S | 0.0 | 0.0 | 0.0 | -5.4 | 1.9 | 0.2 |
| | ENERGY_A | -2.0 | 2.4 | 56.4 | -2.8 | 2.8 | 13.3 |
| | MANUFACTURING | -2.2 | 2.2 | 12.6 | -3.9 | 5.8 | 20.0 |
| | SETTLEMENTS | -2.5 | 2.5 | 15.1 | -4.2 | 4.2 | 8.8 |
| | AVIATION | -2.4 | 2.8 | 0.0 | -1.4 | 1.6 | 0.0 |
| | TRANSPORT | -1.3 | 1.3 | 7.2 | -1.6 | 1.8 | 2.8 |
| | OTHER | -5.0 | 5.0 | 8.7 | -10.1 | 45.3 | 54.9 |
| | *TOTAL* | *-3.3* | *3.6* | *100.0* | *-1.6* | *3.3* | *100.0* |

The uncertainties derived in this study are an upper bound of the uncertainty estimation compared to the uncertainties calculated with more detailed information, as done by the countries and reported to UNFCCC or to the uncertainties calculated with fuel-specific data. Even though sometimes differences might be quite high in %, they are usually quite small in Mtons. Taking into account that fuel data is not publicly available, requires a lot of time to collect and implement, and is not available globally – it was decided not to use it in this study for Tier 1 uncertainty calculations.

## 5.5 Atmospheric sensitivity to nationally disaggregated emissions

The gridded emissions are a required input to the ECMWF model used to simulate atmospheric $CO_2$ globally (Agusti-Panareda et al., 2014; Agusti-Panareda et al., 2019). Ideally, uncertainties at a grid-cell level would be preferred by the models, which is a difficult time-consuming task. In order to check if these calculations are necessary it was decided to run some experiments. High-resolution (~25 km horizontal resolution, 137 vertical levels) simulations with ECMWF Integrated Forecasting System (IFS) model have been performed to assess the atmospheric sensitivity to fully resolved emissions compared to nationally smoothed (global emission budget is conserved), see Figure 7.

Fully resolved emission source distribution

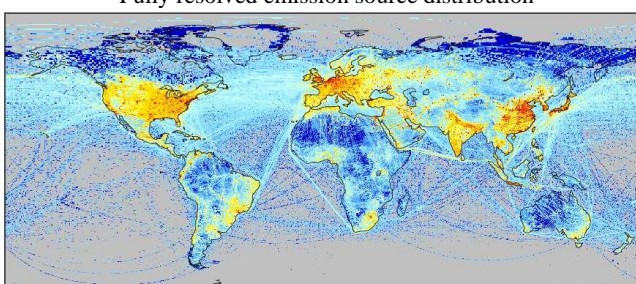

Country aggregated emission source distribution

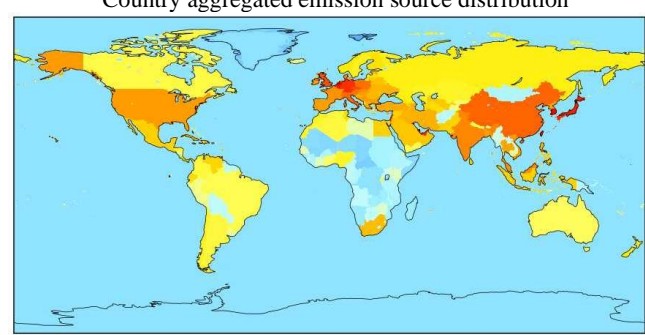

Emissions of CO₂, in kg·m⁻²·s⁻¹

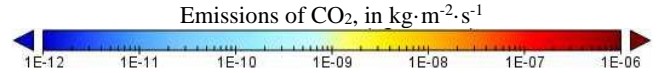

1E-12    1E-11    1E-10    1E-09    1E-08    1E-07    1E-06

**Figure 7: Anthropogenic CO₂ flux source distribution – fully resolved (left), country aggregated (right)**

Model simulations were performed for January 2015 with 3 hourly output. Anthropogenic, fire, ocean and biogenic fluxes (large-scale model BIAS mitigated by biogenic CO₂ flux adjustment scheme BFAS) were considered. For the full model

configuration description see McNorton et al. (2020). The atmospheric response to using either fully or partially resolved emissions compared with nationally smoothed emissions after a 10-day period are shown in Figure 8. It was noted that point sources (e.g. power plants, factories) can be easily detected if they comprise substantial part of countries total emission budget (e.g. in South Africa). If point sources are distributed homogeneously over the country and other areal sources are rather high as well it becomes really difficult to detect one extra/missing emitting hotspot (e.g. in Germany). China is a very

good example for both cases as its western part has very little hotspots and they are easy to detect over the low emitting background, and its eastern part has lots of hotspots and high emitting areal sources which make it almost impossible to disentangle emissions from single power plant or factory from high emitting background. In general, even by resolving a single sector, in this case the energy sector (see Figure 8), a difference in the atmospheric response is evident. Differences of several ppm are detected over multiple regions, highlighting the importance of using high resolution spatially resolved

emissions. With increase of both flux and transport model resolutions these differences are expected to increase further with steeper atmospheric CO₂ gradients.

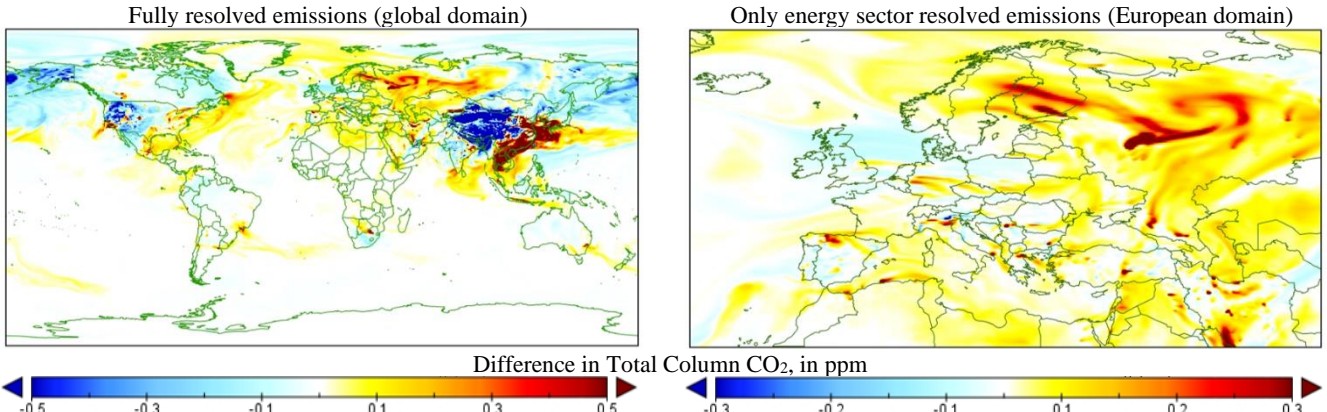

Fully resolved emissions (global domain)           Only energy sector resolved emissions (European domain)

Difference in Total Column CO₂, in ppm

-0.5   -0.3   -0.1   0.1   0.3   0.5      -0.3   -0.2   -0.1   0.1   0.2   0.3

**Figure 8: Difference in atmospheric response to using resolved and country aggregated emissions for January 2015 with IFS model at ~25 km resolution after 10-day simulation; the difference is calculated using both fully resolved emissions (left) or by only**

**resolving the energy sector emissions (right)**

**6 Conclusions and discussion**

The new CHE_EDGAR-ECMWF_2015 dataset with anthropogenic fossil $CO_2$ emissions and their uncertainties and with a new 7×7 covariance matrix for the atmospheric transport model was compiled and tested. The fossil $CO_2$ emissions include
all long-cycle carbon emissions from human activities, such as fossil fuel combustion, industrial processes (e.g. cement) and products use, but excludes emissions from land-use change and forestry. Human $CO_2$ emission inventories were processed into gridded maps to provide an estimate of prior $CO_2$ emissions, aggregated in 7 main emissions groups: 1) energy production super-emitters, 2) energy production standard-emitters, 3) manufacturing, 4) settlements, 5) aviation, 6) other transport at ground level and 7) others, with estimation of their uncertainty and covariance. For the first implementation it is
assumed that each emission group is fully correlated with itself and fully uncorrelated with any other group (only diagonal values are non-zero and equal to log-normal variance).

The CHE_EDGAR-ECMWF_2015 represents the 2015 global fossil $CO_2$ emissions prior at 0.1°×0.1° resolution that has been for the first time to our knowledge bridging the inventory community and the atmospheric modelling community. In fact, the uncertainty calculations fully respect the detailed error propagation approach recommended by IPCC (2006)
guidelines for GHG inventories while these datasets as prior input were processed such that the uncertainty information could be fully taken up by the ECMWF model IFS. Estimation of emission uncertainties is purely based on IPCC (2006) and IPCC-TFI (2019) emission factor and activity data uncertainty values and assumptions – mainly that emissions are fully uncorrelated. Uncertainties related to the spatial distribution (representativeness of the proxy data and their uncertainty) were not assessed in this study, but they can be included by the user on top of the calculated emission uncertainties. All
calculations, performed for the year 2015, are documented so that the methodology and algorithms used can be easily adapted for any other year. The dataset can be directly used in inverse modelling, and ensemble data assimilation applications, such as those envisaged within the Copernicus Atmosphere Monitoring Service (CAMS) system.

CHE_EDGAR-ECMWF_2015 consists of 11 global NetCDF files with gridded yearly and monthly upper and lower bounds of uncertainties in % and kg·m$^{-2}$·s$^{-1}$ per each ECMWF group and their sum, and 1 Excel file with 16 spreadsheets with the
same information listed per country (metadata, emissions, uncertainties, statistical parameters).

Calculated emissions and uncertainties of fossil $CO_2$ have been compared to other data sets based on the country-specific data reported to UNFCCC and on fuel-specific data reported in the energy statistics of IEA. The global values and their uncertainty at a 2$\sigma$ range for the CHE_EDGAR-ECMWF_2015 dataset show the lowest value of -4.7/+9.6 % or ±7.1 % range due to the methodology used. At country level the CHE_EDGAR-ECMWF_2015 dataset provides generally larger
uncertainty ranges, that are reduced when more detailed information is available to reduce the uncertainties; in summary, using the information that is uniformly available for all countries a coherent uncertainty representation is obtained.

The CHE_EDGAR-ECMWF_2015 dataset has been tested to provide the ECMWF Earth system ensemble spread to characterise the $CO_2$ atmospheric concentrations' uncertainties in the prototype of the Copernicus $CO_2$ Monitoring and Verification Support Capacity. Annual and monthly uncertainties have been evaluated in the ECMWF's atmospheric

transport model IFS ensemble simulations as well as the sensitivity to the spatial distribution of anthropogenic $CO_2$ emissions (McNorton et al., 2020). Results show to be rather sensitive to the spatial distribution proxies, and most updated proxies and prior uncertainties are better adapted for data assimilation applications. This needs to be studied in a future research project, the Prototype system for a Copernicus $CO_2$ service (CoCO2), that follows the current CHE research project. Contribution of representativeness errors to uncertainties and time correlation are neglected in CHE_EDGAR-

ECMWF_2015 and will need to be assessed in successive future studies. The estimation of global gridded emissions with their spatially and temporally distributed uncertainties constitute the backbone for atmospheric inversions to estimate anthropogenic emissions from atmospheric concentrations (Pinty et al., 2017). Dedicated satellite missions (e.g. Copernicus anthropogenic $CO_2$ monitoring mission CO2M described in Janssens-Maenhout et al. (2020)) are being planned to monitor anthropogenic emissions from space and substantially reduce emission uncertainties. The developments in the emission

uncertainty based on prior knowledge computation presented in this paper is an important preparatory step for an ensemble-based $CO_2$ Monitoring and Verification System prototype, such as the one developed within the CHE project.

*Data availability.* EDGARv4.3.2 data are open access and available at http://edgar.jrc.ec.europa.eu/overview.php?v=432&SECURE=123, last access: 29 June 2020,

doi:https://data.europa.eu/doi/10.2904/JRC_DATASET_EDGAR, documented in Janssens-Maenhout et al. (2019). CHE_EDGAR-ECMWF_2015 data (Choulga et al., 2020) are freely available https://doi.org/10.5281/zenodo.3967439, and documented in this paper.

*Author contribution.* All the authors participated in the EDGAR_CHE maps generation (methodology, data generation),

model experiment set-up, and analysis of the result. Margarita Choulga and Greet Janssens-Maenhout wrote the manuscript with contributions from all the other authors.

*Competing interests.* The authors declare that they have no conflict of interest.

*Acknowledgements.* The authors thank Glenn Carver (ECMWF) for editorial help and assistance; Anabel Bowen (ECMWF) for invaluable help with figure design. Margarita Choulga was funded by the $CO_2$ Human Emissions (CHE) project which received funding from the European Union's Horizon 2020 research and innovation programme under grant agreement no. 776186.

*Financial support.* This research has been supported by CHE (grant no. 776186).

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
