# Peer review of "Global anthropogenic CO2 emissions and uncertainties as prior for Earth system modelling and data assimilation"

_Earth System Science Data, 2020_

## Referee Comment (RC1) · Anonymous Referee #1 · 29 Apr 2020

**1   General comments**

The atmospheric inverse modeling community has long been waiting for an uncertainty estimate in emission inventories. The lack of such an estimate obligated to make arbitrary assumptions of the uncertainties used in inversions. Since the attribution of emissions to certain regions or processes is highly dependent on the a priori uncertainty assumed, this could lead to wrong results. Therefore, this study is very relevant and an important step into solving this problem and should be published.

However, I find the text and format can be confusing and difficult to read in certain

sections (mainly in sections 1, 2, 3 and 5). I would recommend major reformatting of the text to make it more clear. My main advise would be to view each paragraph as an independent unit of information. The first sentence should give the main take home message of the paragraph. The following sentences should provide supporting information.

**2  Specific comments**

- In the introduction there is a lot of information but there should be more focus on what is the problem, why is it important and what solution is proposed.

- Why the EDGAR sector uncertainty is not purely additive? Please expand on the exemptions.

- On what basis where fuel type assumed, e.g. source or citation? Could you add the assumed fuel type for each sector in a table?

- Emissions from Energy_A, Energy_B and manufacturing are assumed to decrease in the summer. However, data from the US Energy Information Agency suggests that for example natural gas consumption has two seasonal peaks, with consumption patterns predominantly driven by weather. The largest peak occurs during the winter, when cold weather increases the demand for natural gas space heating in the residential and commercial sectors. A second, smaller peak occurs in the summer when air conditioning use increases demand for electric power, which can be provided by natural gas, coal or petroleum-fired generators (Bradley S., 2015 and Comstock O, 2020).

**3  Technical corrections**

- line 41: Since the early 2002s -> 2000s

- line 86: Presence of observations may should better say availability of observations and emission information.

- line 150: lower case S in Savannah

- line 159: What is an autoproducer? Is this an automobile manufacturer?

- line 235: You repeat "per activity" several times

- Table 3 and table 7- why are lower bounds with larger uncertainty than upper bounds

- Better description of ensuring log-normal distribution

- Table S5: why '*', which indicates for residential sector only according to the table caption, on fuel types aviation fuel, motor gasoline, etc?

- I find too many acronyms difficult to follow, make text confusing: AD, NIR, TFI, EF, LDS, WDS, GLB, L, U etc.

- Figures 1 and 2 have text over background images and color of boxes make it difficult to read especially if printed in gray scale, much of it should be rather explained in the text.

- Indenting or centering of equations to distinguish them better from normal text.

- Could section 4.1 be largely substituted by a table and map?

- Please consider adding section S3 to main text as it makes the log-normal distribution more clear.

**4 References**

Bradley, S, 2015, Natural gas use features two seasonal peaks per year, https://www.eia.gov/todayinenergy/detail.php?id=22892

Comstock, O., 2020, U.S. natural gas consumption has both winter and summer peaks, https://www.eia.gov/todayinenergy/detail.php?id=42815

---

## Referee Comment (RC2) · Anonymous Referee #2 · 12 May 2020

The estimation of uncertainties in fossil fuel emissions inventories is an important goal. However, this paper is very difficult to follow. It needs major revision to clarify the details of the study undertaken, its results, and its context in the field. General comments are given below. Specific comments are also provided for the first few pages to give examples of the corrections needed, but the writing and presentation of the study throughout the other sections needs to be improved.

General comments

It is not very clear from the abstract what the actual data product is – emissions uncertainties by sector for each country? For individual grid cells?

[Figure]

Why are emissions uncertainties by sector for each country needed? Does the ECMWF data assimilation system calculate posterior fluxes for individual countries?

The paper does not address the uncertainties in spatial allocation of emissions at all, which could be much larger.

Introduction is not sufficient. It should describe - other studies that estimate emissions uncertainty, their methods and results - "the ECMWF model" (L113) and how it will use the results of this study - methods for spatial allocation of emissions to grid cells by EDGAR

The paper is not clearly organized into sections like methods, results and discussion. There is a lot of background material in the "Comparison and discussion" section.

All of section 2 is very unclear and hard to follow. It needs to be rewritten. What does it mean that "An adequate size for the inversion system of the ECMWF model is less than 50 and a covariance matrix of $7\times7$ has been chosen"?

What is the motivation for separating the super emitting power plants? What is an autoproducer?

In section 3 it is confusing to discuss Tier 1 calculations because it seems like the emissions themselves have already been specified. Are the emissions calculations also Tier 1?

Section 3.2 is hard to follow. Can the authors give an example, and specify which sectors are corrected?

Doesn't Equation 1 assume Gaussian uncertainties? What does it mean that "calculations were performed for upper and lower uncertainty limits separately"? For equation 2, the propagation of uncertainties for sums should not be in percent but in absolute units.

In Table 3 it appears that the lower limits for manufacturing are larger than the upper

limits.

Section 3.4 can be deleted.

Figure 3. It is impossible to read the numbers on the graphs. Why are all the countries shown here WDS countries?

Text on page 16 should be rewritten more clearly and not in bullet point form.

In Figure 4, the authors should add an additional bar to the chart representing total emissions because there is too much text at the top of each panel. Aren't all the datasets omitting biofuels? Why does the "other" category have so much higher uncertainty in CHE, also shown in Table 9? The sentence explaining this graph is very long and confusing.

Table 8 should include references.

Section 4.5 should be removed. Figure 6 is extraneous to this study and the simulations are not described at all, and Figure 7 appears to be already published in McNorton et al. 2020.

In the conclusions it says that "The CHE_EDGAR-ECMWF_2015 represents the 2015 fossil CO2 emissions prior at $0.1° \times 0.1°$ resolution that has been for the first time to our knowledge completed with full uncertainty information with global coverage." This is not true because the uncertainty in spatial allocation of emissions has not been considered. And what about the other datasets that report uncertainties listed in Table 1? Furthermore, there is not even a description of how uncertainties are specified at the grid cell level in this paper – the uncertainties seem to be only given for country totals. The dataset is only described in the Conclusions, but it should be described earlier in the paper with all the details on how grid cell values are specified.

For the actual datasets, users should be able to download these individually as needed rather than having to download everything in a large zipped folder.

Specific comments from the first few pages

Title – the results of this work are the uncertainties only, right? The emissions themselves are already reported by EDGAR? How much different are they from EDGAR? How will the uncertainties be used "as prior for Earth System Modelling"?

L12 How do emissions raise awareness? Rephrase.

L15 prior should be defined. The word prior is probably unnecessary here because the results could have more uses than just as a prior.

L15 Are power and energy different? If not, the same word should be used.

L17 Here and elsewhere (L25, Section 3.4) it seems misleading to say covariance and covariance matrices estimated when actually covariances are just assumed to be zero.

L18 Are the $CO_2$ emissions really going to be included in IFS? I suspect they will be used with IFS in the CAMS reanalysis.

L21 How large are these changes to EDGAR emissions?

L26-31. Hard to understand. Please give some values on the uncertainties, and describe the sensitivity tests a bit more.

L36 I think you mean to say "climate change" rather than "the Earth's radiative balance and climate stability".

L37 'long carbon cycle" should be replaced by "fossil fuel" throughout. This definition is unclear and it would include wood.

L41 "early 2002s"?

L43-8. Sentence needs to be revised or deleted.

L51 Observation not Observatory

L61 emissions not concentrations

L63 What is the Mitchell 1984 reference, and why is it cited when referring to the year 2018?

L68 Andrew 2020 is not in the reference list

Table 1. FFDAS says resolution is annual, then in "Note" it says hourly. In general, the information given in "Note" for each dataset seems random.

L73 Global emissions are the same in CDIAC and ODIAC

L75 3 of the datasets in Table 1 include uncertainties, according to "Note"

L77-8 Unclear

L80 delete "with long carbon cycle"

L81-3 Uncertainties on a 0.1degree grid? What about your revised estimates of emissions?

L86, 90 Incomplete sentences, should start with "the"

L89 Delete – it's not true that there was a stagnation since 2015, it has increased since then.

L100 The Paris Agreement limit is not really 1.5C

Page 6. This entire page is difficult to understand.

L165-6 Needs reference

---

## Author Comment (AC1) · 30 Jul 2020

Final Author Comments to the Anonymous Referee #1 and Anonymous Referee #2 Comments to the manuscript of Margarita Choulga et al. "Global anthropogenic CO2 emissions and uncertainties as prior for Earth system modelling and data assimilation"

Dear Anonymous Referee #1 and Anonymous Referee #2, thank you for the positive evaluation and useful comments. We have expanded considerable effort to address all comments and to improve the manuscript in all its parts: text, figures, tables. We believe that all comments and concerns raised have now been addressed. Please find below our detailed responses to your comments.

[Figure]

Dear Editor, in the supplement there is the revised final version of our manuscript.

Anonymous Reviewer #1 comments and Authors reply 1 General comments The atmospheric inverse modeling community has long been waiting for an uncertainty estimate in emission inventories. The lack of such an estimate obligated to make arbitrary assumptions of the uncertainties used in inversions. Since the attribution of emissions to certain regions or processes is highly dependent on the a priori uncertainty assumed, this could lead to wrong results. Therefore, this study is very relevant and an important step into solving this problem and should be published. REPLY: We thank the reviewer for the supportive comment highlighting the relevance of the study and recommending publication. Indeed, also the Global Emissions Initiative underlined at its recent conference on 23rd June 2020 that the uncertainty assessment of gridded emissions input is urgently needed by atmospheric modellers.

However, I find the text and format can be confusing and difficult to read in certain sections (mainly in sections 1, 2, 3 and 5). I would recommend major reformatting of the text to make it more clear. My main advise would be to view each paragraph as an independent unit of information. The first sentence should give the main take home message of the paragraph. The following sentences should provide supporting information. REPLY: We took on board the comment by the reviewer and have revised the text throughout to ease the reading of the manuscript and help the readers to establish the main messages. The changes have been tracked in the revised submission and major examples are given below: For section 1: - We have included a description of the atmospheric exchanges of carbon between the biosphere, ocean and fossil sources within one single paragraph, that is introduced with the sentence summarising the $CO_2$ growth rate variation and trend. - We have summarised the overview of global gridded anthropogenic $CO_2$ emission datasets with their uncertainties within one paragraph, using also a new Table 1. For section 2: - We have reduced the description of EDGARv4.3.2 in section 2.1 and described the three consecutive modifications on the EDGARv4.3.2_FT2015 dataset to generate the CHE_EDGAR-ECMWF_2015 dataset
with three consecutive paragraphs. For section 3: - The core of the paper, section 3.2 has been completely reformatted using subsections to explain the different steps of the uncertainty calculation and using tables and even an example to help the reader retracing back the uncertainty results. For section 5: - We included the summary of CO2 uncertainty comparison in Table 7. - We reworked former Fig. 4 into new Fig. 6 and restructured the discussion and intercomparison of the results.

2 Specific comments • In the introduction there is a lot of information but there should be more focus on what is the problem, why is it important and what solution is proposed. REPLY: We thank the reviewer for the constructive comment. We have revised the introduction and made it more concise and less dispersive. We tried to eliminate non-essential information from the Introduction and rewrote it in a clearer manner. We considerably shortened the description of the different datasets and the discussion around the base year 2015.

• Why the EDGAR sector uncertainty is not purely additive? Please expand on the exemptions. REPLY: The EDGAR inventory is estimated based upon the sum of terms, each of which is a product (e.g., of emission factors and activity data). Based on the suggestion from IPCC (2006) the error propagation approach is not exact for such multiplicative terms, and corrections should be introduced.

• On what basis where fuel type assumed, e.g. source or citation? Could you add the assumed fuel type for each sector in a table? REPLY: The EDGAR emission database contains highly disaggregated activity data and emission factors which account for human activity sector and subsector, fuel type, technology specifications and cover all anthropogenic emitting sources of CO2. Emission factors by fuel type are mostly derived following the IPCC (2006) guidelines. The development of the EDGAR data base is comprehensively detailed in Janssens-Maenhout et al. (2019) and references therein. We added typical fuel types for each sector in Table 2.

• Emissions from Energy_A, Energy_B and manufacturing are assumed to decrease

in the summer. However, data from the US Energy Information Agency suggests that for example natural gas consumption has two seasonal peaks, with consumption patterns predominantly driven by weather. The largest peak occurs during the winter, when cold weather increases the demand for natural gas space heating in the residential and commercial sectors. A second, smaller peak occurs in the summer when air conditioning use increases demand for electric power, which can be provided by natural gas, coal or petroleum-fired generators (Bradley S., 2015 and Comstock O, 2020). REPLY: The Energy_A and Energy_B sectors as well as manufacturing are assumed to slightly decrease because of the summer holiday break. The natural gas consumption with two seasonal peaks are rather seen in the Settlements sector, which are indeed rather weather driven. We do agree that an update of the temporal profiles could be useful in a next step and would use for that the data of Crippa et al. (2020) .

3 Technical corrections • line 41: Since the early 2002s -> 2000s REPLY: Corrected.

• line 86: Presence of observations may should better say availability of observations and emission information. REPLY: Corrected.

• line 150: lower case S in Savannah REPLY: Corrected.

• line 159: What is an autoproducer? Is this an automobile manufacturer? REPLY: Autoproducers is the energy generated and used specifically for industrial purposes and manufacturing. We added this explanation to the main text.

• line 235: You repeat "per activity" several times REPLY: Corrected.

• Table 3 and table 7- why are lower bounds with larger uncertainty than upper bounds REPLY: Some uncertainty ranges for emission factors and/or activity data in IPCC (2006) and IPCC-TFI (2019) are not symmetrical and have higher uncertainty values for the lower bound than for the upper bound, due to expert knowledge or in-situ data available (these are the base for IPCC values), which lead to the same pattern in final prior uncertainty bounds. Tables 4 and 11 columns "Prior uncertainty bounds, %"

show values based purely on IPCC, so not yet fully corrected to lognormal distribution as for that you need budget values per country/sector – final uncertainties are shown is Tables 4 and 11 columns "Uncertainty bounds, %". We added this explanation to the main text.

• Better description of ensuring log-normal distribution REPLY: We have rewritten an explanation for the yearly uncertainty calculation and added an example how uncertainties were calculated for two different countries TRANSPORT emission group.

• Table S5: why '*', which indicates for residential sector only according to the table caption, on fuel types aviation fuel, motor gasoline, etc? REPLY: Unfortunate misprint. Corrected.

• I find too many acronyms difficult to follow, make text confusing: AD, NIR, TFI, EF, LDS, WDS, GLB, L, U etc. REPLY: We have removed all acronyms that do not refer to international organisations or their reporting in the text. In some tables we still had to use few acronyms to save the space, every acronym is explained in table caption.

• Figures 1 and 2 have text over background images and color of boxes make it difficult to read especially if printed in gray scale, much of it should be rather explained in the text. REPLY: We complemented the main text with an explanation and adopted a more transparent background colours for the figures.

• Indenting or centering of equations to distinguish them better from normal text. REPLY: Unfortunately, ESSD template does not allow changes to the current format of the equations.

• Could section 4.1 be largely substituted by a table and map? REPLY: We have substituted this section by Table 9.

• Please consider adding section S3 to main text as it makes the log-normal distribution more clear. REPLY: Done.

4 References Bradley, S, 2015, Natural gas use features two seasonal

peaks per year, https://www.eia.gov/todayinenergy/detail.php?id=22892 Comstock, O., 2020, U.S. natural gas consumption has both winter and summer peaks, https://www.eia.gov/todayinenergy/detail.php?id=42815 REPLY: We have included the proposed references in the revised manuscript.

Anonymous Reviewer #2 comments and Authors reply The estimation of uncertainties in fossil fuel emissions inventories is an important goal. However, this paper is very difficult to follow. It needs major revision to clarify the details of the study undertaken, its results, and its context in the field. General comments are given below. Specific comments are also provided for the first few pages to give examples of the corrections needed, but the writing and presentation of the study throughout the other sections needs to be improved. REPLY: We thank the reviewer for the useful suggestions, and we have revised the flow of the paper aiming at improving clarity of exposition and description of methods. We believe the paper now reads more easily and that key messages are now easier to grasp.

General comments It is not very clear from the abstract what the actual data product is – emissions uncertainties by sector for each country? For individual grid cells? REPLY: We have added a short description of the dataset to the abstract: "CHE_EDGAR-ECMWF_2015 consists of 11 global NetCDF files with gridded yearly and monthly upper and lower bounds of uncertainties in % and kgÅům-2Åůs-1 per each ECMWF group and their sum, and 1 Excel file with 16 spreadsheets with the same information listed per country (metadata, emissions, uncertainties, statistical parameters).".

Why are emissions uncertainties by sector for each country needed? Does the ECMWF data assimilation system calculate posterior fluxes for individual countries? REPLY: The only source of internationally accepted anthropogenic CO2 emission uncertainty methodology is IPCC (2006), which provides guidance in estimating and reporting uncertainties associated with the national GHG inventories. National uncertainties were applied uniformly across each country to create a gridded map that later on will be used by an ECMWF data assimilation system, which is currently in the development process to include gridded emission sectors. The resulting inversion system will provide gridded posterior fluxes which can then be aggregated for individual countries. This research is the first step and will be followed by adding spatial uncertainty of the proxies.

The paper does not address the uncertainties in spatial allocation of emissions at all, which could be much larger. REPLY: We take the point. We are aware of this important limitation but the estimation of covariances in the spatial proxy is still a steep hurdle. It requires the assessment for the spatial representativeness of the proxy data used, which varies considerably between the regions and depends on the available information (known point sources and traffic lines for energy and transport sector versus population density as proxy for settlements and other sectors for which local information on the sources are missing at global scale). We acknowledge this in the main text of subsections 3.3 and 3.4, and in the Supplementary Information Section S.3. We will devote effort to this aspect in the next step of our research. We refer to first attempts in this direction with EDGARv5.0 by Crippa et al. (2020), which started to assess uncertainties and spatial representativeness, improving the latter e.g. for the settlement sector with weather related information.

Introduction is not sufficient. It should describe - other studies that estimate emissions uncertainty, their methods and results - "the ECMWF model" (L113) and how it will use the results of this study - methods for spatial allocation of emissions to grid cells by EDGAR REPLY: We added a description of other global $CO_2$ studies that also have calculated uncertainties and we refer to the new Table 1 for a short overview. Results of this study are used in the ECMWF data assimilation system which is documented in Bousserez (2019) . As mentioned in the revised version of the manuscript, the calculated uncertainties documented in this manuscript were already tested in McNorton et al. (2020).

The paper is not clearly organized into sections like methods, results and discussion. There is a lot of background material in the "Comparison and discussion" section. RE-

PLY: Following the reviewer's suggestion, we have reorganized several sections of the paper. We deleted most of the background information from the Comparison section.

All of section 2 is very unclear and hard to follow. It needs to be rewritten. REPLY: We have entirely revised Section 2 to make it clearer and easier to follow. We have shortened it and focused on the description of the three consecutive modifications on the EDGARv4.3.2_FT2015 dataset to generate the CHE_EDGAR-ECMWF_2015 dataset with three consecutive paragraphs.

What does it mean that "An adequate size for the inversion system of the ECMWF model is less than 50 and a covariance matrix of $7 \times 7$ has been chosen"? REPLY: At ECMWF we propose to use a 4D-Var and ensemble-based hybrid inversion system. For this reason, only an ensemble size of up to 50 members at the global scale is currently viable. Based on these technical requirements we need a reduced state vector, which requires to aggregate multiple EDGAR sectors into 7 ECMWF emission groups in order to reduce the size of the covariance matrix. We have also reformulated this sentence in the text.

What is the motivation for separating the super emitting power plants? What is an autoproducer? REPLY: The reasoning behind is that the large power plants are operating usually at their maximum capacity, where standard power plants operate on day-to-day basis; also large power plants are large CO2 point sources, generating CO2 plumes that can be directly observed by in-situ or space borne measurements and these CO2 "base-load" emissions contribute a considerable and constant share to the national total. Therefore, their uncertainty is different. All the super power plants that were identified were also verified on their location, so that the spatial representativeness is no issue. For their uncertainty, we assume that they operate at full capacity and maximum availability. According to expert knowledge the upper bound of uncertainty for such supper power plants is smaller (+3.0 %) than for standard power plants, which operate based on day-to-day needs. The manuscript has been updated to clarify better this choice. Autoproducers are defined by IEA energy statistical office and include the

energy (electricity and heat) generated by an industry for its own use, mostly for the manufacturing. We added this explanation to the main text.

In section 3 it is confusing to discuss Tier 1 calculations because it seems like the emissions themselves have already been specified. Are the emissions calculations also Tier 1? REPLY: IPCC uses a tiered approach to calculate uncertainty and to estimate emission factors. Section 3 discusses uncertainty calculations according to Tier 1 (sometimes Tier 2 – fuel specific) approach from IPCC (2006) guidelines. At the same time EDGAR emissions were also calculated according to Tier 1 (sometimes Tier 2) approach., such that the uncertainty calculation is completely consistent with the bottom-up emission calculation. We have tried to explain these aspects in the main text.

Section 3.2 is hard to follow. Can the authors give an example, and specify which sectors are corrected? REPLY: We have reorganized this section and added in consecutive subsections all information on the corrective steps for the uncertainty calculations which we illustrate with an example for the TRANSPORT emission group in two countries with different statistical infrastructure.

Doesn't Equation 1 assume Gaussian uncertainties? What does it mean that "calculations were performed for upper and lower uncertainty limits separately"? REPLY: We performed all calculations separately for upper and lower uncertainty bounds, because IPCC (2006) guidelines provide non-symmetrical ranges, which we wanted to preserve. Tier 1 suggests using higher uncertainty values to create symmetrical ranges, but it was significantly inflating uncertainties and therefore we refined the calculation as recommended by IPCC (2006).

For equation 2, the propagation of uncertainties for sums should not be in percent but in absolute units. REPLY: According to explanation for Equation 3.1 in Volume 1, Chapter 3, IPCC (2006) values should be expressed as a percentage.

In Table 3 it appears that the lower limits for manufacturing are larger than the upper

limits. REPLY: Some uncertainty ranges for emission factors and/or activity data in IPCC (2006) and IPCC-TFI (2019) are not symmetrical and have higher uncertainty values for the lower bound than for the upper bound, due to expert knowledge or in-situ data available (on which IPCC (2006) default values are based), which lead to the same pattern in final prior uncertainty bounds. Tables 4 and 11 columns "Prior uncertainty bounds, %" show values based purely on IPCC (2006), not yet corrected to lognormal distribution. For the correction we need budget values per country/sector and final uncertainties are shown is Tables 4 and 11 columns "Uncertainty bounds, %". We added this explanation to the main text.

Section 3.4 can be deleted. REPLY: We have deleted Section 3.4 and moved the information on the covariance matrix to the Supplementary Information, to which refer a few sentences at the end of Section 3.3.

Figure 3. It is impossible to read the numbers on the graphs. Why are all the countries shown here WDS countries? REPLY: The manuscript will be provided with high resolution figures to better see all the details. We appreciate the interest and suggestion of the reviewer to show also a LDS country and added the Russian Federation as LDS country in Fig.3.

Text on page 16 should be rewritten more clearly and not in bullet point form. REPLY: We have substituted this section by Table 9 for more clarity.

In Figure 4, the authors should add an additional bar to the chart representing total emissions because there is too much text at the top of each panel. Aren't all the datasets omitting biofuels? Why does the "other" category have so much higher uncertainty in CHE, also shown in Table 9? The sentence explaining this graph is very long and confusing. REPLY: We have rearranged Figure 6 in the updated version of the manuscript. All datasets, except ours take biofuels (e.g. blended within the fossil oil) into account. The OTHER emission group has several extremely high uncertain activities. Since we have only the sum of all these activities we have to assume that

all of them are emitted in the same proportion; other datasets have more detailed information and can skip activities with very high uncertainties if their emissions were zero.

Table 8 should include references. REPLY: An extra column has been added to the table that includes the main references Andrew (2020), Hoesley et al. (2018), Janssens-Maenhout et al. (2019), Andres et al (2016), Friedlingstein et al. (2019).

Section 4.5 should be removed. Figure 6 is extraneous to this study and the simulations are not described at all, and Figure 7 appears to be already published in McNorton et al. 2020. REPLY: We have adjusted this section accordingly – deleted Figure 7 and text referring to it, yet we think that it is important to show what impact detailed source distribution has and stress that this is very important to collect detailed information on emission source allocation. The simulations are explained in detail in McNorton et al. (2020). In the revised manuscript we just give a small summary and refer to the McNorton et al. (2020) paper.

In the conclusions it says that "The CHE_EDGAR-ECMWF_2015 represents the 2015 fossil CO2 emissions prior at 0.1âŮȩ×0.1âŮȩ resolution that has been for the first time to our knowledge completed with full uncertainty information with global coverage." This is not true because the uncertainty in spatial allocation of emissions has not been considered. And what about the other datasets that report uncertainties listed in Table 1? Furthermore, there is not even a description of how uncertainties are specified at the grid cell level in this paper – the uncertainties seem to be only given for country totals. The dataset is only described in the Conclusions, but it should be described earlier in the paper with all the details on how grid cell values are specified. REPLY: We agreed to the need for a more refined description of the CHE_EDGAR-ECMWF_2015 dataset and its strength and rephrased it as follows: "The CHE_EDGAR-ECMWF_2015 represents the 2015 global fossil CO2 emissions at 0.1âŮȩ×0.1âŮȩ resolution that has been for the first time to our knowledge bridging the inventory community and the atmospheric modelling community. In fact, the uncertainty calculations fully respect the

detailed error propagation approach recommended by IPCC (2006) guidelines for GHG inventories while the input datasets were processed such that the uncertainty information could be fully taken up by the ECMWF model IFS." Moreover, we emphasised in the main text that currently calculated national uncertainties are applied uniformly across each country to create a gridded map, and that these uncertainties do not take into account spatial allocation, which would be the next step of our research. We have added extra information on other global CO2 datasets that also provide uncertainty information – CDIAC, ODIAC, FFDAS and PKU-FUEL. We have added a description on how the calculated uncertainties were specified at the grid-cell level. The dataset description has also been relocated to a new Section 4. It is also worth noting that the Global Emissions Initiative underlined at its recent conference on 23rd June 2020 the need to address the uncertainty assessment of gridded emission inputs as a crucial piece of information for atmospheric modellers and that this still requires further research efforts.

For the actual datasets, users should be able to download these individually as needed rather than having to download everything in a large zipped folder. REPLY: This has been corrected and the big folder has been split in more clear subfolders. As such, a new Zenodo link is introduced.

Specific comments from the first few pages Title – the results of this work are the uncertainties only, right? The emissions themselves are already reported by EDGAR? How much different are they from EDGAR? How will the uncertainties be used "as prior for Earth System Modelling"? REPLY: The main result of this work are the uncertainties and reprocessing of the EDGARv4.3.2_FT2015 dataset as prior input for the ECMWF atmospheric model. The update of EDGARv4.3.2_FT2015 and difference with CHE_EDGAR-ECMWF_2015 dataset has been clearly described (incl. Table S3 in the Supplementary). Uncertainties will be used in the data assimilation part of the ECMWF IFS model.

L12 How do emissions raise awareness? Rephrase. REPLY: This has been rephrased

as "For an increased understanding of the CO2 emission sources, patterns and trends, a link between the emission inventories and observed CO2 concentrations is best established via Earth system modelling and data assimilation.".

L15 prior should be defined. The word prior is probably unnecessary here because the results could have more uses than just as a prior. REPLY: This has been rephrased, avoiding the term "prior".

L15 Are power and energy different? If not, the same word should be used. REPLY: This has been corrected: we use energy production consequently throughout the paper.

L17 Here and elsewhere (L25, Section 3.4) it seems misleading to say covariance and covariance matrices estimated when actually covariances are just assumed to be zero. REPLY: This assumption is suggested by IPCC (2006) guidelines and it is currently used in our research. The main text was updated to better represent work done concerning covariance matrices.

L18 Are the CO2 emissions really going to be included in IFS? I suspect they will be used with IFS in the CAMS reanalysis. REPLY: These CO2 emissions and uncertainties have been used in the ECMWF IFS CO2 ensemble simulations (McNorton et al., 2020) and in the CHE tier-2 high resolution nature run (https://www.che-project.eu/sites/default/files/2020-01/CHE-D2-6-V1-0.pdf, Agusti-Panareda et al., in preparation). In the near future these CO2 emissions and uncertainties will be used by the CAMS inversion system (currently under development).

L21 How large are these changes to EDGAR emissions? REPLY: Updated improved apportionment of the energy sector decreased emissions by 8 %, and the energy usage for manufacturing increased by 18 %. The extra emission source of the diffusive CO2 emissions from coal mines added 7 Mtons globally but localised to few regions.

L26-31. Hard to understand. Please give some values on the uncertainties, and describe the sensitivity tests a bit more. REPLY: The text is revised, and values and some extra explanation on sensitivity tests are provided.

L36 I think you mean to say "climate change" rather than "the Earth's radiative balance and climate stability". REPLY: This has been rephrased accordingly.

L37 'long carbon cycle" should be replaced by "fossil fuel" throughout. This definition is unclear and it would include wood. REPLY: We have changed it to long-cycle carbon and added clear definition with reference. This is also consistent with the definitions in the paper of Janssens-Maenhout et al. (2019) on EDGARv4.3.2.

L41 "early 2002s"? REPLY: This has been corrected.

L43-8. Sentence needs to be revised or deleted. REPLY: This has been revised.

L51 Observation not Observatory REPLY: This has been corrected.

L61 emissions not concentrations REPLY: This has been corrected.

L63 What is the Mitchell 1984 reference, and why is it cited when referring to the year 2018? REPLY: Mitchell et al. (1984) had conversion factors for different emission units, as this can be rather easily recalculated this citation was deleted.

L68 Andrew 2020 is not in the reference list REPLY: The reference is added to the list.

Table 1. FFDAS says resolution is annual, then in "Note" it says hourly. In general, the information given in "Note" for each dataset seems random. REPLY: Table 1 was updated with more precise wording.

L73 Global emissions are the same in CDIAC and ODIAC REPLY: This has been corrected accordingly in the text.

L75 3 of the datasets in Table 1 include uncertainties, according to "Note" REPLY: Description of these three datasets and short explanation how their uncertainties were calculated are added to the main text.

L77-8 Unclear REPLY: We found this sentence not relevant for the explanation of our research and deleted it.

L80 delete "with long carbon cycle" REPLY: This has been deleted.

L81-3 Uncertainties on a 0.1degree grid? What about your revised estimates of emissions? REPLY: National sectoral uncertainties and revised emissions are both uniformly mapped onto a regular latitude/longitude $0.1° \times 0.1°$ resolution grid.

L86, 90 Incomplete sentences, should start with "the" REPLY: Theses have been corrected.

L89 Delete – it's not true that there was a stagnation since 2015, it has increased since then. REPLY: This sentence has been deleted.

L100 The Paris Agreement limit is not really 1.5C REPLY: We have rephrased the abstract.

Page 6. This entire page is difficult to understand. REPLY: We have revised the entire Section 2 for more clarity.

L165-6 Needs reference REPLY: The text was updated and the following reference was added: Beamish, B.B., and Vance, W.E.: Greenhouse gas contributions from coal mining in Australia and New Zealand, Journal of the Royal Society of New Zealand, 22:2, 153-156, doi:10.1080/03036758.1992.10420812, 1992.

---

## Author Response (AR1)

**Final Author Comments to the Anonymous Referee #1 and Anonymous Referee #2**

Comments to the manuscript of Margarita Choulga et al. "Global anthropogenic CO2 emissions and uncertainties as prior for Earth system modelling and data assimilation"

Dear Anonymous Referee #1 and Anonymous Referee #2, thank you for the positive evaluation and useful comments. We have expanded considerable effort to address all comments and to improve the manuscript in all its parts: text, figures, tables. We believe that all comments and concerns raised have now been addressed. Please find below our detailed responses to your comments.

Dear Editor, in the supplement there is the revised final version of our manuscript.

*Anonymous Reviewer #1 comments and Authors reply*

*1 General comments*

*The atmospheric inverse modeling community has long been waiting for an uncertainty estimate in emission inventories. The lack of such an estimate obligated to make arbitrary assumptions of the uncertainties used in inversions. Since the attribution of emissions to certain regions or processes is highly dependent on the a priori uncertainty assumed, this could lead to wrong results. Therefore, this study is very relevant and an important step into solving this problem and should be published.*

We thank the reviewer for the supportive comment highlighting the relevance of the study and recommending publication. Indeed, also the Global Emissions Initiative underlined at its recent conference on 23$^{rd}$ June 2020 that the uncertainty assessment of gridded emissions input is urgently needed by atmospheric modellers.

*However, I find the text and format can be confusing and difficult to read in certain sections (mainly in sections 1, 2, 3 and 5). I would recommend major reformatting of the text to make it more clear. My main advise would be to view each paragraph as an independent unit of information. The first sentence should give the main take home message of the paragraph. The following sentences should provide supporting information.*

We took on board the comment by the reviewer and have revised the text throughout to ease the reading of the manuscript and help the readers to establish the main messages. The changes have been tracked in the revised submission and major examples are given below:

For section 1:

-        We have included a description of the atmospheric exchanges of carbon between the biosphere, ocean and fossil sources within one single paragraph, that is introduced with the sentence summarising the $CO_2$ growth rate variation and trend.

-        We have summarised the overview of global gridded anthropogenic $CO_2$ emission datasets with their uncertainties within one paragraph, using also a new Table 1.

For section 2:

-        We have reduced the description of EDGARv4.3.2 in section 2.1 and described the three consecutive modifications on the EDGARv4.3.2_FT2015 dataset to generate the CHE_EDGAR-ECMWF_2015 dataset with three consecutive paragraphs.

For section 3:

-        The core of the paper, section 3.2 has been completely reformatted using subsections to explain the different steps of the uncertainty calculation and using tables and even an example to help the reader retracing back the uncertainty results.

For section 5:

-        We included the summary of $CO_2$ uncertainty comparison in Table 7.

-        We reworked former Fig. 4 into new Fig. 6 and restructured the discussion and intercomparison of the results.

*2 Specific comments*

*• In the introduction there is a lot of information but there should be more focus on what is the problem, why is it important and what solution is proposed.*

We thank the reviewer for the constructive comment. We have revised the introduction and made it more concise and less dispersive. We tried to eliminate non-essential information from the Introduction and rewrote it in a clearer manner. We considerably shortened the description of the different datasets and the discussion around the base year 2015.

*• Why the EDGAR sector uncertainty is not purely additive? Please expand on the exemptions.*

The EDGAR inventory is estimated based upon the sum of terms, each of which is a product (e.g., of emission factors and activity data). Based on the suggestion from IPCC (2006) the error propagation approach is not exact for such multiplicative terms, and corrections should be introduced.

*• On what basis where fuel type assumed, e.g. source or citation? Could you add the assumed fuel type for each sector in a table?*

The EDGAR emission database contains highly disaggregated activity data and emission factors which account for human activity sector and subsector, fuel type, technology specifications and cover all anthropogenic emitting sources of $CO_2$. Emission factors by fuel type are mostly derived following the IPCC (2006) guidelines. The development of the EDGAR data base is comprehensively detailed in Janssens-Maenhout et al. (2019) and references therein. We added typical fuel types for each sector in Table 2.

*• Emissions from Energy_A, Energy_B and manufacturing are assumed to decrease in the summer. However, data from the US Energy Information Agency suggests that for example natural gas consumption has two seasonal peaks, with consumption patterns predominantly driven by weather. The largest peak occurs during the winter, when cold weather increases the demand for natural gas space heating in the residential and commercial sectors. A second, smaller peak occurs in the summer when air conditioning use increases demand for electric power, which can be provided by natural gas, coal or petroleum-fired generators (Bradley S., 2015 and Comstock O, 2020).*

The Energy_A and Energy_B sectors as well as manufacturing are assumed to slightly decrease because of the summer holiday break. The natural gas consumption with two seasonal peaks are rather seen in the Settlements sector, which are indeed rather weather driven. We do agree that an update of the temporal profiles could be useful in a next step and would use for that the data of Crippa et al. (2020)[1].

*3 Technical corrections*

*• line 41: Since the early 2002s -> 2000s*

Corrected.

*• line 86: Presence of observations may should better say availability of observations and emission information.*

Corrected.

*• line 150: lower case S in Savannah*

Corrected.

*• line 159: What is an autoproducer? Is this an automobile manufacturer?*

Autoproducers is the energy generated and used specifically for industrial purposes and manufacturing. We added this explanation to the main text.

*• line 235: You repeat "per activity" several times*

Corrected.

*• Table 3 and table 7- why are lower bounds with larger uncertainty than upper bounds*

Some uncertainty ranges for emission factors and/or activity data in IPCC (2006) and IPCC-TFI (2019) are not symmetrical and have higher uncertainty values for the lower bound than for the upper bound, due to expert knowledge or in-situ data available (these are the base for IPCC values), which lead to the same pattern in final prior uncertainty bounds. Tables 4 and 11 columns "Prior uncertainty bounds, %" show values based purely on IPCC, so not yet fully corrected to lognormal distribution as for that you need budget values per country/sector – final uncertainties are shown is Tables 4 and 11 columns "Uncertainty bounds, %". We added this explanation to the main text.

*• Better description of ensuring log-normal distribution*

We have rewritten an explanation for the yearly uncertainty calculation and added an example how uncertainties were calculated for two different countries TRANSPORT emission group.
* * *
[1] https://www.nature.com/articles/s41597-020-0462-2

• *Table S5: why '*', which indicates for residential sector only according to the table caption, on fuel types aviation fuel, motor gasoline, etc?*

Unfortunate misprint. Corrected.

• *I find too many acronyms difficult to follow, make text confusing: AD, NIR, TFI, EF, LDS, WDS, GLB, L, U etc.*

We have removed all acronyms that do not refer to international organisations or their reporting in the text. In some tables we still had to use few acronyms to save the space, every acronym is explained in table caption.

• *Figures 1 and 2 have text over background images and color of boxes make it difficult to read especially if printed in gray scale, much of it should be rather explained in the text.*

We complemented the main text with an explanation and adopted a more transparent background colours for the figures.

• *Indenting or centering of equations to distinguish them better from normal text.*

Unfortunately, ESSD template does not allow changes to the current format of the equations.

• *Could section 4.1 be largely substituted by a table and map?*

We have substituted this section by Table 9.

• *Please consider adding section S3 to main text as it makes the log-normal distribution more clear.*

Done.

*4 References*

*Bradley, S, 2015, Natural gas use features two seasonal peaks per year, https://www.eia.gov/todayinenergy/detail.php?id=22892*

*Comstock, O., 2020, U.S. natural gas consumption has both winter and summer peaks, https://www.eia.gov/todayinenergy/detail.php?id=42815*

We have included the proposed references in the revised manuscript.

*Anonymous Reviewer #2 comments and Authors reply*
*The estimation of uncertainties in fossil fuel emissions inventories is an important goal. However, this paper is very difficult to follow. It needs major revision to clarify the details of the study undertaken, its results, and its context in the field. General comments are given below. Specific comments are also provided for the first few pages to give examples of the corrections needed, but the writing and presentation of the study throughout the other sections needs to be improved.*

We thank the reviewer for the useful suggestions, and we have revised the flow of the paper aiming at improving clarity of exposition and description of methods. We believe the paper now reads more easily and that key messages are now easier to grasp.

*General comments*
*It is not very clear from the abstract what the actual data product is – emissions uncertainties by sector for each country? For individual grid cells?*

We have added a short description of the dataset to the abstract: "CHE_EDGAR-ECMWF_2015 consists of 11 global NetCDF files with gridded yearly and monthly upper and lower bounds of uncertainties in % and kg·m$^{-2}$·s$^{-1}$ per each ECMWF group and their sum, and 1 Excel file with 16 spreadsheets with the same information listed per country (metadata, emissions, uncertainties, statistical parameters).".

*Why are emissions uncertainties by sector for each country needed? Does the ECMWF data assimilation system calculate posterior fluxes for individual countries?*

The only source of internationally accepted anthropogenic $CO_2$ emission uncertainty methodology is IPCC (2006), which provides guidance in estimating and reporting uncertainties associated with the national GHG inventories. National uncertainties were applied uniformly across each country to create a gridded map that later on will be used by an ECMWF data assimilation system, which is currently in the development process to include gridded emission sectors. The resulting inversion system will provide gridded posterior fluxes which can then be aggregated for individual countries. This research is the first step and will be followed by adding spatial uncertainty of the proxies.

*The paper does not address the uncertainties in spatial allocation of emissions at all, which could be much larger.*

We take the point. We are aware of this important limitation but the estimation of covariances in the spatial proxy is still a steep hurdle. It requires the assessment for the spatial representativeness of the proxy data used, which varies considerably between the regions and depends on the available information (known point sources and traffic lines for energy and transport sector versus population density as proxy for settlements and other sectors for which local information on the sources are missing at global scale). We acknowledge this in the main text of subsections 3.3 and 3.4, and in the Supplementary Information Section S.3. We will devote effort to this aspect in the next step of our research. We refer to first attempts in this direction with EDGARv5.0 by Crippa et al. (2020), which started to assess uncertainties and spatial representativeness, improving the latter e.g. for the settlement sector with weather related information.

*Introduction is not sufficient. It should describe - other studies that estimate emissions uncertainty, their methods and results - "the ECMWF model" (L113) and how it will use the results of this study - methods for spatial allocation of emissions to grid cells by EDGAR*

We added a description of other global $CO_2$ studies that also have calculated uncertainties and we refer to the new Table 1 for a short overview.
Results of this study are used in the ECMWF data assimilation system which is documented in Bousserez (2019)[2].
As mentioned in the revised version of the manuscript, the calculated uncertainties documented in this manuscript were already tested in McNorton et al. (2020).

*The paper is not clearly organized into sections like methods, results and discussion. There is a lot of background material in the "Comparison and discussion" section.*

Following the reviewer's suggestion, we have reorganized several sections of the paper. We deleted most of the background information from the Comparison section.

*All of section 2 is very unclear and hard to follow. It needs to be rewritten.*

We have entirely revised Section 2 to make it clearer and easier to follow. We have shortened it and focused on the description of the three consecutive modifications on the EDGARv4.3.2_FT2015 dataset to generate the CHE_EDGAR-ECMWF_2015 dataset with three consecutive paragraphs.
* * *
[2] https://arxiv.org/abs/1910.11727

*What does it mean that "An adequate size for the inversion system of the ECMWF model is less than 50 and a covariance matrix of 7×7 has been chosen"?*

At ECMWF we propose to use a 4D-Var and ensemble-based hybrid inversion system. For this reason, only an ensemble size of up to 50 members at the global scale is currently viable. Based on these technical requirements we need a reduced state vector, which requires to aggregate multiple EDGAR sectors into 7 ECMWF emission groups in order to reduce the size of the covariance matrix. We have also reformulated this sentence in the text.

*What is the motivation for separating the super emitting power plants? What is an autoproducer?*

The reasoning behind is that the large power plants are operating usually at their maximum capacity, where standard power plants operate on day-to-day basis; also large power plants are large $CO_2$ point sources, generating $CO_2$ plumes that can be directly observed by in-situ or space borne measurements and these $CO_2$ "base-load" emissions contribute a considerable and constant share to the national total. Therefore, their uncertainty is different. All the super power plants that were identified were also verified on their location, so that the spatial representativeness is no issue. For their uncertainty, we assume that they operate at full capacity and maximum availability. According to expert knowledge the upper bound of uncertainty for such supper power plants is smaller (+3.0 %) than for standard power plants, which operate based on day-to-day needs. The manuscript has been updated to clarify better this choice.

Autoproducers are defined by IEA energy statistical office and include the energy (electricity and heat) generated by an industry for its own use, mostly for the manufacturing. We added this explanation to the main text.

*In section 3 it is confusing to discuss Tier 1 calculations because it seems like the emissions themselves have already been specified. Are the emissions calculations also Tier 1?*

IPCC uses a tiered approach to calculate uncertainty and to estimate emission factors. Section 3 discusses uncertainty calculations according to Tier 1 (sometimes Tier 2 – fuel specific) approach from IPCC (2006) guidelines. At the same time EDGAR emissions were also calculated according to Tier 1 (sometimes Tier 2) approach., such that the uncertainty calculation is completely consistent with the bottom-up emission calculation. We have tried to explain these aspects in the main text.

*Section 3.2 is hard to follow. Can the authors give an example, and specify which sectors are corrected?*

We have reorganized this section and added in consecutive subsections all information on the corrective steps for the uncertainty calculations which we illustrate with an example for the TRANSPORT emission group in two countries with different statistical infrastructure.

*Doesn't Equation 1 assume Gaussian uncertainties? What does it mean that "calculations were performed for upper and lower uncertainty limits separately"?*

We performed all calculations separately for upper and lower uncertainty bounds, because IPCC (2006) guidelines provide non-symmetrical ranges, which we wanted to preserve. Tier 1 suggests using higher uncertainty values to create symmetrical ranges, but it was significantly inflating uncertainties and therefore we refined the calculation as recommended by IPCC (2006).

*For equation 2, the propagation of uncertainties for sums should not be in percent but in absolute units.*

According to explanation for Equation 3.1 in Volume 1, Chapter 3, IPCC (2006) values should be expressed as a percentage.

*In Table 3 it appears that the lower limits for manufacturing are larger than the upper limits.*

Some uncertainty ranges for emission factors and/or activity data in IPCC (2006) and IPCC-TFI (2019) are not symmetrical and have higher uncertainty values for the lower bound than for the upper bound, due to expert knowledge or in-situ data available (on which IPCC (2006) default values are based), which lead to the same pattern in final prior uncertainty bounds. Tables 4 and 11 columns "Prior uncertainty bounds, %" show values based purely on IPCC (2006), not yet corrected to lognormal distribution. For the correction we need budget values per country/sector and final uncertainties are shown is Tables 4 and 11 columns "Uncertainty bounds, %". We added this explanation to the main text.

*Section 3.4 can be deleted.*

We have deleted Section 3.4 and moved the information on the covariance matrix to the Supplementary Information, to which refer a few sentences at the end of Section 3.3.

*Figure 3. It is impossible to read the numbers on the graphs. Why are all the countries shown here WDS countries?*

The manuscript will be provided with high resolution figures to better see all the details. We appreciate the interest and suggestion of the reviewer to show also a LDS country and added the Russian Federation as LDS country in Fig.3.

*Text on page 16 should be rewritten more clearly and not in bullet point form.*

We have substituted this section by Table 9 for more clarity.

*In Figure 4, the authors should add an additional bar to the chart representing total emissions because there is too much text at the top of each panel. Aren't all the datasets omitting biofuels? Why does the "other" category have so much higher uncertainty in CHE, also shown in Table 9? The sentence explaining this graph is very long and confusing.*

We have rearranged Figure 6 in the updated version of the manuscript.

All datasets, except ours take biofuels (e.g. blended within the fossil oil) into account.

The OTHER emission group has several extremely high uncertain activities. Since we have only the sum of all these activities we have to assume that all of them are emitted in the same proportion; other datasets have more detailed information and can skip activities with very high uncertainties if their emissions were zero.

*Table 8 should include references.*

An extra column has been added to the table that includes the main references Andrew (2020)[3], Hoesley et al. (2018), Janssens-Maenhout et al. (2019), Andres et al (2016), Friedlingstein et al. (2019).

*Section 4.5 should be removed. Figure 6 is extraneous to this study and the simulations are not described at all, and Figure 7 appears to be already published in McNorton et al. 2020.*

We have adjusted this section accordingly – deleted Figure 7 and text referring to it, yet we think that it is important to show what impact detailed source distribution has and stress that this is very important to collect detailed information on emission source allocation. The simulations are explained in detail in McNorton et al. (2020). In the revised manuscript we just give a small summary and refer to the McNorton et al. (2020) paper.

*In the conclusions it says that "The CHE_EDGAR-ECMWF_2015 represents the 2015 fossil CO2 emissions prior at 0.1°×0.1° resolution that has been for the first time to our knowledge completed with full uncertainty information with global coverage." This is not true because the uncertainty in spatial allocation of emissions has not been considered. And what about the other datasets that report uncertainties listed in Table 1? Furthermore, there is not even a description of how uncertainties are specified at the grid cell level in this paper – the uncertainties seem to be only given for country totals. The dataset is only described in the Conclusions, but it should be described earlier in the paper with all the details on how grid cell values are specified.*

We agreed to the need for a more refined description of the CHE_EDGAR-ECMWF_2015 dataset and its strength and rephrased it as follows: "*The CHE_EDGAR-ECMWF_2015 represents the 2015 global fossil CO$_2$ emissions at 0.1°×0.1° resolution that has been for the first time to our knowledge bridging the inventory community and the atmospheric modelling community. In fact, the uncertainty calculations fully respect the detailed error propagation approach recommended by IPCC (2006) guidelines for GHG inventories while the input datasets were processed such that the uncertainty information could be fully taken up by the ECMWF model IFS.*" Moreover, we emphasised in the main text that currently calculated national uncertainties are applied uniformly across each country to create a gridded map, and that these uncertainties do not take into account spatial allocation, which would be the next step of our research.

We have added extra information on other global CO$_2$ datasets that also provide uncertainty information – CDIAC, ODIAC, FFDAS and PKU-FUEL.

We have added a description on how the calculated uncertainties were specified at the grid-cell level.

The dataset description has also been relocated to a new Section 4.

It is also worth noting that the Global Emissions Initiative underlined at its recent conference on 23$^{rd}$ June 2020 the need to address the uncertainty assessment of gridded emission inputs as a crucial piece of information for atmospheric modellers and that this still requires further research efforts.

*For the actual datasets, users should be able to download these individually as needed rather than having to download everything in a large zipped folder.*
* * *
[3] Andrew, R.M: A comparison of estimates of global carbon dioxide emissions from fossil carbon sources, Earth Syst. Sci. Data, 12, 1437–1465, https://doi.org/10.5194/essd-12-1437-2020, 2020.

This has been corrected and the big folder has been split in more clear subfolders. As such, a new Zenodo link is introduced.

*Specific comments from the first few pages*

*Title – the results of this work are the uncertainties only, right? The emissions themselves are already reported by EDGAR? How much different are they from EDGAR? How will the uncertainties be used "as prior for Earth System Modelling"?*

The main result of this work are the uncertainties and reprocessing of the EDGARv4.3.2_FT2015 dataset as prior input for the ECMWF atmospheric model. The update of EDGARv4.3.2_FT2015 and difference with CHE_EDGAR-ECMWF_2015 dataset has been clearly described (incl. Table S3 in the Supplementary). Uncertainties will be used in the data assimilation part of the ECMWF IFS model.

*L12 How do emissions raise awareness? Rephrase.*

This has been rephrased as "*For an increased understanding of the $CO_2$ emission sources, patterns and trends, a link between the emission inventories and observed $CO_2$ concentrations is best established via Earth system modelling and data assimilation.*".

*L15 prior should be defined. The word prior is probably unnecessary here because the results could have more uses than just as a prior.*

This has been rephrased, avoiding the term "prior".

*L15 Are power and energy different? If not, the same word should be used.*

This has been corrected: we use energy production consequently throughout the paper.

*L17 Here and elsewhere (L25, Section 3.4) it seems misleading to say covariance and covariance matrices estimated when actually covariances are just assumed to be zero.*

This assumption is suggested by IPCC (2006) guidelines and it is currently used in our research. The main text was updated to better represent work done concerning covariance matrices.

*L18 Are the CO2 emissions really going to be included in IFS? I suspect they will be used with IFS in the CAMS reanalysis.*

These $CO_2$ emissions and uncertainties have been used in the ECMWF IFS $CO_2$ ensemble simulations (McNorton et al., 2020) and in the CHE tier-2 high resolution nature run (https://www.che-project.eu/sites/default/files/2020-01/CHE-D2-6-V1-0.pdf, Agusti-Panareda et al., in preparation). In the near future these $CO_2$ emissions and uncertainties will be used by the CAMS inversion system (currently under development).

*L21 How large are these changes to EDGAR emissions?*

Updated improved apportionment of the energy sector decreased emissions by 8 %, and the energy usage for manufacturing increased by 18 %. The extra emission source of the diffusive $CO_2$ emissions from coal mines added 7 Mtons globally but localised to few regions.

*L26-31. Hard to understand. Please give some values on the uncertainties, and describe the sensitivity tests a bit more.*

The text is revised, and values and some extra explanation on sensitivity tests are provided.

*L36 I think you mean to say "climate change" rather than "the Earth's radiative balance and climate stability".*

This has been rephrased accordingly.

*L37 'long carbon cycle" should be replaced by "fossil fuel" throughout. This definition is unclear and it would include wood.*

We have changed it to long-cycle carbon and added clear definition with reference. This is also consistent with the definitions in the paper of Janssens-Maenhout et al. (2019) on EDGARv4.3.2.

*L41 "early 2002s"?*

This has been corrected.

*L43-8. Sentence needs to be revised or deleted.*

This has been revised.

*L51 Observation not Observatory*
This has been corrected.

*L61 emissions not concentrations*
This has been corrected.

*L63 What is the Mitchell 1984 reference, and why is it cited when referring to the year 2018?*
Mitchell et al. (1984) had conversion factors for different emission units, as this can be rather easily recalculated this citation was deleted.

*L68 Andrew 2020 is not in the reference list*
The reference is added to the list.

*Table 1. FFDAS says resolution is annual, then in "Note" it says hourly. In general, the information given in "Note" for each dataset seems random.*
Table 1 was updated with more precise wording.

*L73 Global emissions are the same in CDIAC and ODIAC*
This has been corrected accordingly in the text.

*L75 3 of the datasets in Table 1 include uncertainties, according to "Note"*
Description of these three datasets and short explanation how their uncertainties were calculated are added to the main text.

*L77-8 Unclear*
We found this sentence not relevant for the explanation of our research and deleted it.

*L80 delete "with long carbon cycle"*
This has been deleted.

*L81-3 Uncertainties on a 0.1degree grid? What about your revised estimates of emissions?*
National sectoral uncertainties and revised emissions are both uniformly mapped onto a regular latitude/longitude 0.1°×0.1° resolution grid.

*L86, 90 Incomplete sentences, should start with "the"*
Theses have been corrected.

*L89 Delete – it's not true that there was a stagnation since 2015, it has increased since then.*
This sentence has been deleted.

*L100 The Paris Agreement limit is not really 1.5C*
We have rephrased the abstract.

*Page 6. This entire page is difficult to understand.*
We have revised the entire Section 2 for more clarity.

*L165-6 Needs reference*

[revised manuscript text omitted]

---

## Referee Report (RR1)

Review ESSD-2020-68

Editor requests a review, presumably because of non-availability previous reviewers. I accepted. I regret my choice. Difficult challenging paper, ruined my weekend.

Along with others, I felt pleased to see EDGAR 4.3.2 (Janssens-Maenhout et al. 2019, https://doi.org/10.5194/essd-11-959-2019) emerge into peer-reviewed open access. Prospective users certainly benefit; one hopes authors feel likewise. The current manuscript presumably responds to strong recommendation in Janssens-Maenhout et al. 2019: "EDGAR v4.3.2 grid-map uncertainties are currently the subject of scrutiny and are being further investigated under European (Horizon 2020) research projects CO2 Human Emissions (CHE, https://www.che-project.eu/) …" (Section 2.4).

My review benefits from substantial thoughtful efforts of two prior reviewers. If I may crudely simplify assessments from initial reviews, they essentially said "potentially a useful product but badly written". Authors have returned a revised manuscript. Extending over-simplification, I conclude: not clear its utility and still badly written.

I identify two major scientific weaknesses.

First, this product focuses on uncertainties derived from "anthropogenic CO2 emission inventories … processed into gridded maps" (line 15 and many other places). But, we know from other assessments, and from authors' own words, that other errors, particularly in proxy-based population distributions "will likely outweigh the combined uncertainty of activity data and emission factors" (Janssens-Maenhout et al. 2019). Therefore, current product does not in fact represent a systematic effort to understand emissions uncertainties in EDGAR or any other emissions compilation - e.g. starting from Andrew 2020, https://doi.org/10.5194/essd-12-1437-2020 as these authors do primarily in Table 7 - but rather an attempt to reprocess EDGAR product for assimilation into ECMWF's IFS. Fair enough. One presumes EU H2020 funding portended exactly that outcome. But that outcome is not a systematic uncertainty analysis but rather a data assimilation effort. Authors should inform readers from the start. As these authors conclude after comparison of their work to other products, this product "shows lowest values mainly due to the aggregation technique". They have applied skill and effort to improve aggregation for purposes of model assimilation, in this case for a single very-specific (and very skillful) model. An accurate title might read: 'Development of emission uncertainty co-variances and maps to enhance utility of EDGAR 4.3.2 for direct assimilation by ECMWF's Integrated Forecasting System'. I harp on this point because I think previous reviews also raised this issue: what exactly do these authors want to present? A data assimilation effort or a systematic uncertainty assessment? The former might fit better as a JRC or ECMWF technical report while the latter would build nicely on Janssens-Maenhout et al. 2019, compliment EDGAR 4.3.2, and - if presented as a broadly-useful product - prove interesting to ESSD readers. As presented, reader does not know what authors intend.

Second, authors seem to assume stationarity of error terms. They use as starting source data some ill-defined combination of EDGAR 2012 and EDGAR-FT 2015 (they never define or explain FastTrack, reader needs to return to JRC technical literature to learn that terminology) to produce static maps. Authors, ECMWF modelers, and readers need to assume that 2012/2015 values / error maps, however determined, apply to prior and subsequent years? The IFS, assimilating e.g. daily upper air data, will apply one standard fixed error co-variance to all CO2 observations / reports over all years? Even given large delays and even-larger structural deficiencies of UNFCCC reports, error terms from 2012 or 2015 will not pertain for 2020? Authors mention necessary corrections to coal emission data from China. In USA and perhaps Europe, sector based emissions from transport will probably (have probably?) passed energy as largest CO2 emission sources. Again, fair enough that this work derives from a single

funded project with a targeted outcome, but authors should at least admit this further limitation of their work and inform readers what authors or readers will need to do to extend this work both in impact and in utility for future use. In particular, their dismissal of wildfire sources (e.g. what they call 'short-cycle' carbon emissions and other LULUC changes based on UNFCCC definitions) will - this reviewer suggests - prove very short-sighted. Again, if they only intend to improve EDGAR for ECMWF assimilation, good enough. If they intend a broad review of emissions uncertainties, they have missed / dismissed too many factors.

Authors moved some text, composed new paragraphs, revised some tables, and corrected typographic errors but the manuscript remains very difficult to read. It continues to include redundancies, omissions, non-sequiturs (in one paragraph we read about global carbon budgets while in the next we jump to super power plants located in single grid cells); too much material distributed with a general lack of focus, very disorderly. JRC staff must include experienced technical writers; please use them. Or hire someone outside. Otherwise authors only re-arrange their own text; like most of us they evidently remain too close to project and work to recognize overall lack of focus.

Authors declare, in abstract, intent to adhere to consistent units: "uncertainties in % and kg·m-2·s-1 for each *ECMWF* group" (my emphasis, see point above about specificity to ECWMF assimilation system). Unfortunately, in handling large variety of external data as part of their discussion, they promptly violate those assertions. One finds GtC, GtCO2, MtCO2, Pg, ppm, etc. At one point (lines 67 to 69) reader encounters GtCO2 and ppm in a single sentence, with also percent expressed in two (!) significant figures. Authors should follow example of Janssens-Maenhout et al. 2019 (or Andrew 2020 or global carbon budget or most other ESSD papers): set and scrupulously adhere to consistent units and uncertainty terms. Ideally they would apply identical units and terms in text as they do in data products but one understands in this case why they might need different units. Two different units, not twenty.

One wishes that these authors had / took time to browse through rapidly-emerging emission literature. Even if - as I barely manage - they peruse only recent ESSD literature, they will find updates to India emissions, assessment (with uncertainties) of population data products, comparison of non-gridded emission source data (e.g. Andrew 2020 already mentioned), global energy imbalance (of direct relevance to emissions calculations), etc. In this product as written one gets the sense - not surprising for any of us - that in their focus to complete CHE task they missed developments occurring parallel to but outside of that project. In these very difficult days of managing health of one's self and family, one understands necessary attention to task at hand, but this reviewer senses that manuscript lacks a slightly broader outlook that would extend beyond JRC-ECMWF axis and serve to re-assure readers of quality and utility of work?

Finally, a word about IPCC. Adopting once again my crude simplification, these authors have most often answered reviewer comments with the phrase 'IPCC made us do it'. Please understand IPCC as our on-going process, subject to constant revision and improvement. If past IPCC standards begin to 'control' or 'direct' our science (as we see in impact of IPCC cutoff dates on manuscript submissions), we have put cart before horse. IPCC should exist to adopt and share progressive evolving state-of-the-art guidelines and standards. JRC, with strengths and prominence, should represent one of our community's most influential advocates for IPCC improvements. To read, instead, that these authors justify their terms, approaches, definitions (ignoring for the moment how they 'happily' distort and re-sort IPCC sectors to better fit ECMWF assimilations) based on IPCC 2006 or even IPCC 2019, disappoints this reviewer in part because one hates to see this group 'tailor' their science to meet old IPCC standards but also in part because JRC should set a good community example by challenging and contending IPCC definitions when those definitions seem orthogonal to or obsolete for science needs.

---

## Referee Report (RR2)

Substantial improvements from prior version! Credit authors for very positive changes in a new manuscript. Many comments/questions follow but overall the topic, product and description seem to definitely qualify for publication in ESSD.

Line 92: "IPCC has been addressing uncertainty from the beginning of its creation." From the beginning or from its creation but not both.

Line 94: "emissions are considered to be fully uncorrelated" uncorrelated by type, sector, NIR? Uncorrelation represents a key assumption, reader needs better explanation. Later the authors describe situations (e.g. monthly or using IPCC definitions of fuel type) where correlation exists and benefit. Some clarity initially about what they mean by 'uncorrelated' and why that is important will help many readers.

Lines 99-108, definitions and treatment of so-called super power plants. Clear discussion but reader needs to know if definitions derive from these authors or from other prior work. If other work, needs citation? Supplement implies the power plant distinctions arise from these authors but this reader suspects someone else prior has done similar assessment? E.g. the concept of 'super power plants' does not originate here? Not sure that 7.92 kg m-2 s-2 (grid number 30) to 7.85 kg m-2 s-2 (estimate) for grid number 31 represents a 'step function'? Seems rather more arbitrary?

Line 135: "emissions are on an annual national level" word missing here? Emissions are <defined>, or emissions are <reported>?

Line 142: something wrong or missing in this second line:
"$UCsector\_j, UCsector\_j < 100\% \cup UCsector\_j > 230\%$ "

Line 149: "logarithmically" or log-normally?

Line 169, Figure 1: very helpful figure but again this confusion of logarithmically versus log-normally.

Line 241, Table 2: very helpful, particularly to help readers understand how authors aggregated multiple IPCC sectors into 7 groups for IFS. Should each group have a 'total' line, to show consistency / difference of EDGAR 4.3.2_FT2015 to CHE_EDGAR_ECMWF_2015? Should Table 2 include a bottom line summary for all sectors and all groups, again for comparison purposes? E.g. to show <very minor> changes? Italics not very effective to show differences. Group 7 OTHER 1.B.1.a Coal Production 0.0 vs 7.0 - not a very big difference (e.g. not worth italicizing?)

Zenodo link works well, thank you.

Line 295, Figure 2: hard to compare WDS versus LDS because vertical scales differ substantially?

Line 304: "For example, by following Oda et al. (2019) to characterize spatial patterns of the" not a complete sentence?

Line 306, Figure 3: would this be more informative in relative (%) terms rather than in absolute (kg m-2 s-2) terms?

Line 330: word missing here?

Line 356: proof-readers will catch this but vice versa rather than vice a versa?

Line 390 and following: "≙ " pardon my ignorance but what does this symbol mean? Other readers may have same question?

Line 418, 419: "Second, try to harmonise data inclusion or omission across datasets to have more clarity in the discrepancies." Not a complete sentence?

Line 446: "CHE_EDGAR-ECMWF_2015 the highest one." Not true? Where statistically-significant differences occur (e.g. following your upper and lower limits), CHE_EDGAR-ECMWF_2015 often lower than other three? Better to emphasize this statement (Line 456, 457) "Overall, there is quite good agreement in emission budgets and uncertainties from different sources of emission data.". Statistically, this reader accepts the latter statement but not the former?

Line 459, Figure 6 (might apply to earlier figures as well): Use formal panel labels for graphics included within one Figure? Country labels here (e.g France, in standard text font) tend to get lost with page breaks.

Line 476: just to confirm (after reading prior paragraph): lower uncertainties = less uncertainty = improvement in reliability of the central estimate?

Line 497: "(i) countries total" here you need the additional apostrophe (as you used earlier): countries' total?

Line 508: "usually quite small in Megatonne." Megatonne does not need capitalization?

Line 520: "fluxes (large-scale model BIAS mitigated by biogenic CO2 flux adjustment scheme BFAS) were considered" reader needs definition of these two new acronyms?

Line 542: "checked w.r.t. their spatial location" again, proof-readers will know but I suspect Copernicus journals to not allow these colloquial abbreviations.

Additional comments:

Supplement details (in this order): super power plant definition and selection (S1); coal emissions (S2); uncertainty calculation details (S3); the CHE uncertainty tool (S4); geographic assumptions (S5); and fuel assumptions (S6). Main text refers to S1, then S3, then S4, then (not until later at Line 232) S2. Later S5 followed by S6. Does order matter here?

Use of the IFS 50-member ensemble proves efficiency and skill of approach. Will other modeling centres follow suit? If / where computational resources prove different (better or worse) would author recommend more (or fewer) groups, more (or fewer) ensemble members, etc? Recommendations seems to focus on future European developments (e.g. CoCO2) but authors should address a wider range of institutions and readers, at least with final recommendations?

---

## Referee Report (RR3)

The paper has been throughly restructured and made much clearer. As previously stated, it will be a very useful publication for the atmospheric inverse modeling field. I suggest a few corrections:

General remarks:

- Please have the native english-speaking co-authors review the language style.

- Take care of formatting in the tables, e.g. capitalizations and indenting.

I also have the following specific remarks:

- line 19: "though often limited for bottom-up anthropogenic $CO_2$ emission" not clear for reader. Better say that it is not often known or available.

- line 30: "sensitivity studies", experiments better than studies.

- Abstract: the main result of sensitivity experiments 1 and 2 should also be included.

- lines 41, 42, repeats too much the phrase "for example" (could be omitted in some cases).

- line 43: "All measurements are assimilated by global tracer transport models to infer atmospheric $CO_2$ changes, or by flux inversion systems to estimate the large-scale surface $CO_2$ fluxes. " Is not correct because of the following reasons:

  1. The atmospheric transport models do not assimilate the measurements, the inversion systems assimilates the measurements (model mole fractions from transport models can be compared to observations for manual analysis)

  2. Not all measurements can be assimilated by models, it depends on the model. Some models are not able to represent certain measurements accurately because they are too coarse.

  3. It is not just global model but regional models as well.

- Line 48: The global transport models require an initial best estimate of the CO2 emission fields with uncertainties, the so-called prior information. This Is not accurate. It is not the transport model, but the inversion that requires a prior to stabilize the calculation. Using the initial best estimate is an approach. There are other approaches such as using a yearly average, mask of emission regions, a linear model, etc.

- In page 2, there is too much use of "bottom-up", sometimes "emission inventories" would be enough

- Section 2.1 is better in the introduction except for parts of the last paragraph.

- Section 2.2.1: Not very clear what UC and AD stand for, you may consider want to consider $\sigma$ as a standard variable for uncertainty

- Section 2.2.2:

  – One more reason why sectors are merged is that some sectors have very low emissions, which are not distinguishable from a global or large regional modeling perspective.

  – It is not clear why activity and emission factor uncertainties are not log-normal themselves

- Table 3 could be included in supplemental information and further deisaggreated into different tables to make it more readable.

- Section 3.4 is more a result

- A transposed presentation of tables 4a- could make the comparison between the countries easier, as well as having the curves in the same plot in figure 2.

- Table 5 highlight *this study* so we can know the relevance of this study just by looking at the table

- Table 6 could be replaced by map in which the countries are color coded according to type.

- Section 4.2: figure 5 because we also talk of atmospheric inversions, it might not be wise to use of "inverse" type because it can lead to confusion. Maybe with instead "inverse" type use "switched"

- Section 4.3 why such an arbitrary boosting factor and not simply uncertainty propagation?

$$\sigma_{monthly} = \sqrt{\left(\frac{\sigma_{yearly}}{E_{yearly}}\right)^2 + \left(\frac{\sigma_{monthly_factor}}{monthly_factor}\right)^2} * E_{monthly} \tag{1}$$

  When aggregating the monthly emissions to yearly, the aggregated emissions should have the same uncertainty.

- Tables 9 and S8 could be clearer as 3 maps:

  1. EDGAR-JRC
  2. CHE_EDGAR-ECMWF
  3. Difference between both

---

## Author Response (AR2)

**Final Author Comments to the Topical Editor Decision**

Comments to the manuscript of Margarita Choulga et al. "Global anthropogenic CO2 emissions and uncertainties as prior for Earth system modelling and data assimilation"

Dear Editor, thank you for the constructive ideas and useful comments. We have expanded considerable effort to address all comments and to improve the manuscript in all its parts: text, figures, tables. We believe that all comments and concerns raised have now been addressed. Please find below our response to your comment. In the supplement there is the revised final version of our manuscript.

***Topical Editor Decision comments and Authors reply***

*Comments to the Author*

*Requires substantial revision, to show usefulness of underlying tools in identifying emissions uncertainties while using ECMWF modeling system as one - but not the only - example. I have corresponded with authors, they will consider these modifications.*

We thank the Editor for all useful ideas and comments, guidance and help during the manuscript rewriting. We tried to reorganise our manuscript in a proposed manner and emphasise that uncertainty calculation tool is generic and can be used by any user for any purpose, in addition we also explained how tool can be customised according to the user needs.

---

## Author Response (AR3)

**Final Author Comments to the Anonymous Referee #1 and Anonymous Referee #2**

Comments to the manuscript of Margarita Choulga et al. "Global anthropogenic $CO_2$ emissions and uncertainties as prior for Earth system modelling and data assimilation"

Dear Anonymous Referee #1 and Anonymous Referee #2, thank you for the positive evaluation and useful comments. We have addressed all comments with the aim of reaching an improved and finalised manuscript. We believe that all comments and concerns raised have now been addressed. Please find below our detailed responses to your comments.

Dear Editor, in the supplement there is the revised final version of our manuscript.

*Anonymous Reviewer #1 comments and Authors reply*

*Substantial improvements from prior version! Credit authors for very positive changes in a new manuscript. Many comments/questions follow but overall the topic, product and description seem to definitely qualify for publication in ESSD.*

We thank the reviewer for the supportive comment highlighting effort made and recommendation to publication.

*Line 92: "IPCC has been addressing uncertainty from the beginning of its creation." From the beginning or from its creation but not both.*

Rephrased to "IPCC has been addressing uncertainty from the beginning.".

*Line 94: "emissions are considered to be fully uncorrelated" uncorrelated by type, sector, NIR? Uncorrelation represents a key assumption, reader needs better explanation. Later the authors describe situations (e.g. monthly or using IPCC definitions of fuel type) where correlation exists and benefit. Some clarity initially about what they mean by 'uncorrelated' and why that is important will help many readers.*

Rephrased to "Also, the assumptions are based on IPCC (2006), so all emissions are considered to be fully uncorrelated by activity (and so by sector and by type) (i.e. all activities from IPCC (2006) are fully uncorrelated with each other), for the calculation of the uncertainty as well as of the covariance matrices.".

*Lines 99-108, definitions and treatment of so-called super power plants. Clear discussion but reader needs to know if definitions derive from these authors or from other prior work. If other work, needs citation? Supplement implies the power plant distinctions arise from these authors but this reader suspects someone else prior has done similar assessment? E.g. the concept of 'super power plants' does not originate here? Not sure that 7.92 kg m-2 s-2 (grid number 30) to 7.85 kg m-2 s-2 (estimate) for grid number 31 represents a 'step function'? Seems rather more arbitrary?*

Super power plant definition was derived from current authors and is based on super emitting energy generating point sources. These point sources are so important that their location, fuel type, capacity and operational parameters and actual conditions were checked thoroughly one by one to avoid a blow-up of uncertainties. Ranking emissions by the most emitting to the least emitting grid-cells, it was noticed that emissions decay in groups. A separation from the bulk of emission is found after the first 30 grid-cells, therefore those are labelled as super power plants providing a useful group for the statistical analysis presented in the study. The emissions of the power plants next the row after that are decaying more gradually.

*Line 135: "emissions are on an annual national level" word missing here? Emissions are <defined>, or emissions are <reported>?*

Rephrased to "emissions are reported on an annual national level".

*Line 142: something wrong or missing in this second line: "$UCsector\_j, UCsector\_j < 100\% \cup UCsector\_j > 230\%$ "*

Here mathematical symbol "∪" ("union") was used to show that this case is used when uncertainty is less than 100 % OR more than 230 %.

*Line 149: "logarithmically" or log-normally?*

Corrected to "log-normally".

*Line 169, Figure 1: very helpful figure but again this confusion of logarithmically versus log-normally.*

Corrected to "log-normally".

*Line 241, Table 2: very helpful, particularly to help readers understand how authors aggregated multiple IPCC sectors into 7 groups for IFS. Should each group have a 'total' line, to show consistency / difference of EDGAR 4.3.2_FT2015 to CHE_EDGAR_ECMWF_2015? Should Table 2 include a bottom line summary for all sectors and all groups, again for comparison purposes? E.g. to show <very minor> changes? Italics not very effective to show differences. Group 7 OTHER 1.B.1.a Coal Production 0.0 vs 7.0 - not a very big difference (e.g. not worth italicizing?)*

Extra columns with the sum per "group" were added in the table, with the difference shown in *italics*, the total sum per dataset was added in the caption of Table 2. Coal production is marked to emphasise that it was missing in EDGARv4.3.2_FT2015 and added in CHE_EDGAR-ECMWF_2015.

*Zenodo link works well, thank you.*

Thank you.

*Line 295, Figure 2: hard to compare WDS versus LDS because vertical scales differ substantially?*

Figure 2 was replotted with the same vertical and horizontal scales for both countries to ease the comparison.

*Line 304: "For example, by following Oda et al. (2019) to characterize spatial patterns of the" not a complete sentence?*

Checked – "For example, by following Oda et al. (2019) to characterize spatial patterns of the disaggregation errors in the emission maps.".

*Line 306, Figure 3: would this be more informative in relative (%) terms rather than in absolute (kg m-2 s-2) terms?*

Map of uncertainty values in % might be misleading as in some places uncertainty values can be high but emissions themselves still rather low and vice versa.

*Line 330: word missing here?*

Rephrased to "ENERGY_A (and ENERGY_S) "group" contributes the most over power plant (and super power plant) location grid-cells (e.g. South Africa).".

*Line 356: proof-readers will catch this but vice versa rather than vice a versa?*

Corrected to "vice versa".

*Line 390 and following: "≙ " pardon my ignorance but what does this symbol mean? Other readers may have same question?*

Here mathematical symbol "≙" ("correspondence") was used to show that e.g. -25 % corresponds to α = 1.5. The sentence was rephrased to "(e.g. -25.0 % ≙ α = 1.5 and -124.0 % ≙ α = 2.6; +25.0 % ≙ α = 0.8 and +124.0 % ≙ α = 1.2; ≙ means "corresponds to")".

*Line 418, 419: "Second, try to harmonise data inclusion or omission across datasets to have more clarity in the discrepancies." Not a complete sentence?*

Rephrased to "Datasets might use the same country-level information as primary input, though differences in inclusion, interpretation, and treatment of that data lead to diverse results in emissions. It is necessary to try to harmonise data inclusion or omission across datasets to have more clarity in the discrepancies.".

*Line 446: "CHE_EDGAR-ECMWF_2015 the highest one." Not true? Where statistically-significant differences occur (e.g. following your upper and lower limits), CHE_EDGAR-ECMWF_2015 often lower than other three? Better to emphasize this statement (Line 456, 457) "Overall, there is quite good agreement in emission budgets and uncertainties from different sources of emission data.". Statistically, this reader accepts the latter statement but not the former?*

The sentence "Out of the four different sources, usually UNFCCC and TNO_GHGco_v1.1 Tier 2 uncertainties are the lowest ones and CHE_EDGAR-ECMWF_2015 the highest one." refers to separate countries. After aggregation the CHE_EDGAR-ECMWF_2015 uncertainty values are usually the lowest ones.

The sentence "Overall, there is quite good agreement in emission budgets and uncertainties from different sources of emission data." refers to the emission and uncertainty values in general, meaning that there is no gross systematic overestimation or underestimation, and the values from these four different datasets do align well.

*Line 459, Figure 6 (might apply to earlier figures as well): Use formal panel labels for graphics included within one Figure? Country labels here (e.g France, in standard text font) tend to get lost with page breaks.*

Figure 6 was replotted with all needed information in the header.

*Line 476: just to confirm (after reading prior paragraph): lower uncertainties = less uncertainty = improvement in reliability of the central estimate?*

Yes, lower uncertainties = less uncertain = reported emission values are more certain, i.e. we have more certainty in the reported emission values.

*Line 497: "(i) countries total" here you need the additional apostrophe (as you used earlier): countries' total?*

Corrected to "(i) countries' total uncertainty".

*Line 508: "usually quite small in Megatonne." Megatonne does not need capitalization?*

Corrected to "megatonne".

*Line 520: "fluxes (large-scale model BIAS mitigated by biogenic CO2 flux adjustment scheme BFAS) were considered" reader needs definition of these two new acronyms?*

Corrected to "(large-scale model bias mitigated by biogenic CO2 flux adjustment scheme (BFAS))".

*Line 542: "checked w.r.t. their spatial location" again, proof-readers will know but I suspect Copernicus journals to not allow these colloquial abbreviations.*

Corrected to "with reference to".

*Supplement details (in this order): super power plant definition and selection (S1); coal emissions (S2); uncertainty calculation details (S3); the CHE uncertainty tool (S4); geographic assumptions (S5); and fuel assumptions (S6). Main text refers to S1, then S3, then S4, then (not until later at Line 232) S2. Later S5 followed by S6. Does order matter here?*

It was done on purpose to have some logical order of information in Supplementary Information sections too, not only main paper.

*Use of the IFS 50-member ensemble proves efficiency and skill of approach. Will other modeling centres follow suit? If / where computational resources prove different (better or worse) would author recommend more (or fewer) groups, more (or fewer) ensemble members, etc? Recommendations seems to focus on future European developments (e.g. CoCO2) but authors should address a wider range of institutions and readers, at least with final recommendations?*

Text added to the Section 5 Recommendations and conclusion: "The use of ensemble technique to estimate $CO_2$ uncertainties is recommended. The optimal number of ensemble members is bounded by practical considerations on computational costs. Leutbecher (2018) found a minimum of 8-member ensemble can mimic some of the skill of larger ensembles, with a 20-member ensemble being a typical value used by several modelling systems and with 50-member being a desirable target. Further grouping of anthropogenic emissions into e.g. one to reduce the dimensions of the problem is also possible with the tool CHE_UNC_APP (Choulga et al., 2021).".

**Anonymous Reviewer #2 comments and Authors reply**

*The paper has been throughly restructured and made much clearer. As previously stated, it will be a very useful publication for the atmospheric inverse modeling field. I suggest a few corrections:*

We thank the reviewer for the supportive comment highlighting the effort made and the relevance of the study.

*General remarks:*

*• Please have the native english-speaking co-authors review the language style.*

Paper has been proof-read by several English-speaking co-authors and one external proof-reader.

*• Take care of formatting in the tables, e.g. capitalizations and indenting.*

Paper has been checked for formatting in tables and figures.

*I also have the following specific remarks:*

*• line 19: "though often limited for bottom-up anthropogenic CO2 emission" not clear for reader. Better say that it is not often known or available.*

Rephrased to "though often not available for bottom-up anthropogenic $CO_2$ emissions".

*• line 30: "sensitivity studies", experiments better than studies.*

Corrected to "sensitivity experiments".

*• Abstract: the main result of sensitivity experiments 1 and 2 should also be included.*

Rephrased to "Several sensitivity experiments are performed to check: 1) the country dependence – by analysing the impact of assuming either a well- or less well-developed statistical infrastructure, 2) the fuel type dependence – by adding explicit information for each fuel type used per activity from the Intergovernmental Panel on Climate Change, and 3) the spatial source distribution dependence – by aggregating all emission sources and comparing the effect against an even redistribution over the country. The first experiment shows the SETTLEMENT group (of energy for buildings) uncertainty changes the most when development level is changed. The second experiment shows that fuel specific information reduces uncertainty in emissions only when a country uses several different fuels in the same amount, when a country mainly uses globally most typical fuel for an activity uncertainty values computed with and without detailed fuel information are the same. The third experiment highlights the importance of spatial mapping.".

*• lines 41, 42, repeats too much the phrase "for example" (could be omitted in some cases).*

Rephrased to "(in-situ from, for example, the Integrated Carbon Observation System, ICOS, air-borne e.g. aircraft campaigns, or space-borne e.g. the Orbiting Carbon Observatory, OCO-2, and the Greenhouse gases Observing Satellite, GOSAT)".

*• line 43: "All measurements are assimilated by global tracer transport models to infer atmospheric CO2 changes, or by flux inversion systems to estimate the large-scale surface CO2 fluxes. " Is not correct because of the following reasons:*

*1. The atmospheric transport models do not assimilate the measurements, the inversion systems assimilates the measurements (model mole fractions from transport models can be compared to observations for manual analysis)*

*2. Not all measurements can be assimilated by models, it depends on the model. Some models are not able to represent certain measurements accurately because they are too coarse.*

*3. It is not just global model but regional models as well.*

Rephrased to "Atmospheric measurements of $CO_2$ and co-emitted species can be assimilated into flux inversion systems to provide top-down estimates of $CO_2$ fluxes at multiple spatiotemporal scales.".

*• Line 48: The global transport models require an initial best estimate of the CO2 emission fields with uncertainties, the so-called prior information. This Is not accurate. It is not the transport model, but the inversion that requires a prior to stabilize the calculation. Using the initial best estimate is an approach. There are other approaches such as using a yearly average, mask of emission regions, a linear model, etc.*

Rephrased to "The European Centre for Medium-Range Weather Forecasts (ECMWF), for example, aims to develop an operational inversion system to estimate $CO_2$ fluxes using observed atmospheric concentrations of $CO_2$ and other relevant species.".

*• In page 2, there is too much use of "bottom-up", sometimes "emission inventories" would be enough*

Rephrased.

*• Section 2.1 is better in the introduction except for parts of the last paragraph.*

Section 2.1 contains all information on IPCC and how it was used in the current study. First paragraph is more generic and in theory could be moved to the introduction, but we think for the flow of the information it is better to keep all IPCC related information together.

*• Section 2.2.1: Not very clear what UC and AD stand for, you may consider want to consider σ as a standard variable for uncertainty*

In the Eq. (1) explanation UC stands for combined uncertainty, AD – for activity data uncertainty.

*• Section 2.2.2:*

*– One more reason why sectors are merged is that some sectors have very low emissions, which are not distinguishable from a global or large regional modeling perspective.*

Rephrased to "Usually, there are computational restrictions for operational modelling: the number of emission input fields read by the model can't be too large or emission values are too low to be distinguishable from a global or large regional modelling perspective, so some "sectors" need to be merged.".

*– It is not clear why activity and emission factor uncertainties are not log-normal themselves*

Most of uncertainty bound values are expert based and sometimes are higher than 50 % and even 100 % which leads to negative emission sampling.

*• Table 3 could be included in supplemental information and further deisaggreated into different tables to make it more readable.*

Table 3 is assumed to be rather important part of current study which shows results of the tool presented in the paper. We think Table 3 should stay in the main text.

*• Section 3.4 is more a result*

Section 3.4 is specially designed to show the flow of the tool presented in the paper – shows the intermediate results and helps to understand the tool.

*• A transposed presentation of tables 4a- could make the comparison between the countries easier, as well as having the curves in the same plot in figure 2.*

Tables 4a-c were transposed to ease the comparison between the countries. Figure 2 was replotted with the same vertical and horizontal scales for both countries, curves remained in separate plots to show additional information per curve in the header of each plot.

*• Table 5 highlight this study so we can know the relevance of this study just by looking at the table*

Current study is put in *italic*.

*• Table 6 could be replaced by map in which the countries are color coded according to type.*

In order not to expand the paper we think information in Table 6 should remain as a table.

*• Section 4.2: figure 5 because we also talk of atmospheric inversions, it might not be wise to use of "inverse" type because it can lead to confusion. Maybe with instead "inverse" type use "switched"*

Rephrased to "switched type".

*• Section 4.3 why such an arbitrary boosting factor and not simply uncertainty propagation? When aggregating the monthly emissions to yearly, the aggregated emissions should have the same uncertainty.*

Boosting factor is based on simple uncertainty propagation – if month uncertainty is combined with error propagation method yearly uncertainties are obtained.

*• Tables 9 and S8 could be clearer as 3 maps:*

*1. EDGAR-JRC*

*2. CHE EDGAR-ECMWF*

*3. Difference between both*

Tables 9 and S8 represent countries from different locations all over the world, each country has upper and lower uncertainty bounds for 7 different "groups". In order not to expand the paper we think information in Tables 9 and S8 should remain as tables.